# LEOPARD: missing view completion for multi-timepoint omics data via representation disentanglement and temporal knowledge transfer

Siyu Han [1,2,3], Shixiang Yu[1,2,3], Mengya Shi[1,2,3], Makoto Harada[1,3], Jianhong Ge[1,2,3], Jiesheng Lin[4,5], Cornelia Prehn [6], Agnese Petrera[6], Ying Li[7], Flora Sam [8,9], Giuseppe Matullo[10], Jerzy Adamski [11,12,13], Karsten Suhre [14,15], Christian Gieger [4,16], Stefanie M. Hauck [6], Christian Herder[17,18,19], Michael Roden [17,18,19], Francesco Paolo Casale [20,21,22], Na Cai[2,21], Annette Peters [3,4,5,23] & Rui Wang-Sattler [1,3,5] ✉

Longitudinal multi-view omics data offer unique insights into the temporal dynamics of individual-level physiology, which provides opportunities to advance personalized healthcare. However, the common occurrence of incomplete views makes extrapolation tasks difficult, and there is a lack of tailored methods for this critical issue. Here, we introduce LEOPARD, an innovative approach specifically designed to complete missing views in multi-timepoint omics data. By disentangling longitudinal omics data into content and temporal representations, LEOPARD transfers the temporal knowledge to the omics-specific content, thereby completing missing views. The effectiveness of LEOPARD is validated on four real-world omics datasets constructed with data from the MGH COVID study and the KORA cohort, spanning periods from 3 days to 14 years. Compared to conventional imputation methods, such as missForest, PMM, GLMM, and cGAN, LEOPARD yields the most robust results across the benchmark datasets. LEOPARD-imputed data also achieve the highest agreement with observed data in our analyses for age-associated metabolites detection, estimated glomerular filtration rate-associated proteins identification, and chronic kidney disease prediction. Our work takes the first step toward a generalized treatment of missing views in longitudinal omics data, enabling comprehensive exploration of temporal dynamics and providing valuable insights into personalized healthcare.

The rapid advancement of omics technologies has enabled researchers to obtain high-dimensional datasets across multiple views, enabling unprecedented explorations into the biology behind complex diseases[1]. Each view corresponds to a different type of omics data, or data acquired through a different platform, each contributing a partial or entirely independent perspective on complex biological systems[2]. While advancements in multi-omics measurements have increased throughput and enabled the acquisition of multiple views in a single

assay[3], data preprocessing, analysis, and interpretation remain significant and important challenges.

One of the most pressing challenges is the presence of missing data[4], which at its best (when missingness occurs at random) reduces statistical power, and at its worst (when it is not random) can lead to biased discoveries. Unlike missing data points that may be scattered across the entire dataset, a missing view refers to the complete absence of all features from a certain view, as shown in Fig. 1a. Missing views or incomplete multi-omics profiles are a common challenge, particularly in cohort studies[5–7]. In longitudinal studies that can span decades, this issue becomes increasingly common due to factors such as dropout in omics measurements, experimental errors, or unavailability of specific omics profiling platforms at certain timepoints[6,8]. The incompleteness of these datasets hinders multi-omics integration[9] and investigations into predisposing factors (such as age and genetics), enabling factors (such as healthcare service and physical activity), and

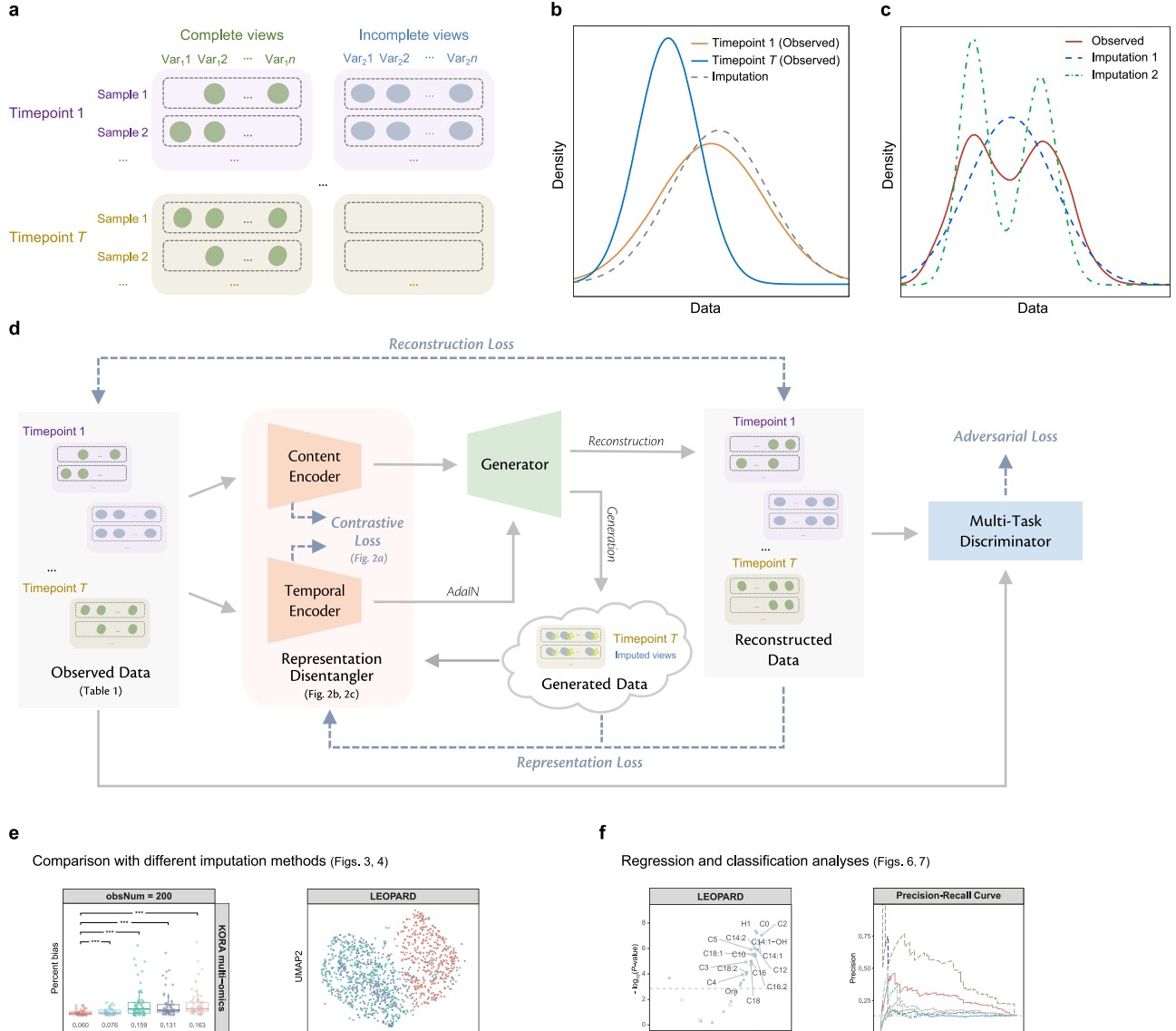

**Fig. 1 | Problem description and overview of LEOPARD architecture. a** An example of a missing view in a longitudinal multi-omics dataset. Here, some views at Timepoint $T$ are absent. The observed views may contain additional missing data points. **b** An example of data density calculated from a variable in observed data (Timepoint 1 and Timepoint $T$) and imputed data. The data density indicates a distribution shift across the two timepoints. Imputation methods developed for cross-sectional data cannot account for the temporal changes within the data, and their imputation models built with data from one timepoint, such as Timepoint 1, might not be appropriate for inferring data from another timepoint, such as Timepoint $T$. **c** Compared to Raw data, data of Imputation 1 may exhibit lower MSE than data of Imputation 2, but Imputation 1 potentially lose biological variations present in the data. **d** The architecture of LEOPARD. Omics data from multiple timepoints are disentangled into omics-specific content representation and timepoint-specific temporal knowledge by the content and temporal encoders. The generator learns mappings between two views, while temporal knowledge is injected into content

representation via the AdaIN operation. The multi-task discriminator encourages the distributions of reconstructed data to align more closely with the actual distribution. Contrastive loss enhances the representation learning process. Reconstruction loss measures the MSE between the input and reconstructed data. Representation loss stabilizes the training process by minimizing the MSE between the representations factorized from the reconstructed and actual data. Adversarial loss is incorporated to alleviate the element-wise averaging issue of the MSE loss. **e** the performance of LEOPARD is evaluated with percent bias and UMAP. The central line in the box plot represents the median. The box spans the interquartile range (IQR), and whiskers extend to values within 1.5 times the IQR. Data points outside this range are plotted as outliers. The two-sided paired Wilcoxon test is used to compare percent bias across methods. $P$-values are Bonferroni-adjusted, with significance denoted as: ns (not significant), * ($< 0.05$), ** ($< 0.01$), *** ($< 0.001$). **f** several case studies, including both regression and classification analyses are performed to evaluate if biological information is preserved in the imputed data.

biomarkers for diseases. In many medical studies, samples with incomplete omics data are removed to facilitate statistical analysis and ensure reliable results[10,11]. However, this reduces the sample size and requires the remaining dataset to be sufficient to maintain adequate statistical power.

Missing view completion refers to the estimation of missing omics data in a multi-view context. The task of missing view completion in the multi-timepoint scenario is more complex than cross-sectional missing value imputation, as it needs to harmonize data distributions across views[12] and capture temporal patterns[13]. Generic methods, such as PMM (Predictive Mean Matching)[14], missForest[15], and KNNimpute[16], learn direct mappings between views from observed data without missingness. However, this strategy is inadequate or suboptimal[17] for longitudinal data, as it precludes investigations into temporal variation, which can be of great interest: the learned mappings may overfit the training timepoints, making them unsuitable for inferring data at other timepoints, especially when biological variations cause distribution shifts over time (Fig. 1b). To address the complexities of longitudinal data, numerous effective imputation methods have been developed based on the generalized linear mixed effect model (GLMM)[18]. Existing studies have also explored the use of spline interpolation and Gaussian processes to extrapolate or interpolate missing timepoints[19–23]. However, the typically limited number of timepoints in current human cohorts can restrict the effectiveness of these longitudinal methods. Given these challenges, there is a growing need for view completion methods that are specifically designed for multi-timepoint omics data. While metrics like mean squared error (MSE) and percent bias (PB) are commonly used to evaluate imputation results[24], these quantitative metrics alone may not fully capture data quality in the context of omics data. As depicted in Fig. 1c, data imputed by method 1 may have a lower MSE than that imputed by method 2, but at a loss of biologically meaningful variations. Further case studies would be helpful to evaluate imputation methods.

In this paper, we introduce LEOPARD (missing view completion for multi-timepoint omics data via representation disentanglement and temporal knowledge transfer), a neural network-based approach that offers an effective solution to this challenge (Fig.1d). LEOPARD extends representation disentanglement[25] and style transfer[26] techniques, which have been widely applied in various contexts such as image classification[27], image synthesis[28], and voice conversion[29], to missing view completion in longitudinal omics data. LEOPARD factorizes omics data from different timepoints into omics-specific content and timepoint-specific knowledge via contrastive learning. Missing views are completed by transferring temporal knowledge to the corresponding omics-specific content.

We demonstrate the effectiveness of LEOPARD through extensive simulations using human proteomics and metabolomics data from the MGH (Massachusetts General Hospital) COVID study[30] and the KORA (Cooperative Health Research in the Region of Augsburg) cohort[31] (Fig. 1e). Additionally, we perform multiple case studies using real omics data to assess whether biological information is preserved in the imputed data, providing a comprehensive assessment of LEOPARD's performance in both regression and classification tasks (Fig.1f).

In this work, we make the following contributions:

- We propose LEOPARD, a method tailored for missing view completion in multi-timepoint omics data that innovatively applies representation disentanglement and style transfer.
- Our study shows that generic imputation methods designed for cross-sectional data are not suitable for longitudinal data, emphasizing the need for tailored approaches. Additionally, we highlight that canonical evaluation metrics do not adequately reflect the quality of imputed biomedical data. Further investigations, including regression and classification analyses, can augment these metrics in the assessment of imputation quality and preservation of biological variations.

- Our research reveals that omics data across timepoints can be factorized into content and temporal knowledge, providing a foundation for further explorations into biological temporal dynamics. This insight offers a novel perspective for predictive healthcare that extends beyond the problem of data imputation.

## Results

### Characterization of the evaluation datasets

We evaluate LEOPARD using four real longitudinal omics datasets. These distinct datasets are designed based on data variations, time spans, and sample sizes (Table 1, Methods).

Two mono-omics datasets and one multi-omics dataset are used as benchmark datasets to evaluate the performance of LEOPARD and established methods. The mono-omics datasets are constructed with the proteomics data from the MGH COVID study and the metabolomics from the KORA cohort, respectively. Views in both datasets correspond to panels or biochemical classes. Missingness in these datasets exemplifies a common issue encountered in longitudinal studies where data from certain panels or biochemical classes are incomplete in some but not all timepoints. The third dataset is a multi-omics dataset consisting of both metabolomics and proteomics data from the KORA cohort. In this dataset, views correspond to different omics. This dataset exemplifies the situation where data of a type of omics is incomplete. These three datasets comprise data of two views or view groups (v1 and v2) from two timepoints (t1 and t2). The samples from each dataset are split into training, validation, and test sets in a 64%, 16%, and 20% ratio, respectively. We use $\mathcal{D}_{v,t}^{\text{split}}$ to denote data split from different views and timepoints. The test data in v2 at t2 (i.e. $\mathcal{D}_{v=v2,\,t=t2}^{\text{test}}$) are masked for performance evaluation.

To further assess LEOPARD's applicability to multi-timepoint omics data, the Extend KORA metabolomics dataset is constructed to span three timepoints. The test data $\mathcal{D}_{v=v2,\,t=t1}^{\text{test}}$ and $\mathcal{D}_{v=v1,\,t=t3}^{\text{test}}$ are masked individually for evaluation (Methods).

### CGAN architecture as a reference method

Existing neural network-based missing view completion methods[32–34] have shown remarkable performance in the field of computer vision. Given their inapplicability to omics data, we designed a conditional generative adversarial network (cGAN) model specifically tailored for omics data, as a reference method. This architecture is inspired by VIGAN (View Imputation via Generative Adversarial Networks)[35] and a method proposed by Cai et al.[36], both initially designed for multi-modality image completion.

In the training phase, the generator of the cGAN is trained on observed data from the training set to capture the mappings between two views. The discriminator guides the generator to produce data with a distribution similar to that of actual data. The discriminator also has an auxiliary classifier[37] to ensure the generated data can be paired with input data (Methods). In the inference phase, the generator utilizes the mappings it has learned from the observed data to impute the missing view in the test set. Compared to methods PMM and missForest, our cGAN model has the potential to learn more complex mappings between views. However, these three methods are not able to capture temporal changes within longitudinal data and can only learn from samples where both views are observed.

### Overview of LEOPARD architecture

Instead of relying on learning direct mappings between views, LEOPARD captures and transfers temporal knowledge to complete missing data, which also enables it to learn from samples even when only one view is available. As illustrated in Fig. 1d, the LEOPARD architecture comprises several hierarchical components. First, data of each view are transformed into vectors of equal dimensions using corresponding pre-layers. Subsequently, omics data of all views are decomposed into content and temporal representations. The content encoder captures

**Table 1 | Summary of the datasets used in this study**

| | MGH COVID proteomics | KORA metabolomics | KORA multi-omics | Extended KORA metabolomics |
|---|---|---|---|---|
| **Omics type** | proteomics | metabolomics | metabolomics; proteomics | metabolomics |
| **View** | | | | |
| v1 | panel: Explore 384 cardiometabolic | biochemical class[*]: GPL | omics: metabolomics | biochemical class[*]: GPL |
| v2 | panel: Explore 384 inflammation | biochemical class[*]: AC, SL, AA, MS | omics: proteomics | biochemical class[*]: AC, SL, AA, MS |
| **Variable number** | Explore 384 cardiometabolic: 322, Explore 384 inflammation: 295 | GPL: 70, AC: 15, SL: 11, AA: 9, MS: 1 | metabolites: 104, proteins: 66 | GPL: 69, AC: 14, SL: 11, AA: 9, MS: 1 |
| **Timepoint** | | | | |
| t1 | D0 | F4 | S4 | S4 |
| t2 | D3 | FF4 | F4 | F4 |
| t3 | | | | FF4 |
| **Time span** | 3 days | 7 years | 7 years | 14 years in total |
| **Sample number** | | | | |
| Total (100%) | 218 | 2085 | 1062 | 614 |
| Training (64%) | 140 | 1335 | 680 | 393 |
| Validation (16%) | 35 | 333 | 170 | 98 |
| Test (20%) | 43 | 417 | 212 | 123 |
| **Masked data for completion** | inflammation proteins from D3 | non-GPL metabolites from FF4 | proteins from F4 | completion 1: Non-GPL metabolites from S4;completion 2: GPL metabolites from FF4 |

[*]The targeted metabolites from KORA are classified into five analyte classes, namely glycerophospholipid (GPL), acylcarnitine (AC), sphingolipid (SL), amino acid (AA), and monosaccharide (MS). One view group comprises metabolites from the glycerophospholipid class, while the other view group comprises metabolites from the other classes.

the intrinsic content of the views, while the temporal encoder extracts knowledge specific to different timepoints. A generator then reconstructs observed views or completes missing views by transferring the temporal knowledge to the view-specific content using Adaptive Instance Normalization (AdaIN)[26]. Lastly, we use a multi-task discriminator to discriminate between real and generated data (Methods).

The LEOPARD model is trained by minimizing four types of losses: contrastive loss, representation loss, reconstruction loss, and adversarial loss. An ablation test is performed to evaluate the contribution of each loss (Methods). Minimizing the normalized temperature-scaled cross-entropy (NT-Xent)-based contrastive loss[38] optimizes the factorization of data into content and temporal representation. For both representations, minimizing the contrastive loss brings together the data pairs from the same view or timepoint and pushes apart the data pairs from different ones, so that the encoders learn similar intrinsic content (or temporal knowledge) across timepoints (or views). The representation loss, also computed on content and temporal representations, measures the MSE of the representations factorized from the actual and reconstructed data. LEOPARD minimizes this loss based on the intuition that the representations of the actual and reconstructed data should be alike. The reconstruction loss measures the MSE between imputed and observed values. Previous studies[39–41] have demonstrated that the optimization of MSE loss often results in averaged outputs, leading to blurring effects when generating images. In our context, this might diminish biological variations present in omics data. To alleviate this issue, we use adversarial loss to encourage the predicted distribution to align more closely with the actual distribution.

LEOPARD has three unique features compared to conventional architectures for multi-view data completion. First, instead of focusing on direct mappings between views, which can only be learned from paired data where both views are present, LEOPARD formulates this imputation task in terms of representation learning and style transfer.

This allows LEOPARD to utilize all available data, including observations present in only one view or timepoint. Second, it incorporates contrastive loss to disentangle the representations unique to views and timepoints, which enables the model to learn more generalized and structured representations. Our experiments show that this is of importance to improve data quality. Third, its multi-task discriminator solves multiple adversarial classification tasks simultaneously by yielding multiple binary prediction results, which has been proven to be more efficient and effective than a discriminator for a multi-class classification problem[42].

## The representation disentanglement of LEOPARD

We use the KORA multi-omics dataset to examine if LEOPARD can effectively disentangle content and temporal representations from omics data. In this analysis, the model is trained for 600 epochs to ensure that the contrastive loss stabilizes and reaches full saturation (Fig. 2a). We use the uniform manifold approximation and projection (UMAP)[43] for visualizing the content and temporal representations of the validation set across different views and timepoints. As the training progresses, we expect similar representations to gradually cluster together in UMAP, while dissimilar ones form distinct clusters.

As the model trains, the contrastive loss decreases (Fig. 2a), indicating that LEOPARD is increasingly able to encode the representations for different views and timepoints. The content representation embeddings (Fig. 2b) of observed v1 (in blue) and v2 (in red) separate rapidly during training (epoch 5), but those of the imputed v2 for t2 (in light red) do not mix with those of the observed v2 at t1 (in dark red). This suggests that while LEOPARD can distinguish between v1 and v2 after only a few training epochs, it is not yet capable of producing high-quality v2 for t2 that have similar content information as the observed v2 at t1. After 30 epochs of training, the content representation of v2 at t2 is better encoded, with its embeddings mixing with those of v2 at t1. Similar trends are observed in the temporal representations (Fig. 2c), where

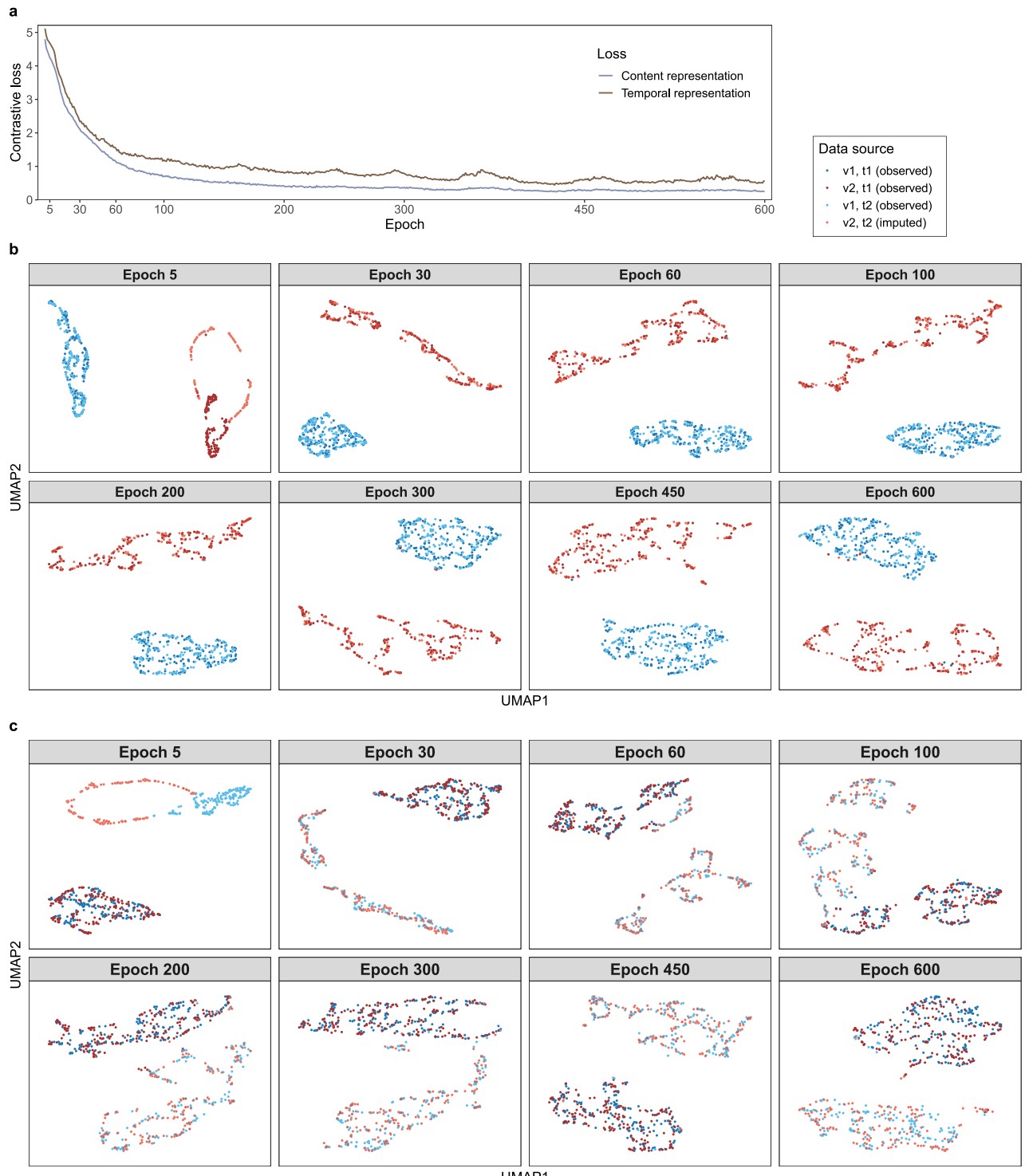

**Fig. 2 | The representation disentanglement process of LEOPARD on the KORA multi-omics dataset. a** The normalized temperature-scaled cross-entropy (NT-Xent)-based contrastive loss is computed for content and temporal representations. **b–c** Uniform manifold approximation and projection (UMAP) embeddings of content (**b**) and temporal (**c**) representations at various training epochs are visualized for the KORA multi-omics dataset's validation set. Representations encoded from data of v1 and v2 (metabolomics and proteomics, depicted by blue and red dots) at timepoints t1 and t2 (S4 and F4, depicted by dark- and light-colored dots) are plotted. The data of v2 at t2 are imputed data produced after each training epoch, while the other data are from the observed samples in the validation set.

LEOPARD's content and temporal encoders capture signals unique to omics-specific content and temporal variations. In **b**, as the training progresses, one cluster is formed by the data of v2 at t1 and t2 (dark and light red dots), while the other cluster is formed by the data of v1 at t1 and t2 (dark and light blue dots), indicating that the content encoder is able to encode timepoint-invariant content representations. Similarly, in (**c**), embeddings from the same timepoint cluster together. One cluster is formed by the data of v1 and v2 at t1 (dark blue and red dots), and the other is formed by the data of v1 and v2 at t2 (light blue and red dots). This demonstrates that LEOPARD can effectively factorize omics data into content and temporal representations. Source data are provided as a Source Data file.

embeddings of each timepoint (t1 and t2 in dark and light colors, respectively) gradually form their corresponding clusters as training progresses. We notice that the temporal representations take more epochs than the content representations to form distinct clusters. Even after 100 epochs, some temporal representation embeddings of t1 are still mixed with those of t2. However, after around 450 epochs, LEOPARD is demonstrably able to encode temporal information that is unique to t1 and t2 (Fig. 2c).

### Approach for benchmarking LEOPARD against conventional methods

Due to the lack of established methods specifically designed for missing view completion in multi-timepoint omics datasets, we benchmark LEOPARD against three widely recognized generic imputation methods: missForest, PMM, and GLMM, as well as a cGAN model designed for this study. The cGAN serves as a reference model to demonstrate how existing neural network approaches, typically suited for cross-sectional data, perform in longitudinal scenarios. MissForest, as a representative non-parametric method, was chosen for its robustness and ability to handle complex, non-linear relationships among variables. PMM and GLMM, both implemented within the MICE (Multivariate Imputation by Chained Equations)[44] framework, represent established multiple imputation methods that not only address missing values but also allow for the assessment of imputation uncertainty. GLMM, with its ability to capture temporal patterns inherent in longitudinal data, is particularly advantageous for data imputation in longitudinal scenarios.

We assess the performance of LEOPARD, cGAN, missForest, PMM, and GLMM on $\mathcal{D}^{\text{test}}_{v=v2, t=t2}$ of each of our benchmark datasets: MGH COVID proteomics, KORA metabolomics, and KORA multi-omics datasets. During training, the methods build models using data from $\mathcal{D}^{\text{train}}_{v=v1, t=t1}$, $\mathcal{D}^{\text{train}}_{v=v2, t=t1}$, and $\mathcal{D}^{\text{train}}_{v=v1, t=t2}$, along with different numbers of training observations (obsNum) from the data block to be completed (i.e. $\mathcal{D}^{\text{train}}_{v=v2, t=t2}$ in this case). By varying obsNum, we assess how additional observed data from t2 affects their imputation performances. When obsNum is zero, data of v2 at t2 are assumed as completely missing, and GLMM cannot be trained due to limited longitudinal information. In this scenario, we train a linear model (LM)[45] to complete the missing view (Methods). This additionally allows us to evaluate how the performance of GLMM compares to that of the simpler LM method.

Two evaluation metrics, PB and UMAP visualization, are used for performance evaluation. PB quantifies the median absolute error ratio between the observed and imputed values for each variable, offering a variable-level assessment of imputation performance (Methods). In contrast, UMAP visualization illustrates overall similarities between observed and imputed datasets, providing a dataset-level evaluation. For the multiple imputation methods, each individual imputation is also evaluated using PB (Supplementary Fig. 1) and UMAP (Supplementary Fig. 2).

Apart from missing views, the presence of missing data points in observed views is also very common in omics analysis. Therefore, LEOPARD is designed to tolerate a small number of missing data points in the observed views (Methods). The benchmark datasets are further used to evaluate the performance of different methods when observed views contain missing values. We simulate missing values by randomly masking 1%, 3%, 5%, 10%, and 20% of the data in the observed views (maskObs) under the assumption that data points are missing completely at random (MCAR). The experiment is repeated 10 times for each specified proportion, and the results are evaluated by PB and UMAP. The PB values are averaged across 10 repetitions. The UMAP plots visualize the repetition that exhibits the lowest median of PB. Due to the exceptionally high computational demands of PMM and GLMM on the MGH COVID proteomics dataset, these two methods are not evaluated in this scenario.

### Evaluation results on the mono-omics datasets

For the MGH COVID proteomics dataset, missForest overall exhibits the lowest PB, whereas LEOPARD performs worse than missForest and its neural network-based counterpart, cGAN (Fig. 3, upper row). When compared to LM, GLMM does not show improved performance. As obsNum increases, the PB values of all methods tend to decrease. Specifically, when obsNum is 100, the UMAP representation (Fig. 4, upper row) reveals that the clusters of the imputed data generated by all five methods (green dots) closely approximated the actual data (blue dots), indicating high similarity between the imputed and original datasets.

Interestingly, we observe that missForest, despite yielding the best performance for the MGH COVID dataset, produces unstable results for the KORA metabolomics dataset with large interquartile range (IQR) values. When obsNum is 0, missForest displays the largest IQR of 0.186 with a median of 0.205, though it is not significantly different from LEOPARD after Bonferroni adjustment (Fig. 3, middle row). LEOPARD achieves the smallest IQR of 0.094 with a median of 0.209 under the same condition, while cGAN, PMM, and LM obtain IQR values of 0.125, 0.132, and 0.166, with corresponding medians of 0.229, 0.253, and 0.254, respectively. As obsNum increases to 200, LEOPARD, cGAN, missForest, and PMM lower their median PB values to 0.142, 0.172, 0.177, and 0.204, respectively. However, GLMM obtains a median PB of 0.291 and does not outperform LM. From the UMAP plots generated from the data imputed under obsNum = 200, we notice a large amount of the embeddings from $\hat{\mathcal{D}}^{\text{test}}_{v=v2, t=t2}$ generated by missForest and PMM are mixed with those of $\mathcal{D}^{\text{test}}_{v=v2, t=t1}$, instead of $\mathcal{D}^{\text{test}}_{v=v2, t=t2}$, implying that they overfit to t1 and do not generalize well to the second timepoint (Fig. 4, middle row). Moreover, the UMAP embeddings of $\hat{\mathcal{D}}^{\text{test}}_{v=v2, t=t2}$ from missForest, PMM, and GLMM only partly overlap with those of $\mathcal{D}^{\text{test}}_{v=v2, t=t2}$, suggesting that some variations in the observed data have not been captured. In contrast, the embeddings of the data imputed by LEOPARD widely spread within the embedding space of the observed data, which demonstrates that LEOPARD has effectively learned and approximated the observed data's distribution.

### Evaluation results on the KORA multi-omics dataset

In contrast to mono-omics datasets, where both views are from the same omics type, multi-omics datasets require imputation methods to capture more intricate relationships between omics data to ensure accurate results. In our evaluation of the KORA multi-omics data, all methods show some extremely high PB values when obsNum is 0 (Fig. 3, lower row). Additionally, under this condition, there are no significant differences in the results of LEOPARD, cGAN, and missForest. As obsNum increases to 200, LEOPARD greatly reduces its median PB from 0.161 to 0.060, outperforming its closest competitor, cGAN, which reduces its median PB from 0.158 to 0.076. In contrast, the performances of missForest (from 0.156 to 0.159) and PMM (from 0.177 to 0.131) do not show similar improvements. GLMM reduces its median PB from 0.176 to 0.164 as obsNum increases from 50 to 200. The UMAP visualization further reveals a limited ability of missForest, PMM, and GLMM to capture signals from the t2 timepoint, as their embeddings of $\hat{\mathcal{D}}^{\text{test}}_{v=v2, t=t2}$ cluster with $\mathcal{D}^{\text{test}}_{v=v2, t=t1}$, not $\mathcal{D}^{\text{test}}_{v=v2, t=t2}$ (Fig. 4, lower row). LEOPARD's performance is further validated by a high similarity between the distributions of the imputed and observed data embeddings in the UMAP space.

### Analysis on imputed data with extremely high PB values

We then investigate the extremely high PB values (>0.8) observed in the KORA multi-omics dataset. Under obsNum = 0, we notice that proteins with low abundance ( < 4.0) tend to exhibit extremely high PB in the imputed values (Fig. 5). For instance, SCF (stem cell factor), with a median abundance of 9.950, has a PB of 0.090 calculated from the LEOPARD-imputed data. In contrast, NT3 (Neurotrophin-3), with a

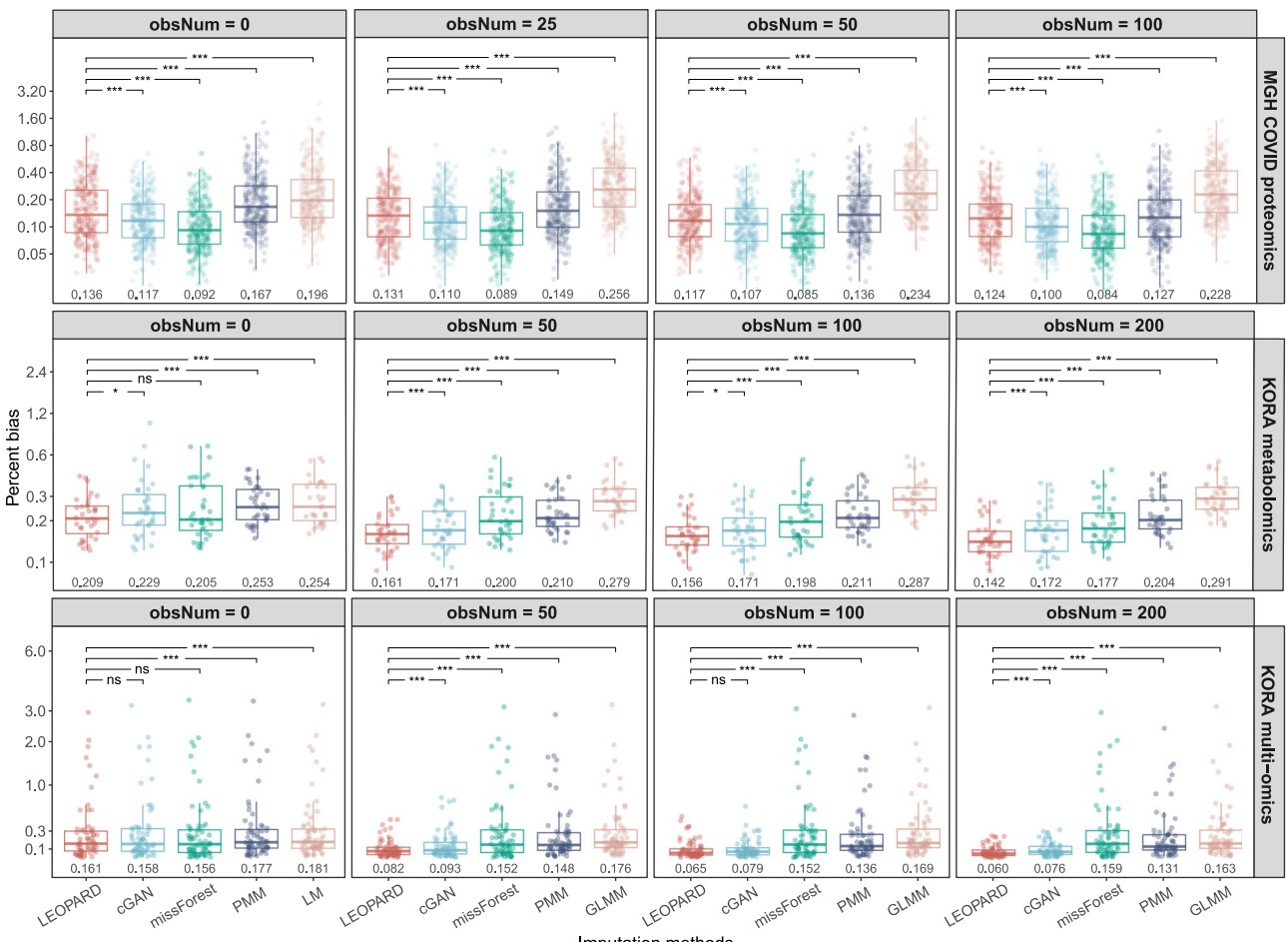

**Fig. 3 | Percent bias of imputed results for the test sets of three benchmark datasets.** Percent bias is evaluated on $\mathcal{D}^{test}_{v=v2,\,t=t2}$ of the three benchmark datasets: MGH COVID proteomics dataset (upper row), KORA metabolomics dataset (middle row), and KORA multi-omics dataset (lower row), under various numbers of training observations (obsNum) from the data block to be completed. Please note that LM is used for imputation instead of GLMM when obsNum = 0. Each dot in the plots represents a percent bias value for a variable. The value below each box indicates the median, which is also represented by the central line in each box plot. The box extends from the first quartile to the third quartile, capturing the interquartile range (IQR). Whiskers extend to the smallest and largest values within 1.5 times the IQR from the quartile boundaries. Data points outside this range are plotted as outliers. The two-sided paired Wilcoxon test is used to compare percent bias across methods, with LEOPARD as the reference group. *P*-values are adjusted for multiple comparisons using the Bonferroni method, and significance is annotated based on cutpoints: not significant (ns), *P* < 0.05 (*), *P* < 0.01 (**), and *P* < 0.001 (***). Source data are provided as a Source Data file.

much lower median abundance of 0.982, shows a PB of 1.187 calculated from the same imputed data. Increasing obsNum can substantially lower these extremely high PB values for LEOPARD and cGAN, but makes no similar contributions for missForest, PMM, and GLMM.

### Evaluation on observed views with missing data points

We further evaluate how different methods perform when observed views contain missing values. Evaluated on the KORA metabolomics dataset, which has the largest sample size, our findings indicate LEOPARD and missForest are robust to the missing data points in terms of PB (Supplementary Fig. 3). In contrast, cGAN and GLMM exhibit high sensitivity to those missing values. Method cGAN does not show similar improvement with the increase of obsNum as it performs in Fig. 3 (middle row) and is gradually surpassed by missForest. GLMM overall exhibits higher PB than the other methods. The UMAP plots (Supplementary Fig. 4) further demonstrate that LEOPARD's performance remains comparable to scenarios with no missing data in the observed views (Fig. 4, middle row), unlike the other methods which display overfitting or a great loss of data variation. Although LEOPARD outperforms other methods, we observed a change in the distribution of the imputed data (blue dots): as maskObs increases, these blue dots begin to shrink toward their center and become more concentrated.

This leads to a reduced coverage of the outer areas of the ground truth embeddings (green dots) and suggests that the imputed data might not capture the full variability of the data when the proportion of missing data is high. The results on the MGH COVID proteomics and the KORA multi-omics datasets are shown in Supplementary Figs. 5–8.

### Case studies for regression analysis on the KORA-derived datasets

Several case studies, covering both regression and classification tasks, are performed to investigate whether biological signals are preserved in the imputed data obtained at obsNum = 0. The regression models are fitted using the observed data and different imputed data corresponding to $\mathcal{D}^{test}_{v=v2,\,t=t2}$. To assess the robustness of the imputation methods in preserving original data characteristics, we evaluate the consistency of the effect signs, the Spearman correlation of effect sizes, and the agreement of significant variables between imputed and observed data. The performance of LEOPARD, cGAN, and missForest are evaluated on their imputed data directly, while two multiple imputation methods, PMM and LM, are evaluated by pooling their multiple estimates using Rubin's rules[45] (Methods).

We use the KORA metabolomics dataset to identify metabolites associated with age, controlling for sex. The models are fitted

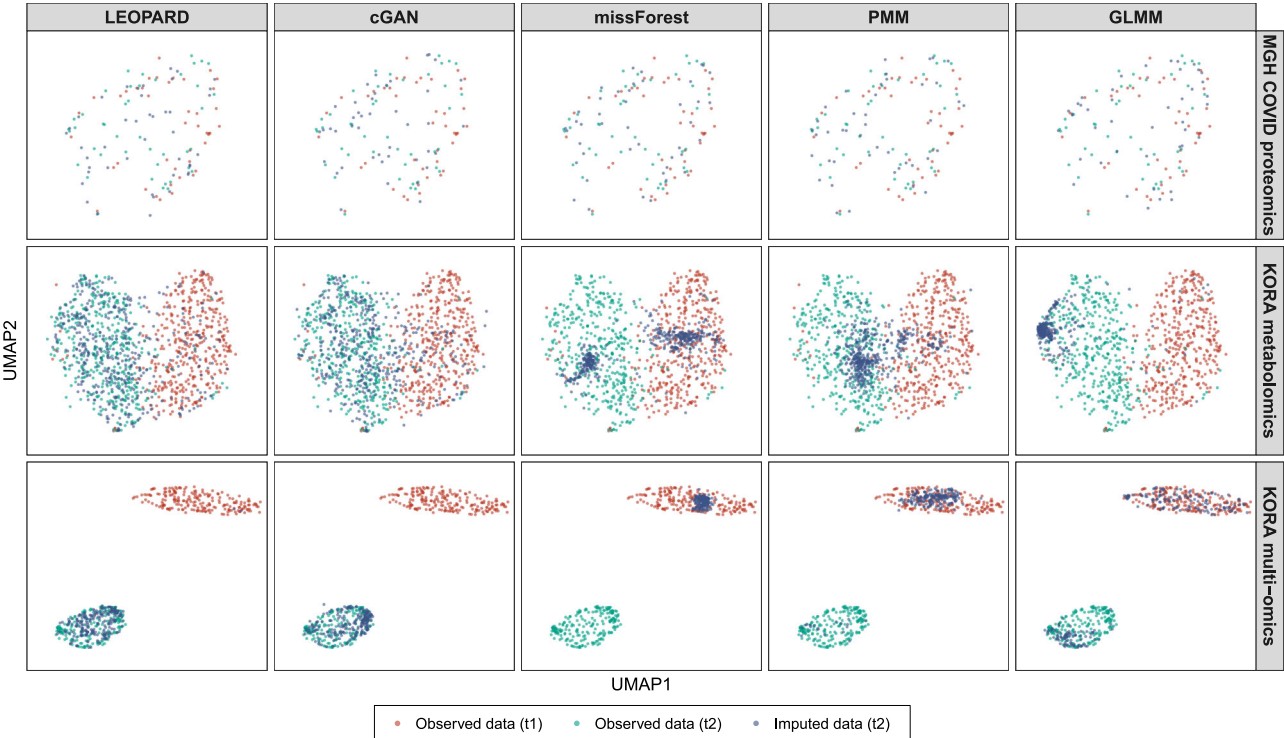

**Fig. 4 | UMAP representations of the imputed values and corresponding observed data from the benchmark datasets.** Uniform manifold approximation and projection (UMAP) models are initially fitted with the training data from the MGH COVID proteomics dataset (upper row, t1: D0, t2: D3), KORA metabolomics dataset (middle row, t1: F4, t2: FF4), and KORA multi-omics dataset (lower row, t1: S4, t2: F4). Subsequently, the trained models are applied to the corresponding observed data (represented by red and green dots for t1 and t2) and the data imputed by different methods (represented by blue dots) under the setting of obsNum = 100 for the MGH COVID dataset and obsNum = 200 for the two KORA-derived datasets. The distributions of red and green dots illustrate the variation across the two timepoints, while the similarity between the distributions of blue and green dots indicates the quality of the imputed data. A high degree of similarity suggests a strong resemblance between the imputed and observed data. Source data are provided as a Source Data file.

separately to each of the 36 metabolites ($N = 417$). LEOPARD and cGAN each demonstrate 88.9% of metabolites with matching effect signs, followed by LM 75.0%, and both missForest and PMM 69.4%. The Spearman correlation between effect sizes from the observed and imputed data also varies across methods, with LEOPARD showing the highest correlation of 0.708, followed by PMM (0.440), cGAN (0.319), and GLMM (0.110). In contrast, missForest shows a negative correlation of −0.074. Of the 18 metabolites significantly associated with age after a Bonferroni correction for multiple testing ($P < 0.05/36$) in the observed data, 17 are also significant in the data imputed by LEOPARD (see Fig. 6a). Among these 17 metabolites, several, including C14:1 (Tetradecenoylcarnitine), C18 (Octadecanoylcarnitine), C18:1 (Octadecenoylcarnitine), and Orn (Ornithine), have been validated by previous research[46–49] showing that they might be particularly relevant in aging and age-related metabolic conditions. In contrast, only one metabolite is significantly associated with age in the data imputed by missForest. No metabolite is identified as significant in the data imputed by cGAN, PMM, and LM. The results on each imputation of PMM and LM are shown in Supplementary Fig. 9.

We then use the KORA multi-omics dataset to identify proteins associated with the estimated glomerular filtration rate (eGFR), controlling for age and sex. Each model is individually fitted to one of the 66 proteins ($N = 212$). The percentages of proteins with matching effect signs across different methods are as follows: LEOPARD demonstrates 92.4%, missForest 87.9%, cGAN 74.2%, PMM 68.2%, and LM 57.6%. LEOPARD obtains the Spearman correlation score of 0.539, followed by LM (0.473), PMM (0.421), missForest (0.317), and cGAN (0.277). In the observed data, 28 proteins are significantly associated with eGFR after a Bonferroni correction ($P < 0.05/66$). Of these, 10 remain

significant in the data imputed by LEOPARD (see Fig. 6b), while one is significant in the data from cGAN, and none are identified as significant in the data from missForest, PMM, and LM. Among the 10 proteins detected from the LEOPARD-imputed data, eight (TNFRSF9, IL10RB, CSF1, FGF21, HGF, IL10, CXCL9, and IL12B) have been validated by prior research[50]. The results on each imputation of PMM and LM are shown in Supplementary Fig. 10.

## Case studies for classification analysis on the KORA-derived datasets

For the classification tasks, we employ balanced random forest (BRF)[51] classifiers to predict chronic kidney disease (CKD) using both the KORA metabolomics and multi-omics datasets. The classifiers are individually fitted using the observed and different imputed data of $\mathcal{D}^{\text{test}}_{v=v2, t=t2}$, corresponding to 36 metabolites from the KORA metabolomics dataset ($N = 416$, one sample removed due to a missing CKD label) and 66 proteins from the KORA multi-omics dataset ($N = 212$). CKD cases are defined as having an eGFR <60 mL/min/1.73m²[52]. In the two datasets, 56 and 36 individuals are identified as CKD cases, respectively. We train the classifiers using identical hyperparameters and use leave-one-out-cross-validation (LOOCV) to evaluate their performance. The models for LEOPARD, cGAN, and missForest are trained using their respective imputed data, while the models for PMM and LM are trained on the average estimates across their multiple imputations (Methods).

For the KORA metabolomics dataset, the observed data obtain an F1 Score of 0.439, and the data imputed by LEOPARD achieves the closest performance with an F1 Score of 0.358 (Fig. 7a, Supplementary Table 1). LEOPARD also outperforms its competitors in terms of accuracy, sensitivity, precision, AUROC (area under the

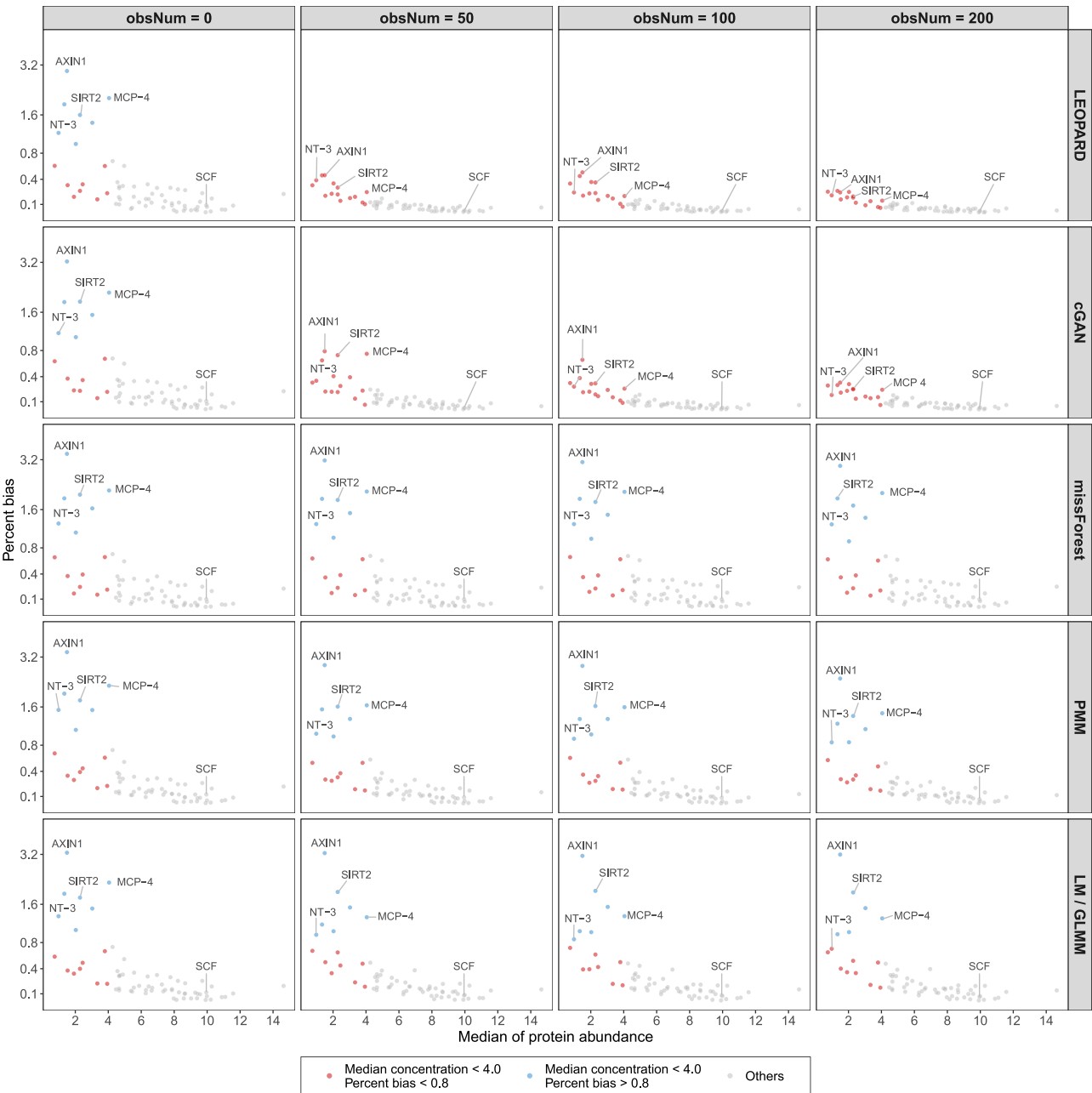

**Fig. 5 | Proteins with low abundance tend to exhibit high percent bias in the imputed values.** The proteins with low abundance (median concentration <4.0) tend to exhibit extremely high percent bias (> 0.8) in the imputed values obtained under numbers of training observations (obsNum) is zero. The extremely high percent bias values of LEOPARD can be lowered by increasing obsNum. Please note that LM is used for imputation instead of GLMM when obsNum = 0. Source data are provided as a Source Data file.

receiver operating characteristic curve), and AUPRC (area under the precision-recall curve). The proteins from the KORA multi-omics dataset perform better than the metabolites from the metabolomics dataset for this task. The F1 Score increases to 0.544 for the observed data of the KORA multi-omics dataset. LEOPARD outperforms its competitors with an F1 Score of 0.403, an AUROC of 0.725, and an AUPRC of 0.435 (Fig. 7b, Supplementary Table 2). The prediction results on each individual imputation of PMM and LMM are displayed in Supplementary Figs. 11 and 12.

**Case studies on the MGH COVID proteomics dataset**
We additionally use the observed and imputed data corresponding to $\mathcal{D}^{\text{test}}_{v=v2,\,t=t2}$ ($N = 43$) of the MGH COVID dataset to identify proteins associated with neutralization levels and to predict neutralization

levels. These analyses are adapted from those conducted in the original study of this dataset[30].

For each protein, we fit a logistic regression model using the proteomics data as the predicator and neutralization levels as the response. Bonferroni correction is used for multiple tests ($P < 0.05/$ 295). Surprisingly, no proteins reach statistical significance even in the observed data (Supplementary Fig. 13 a), which contradicts the results of the original study. The 10 proteins with the lowest $P$-values are highlighted in the volcano plots. We observe that patterns in the missForest-imputed data are more similar to those in the observed data compared to data imputed by other methods.

For the prediction of neutralization level, we use the proteomics data from Day 3 to predict the same day's neutralization levels. We build BRF models using the same settings as in our previous

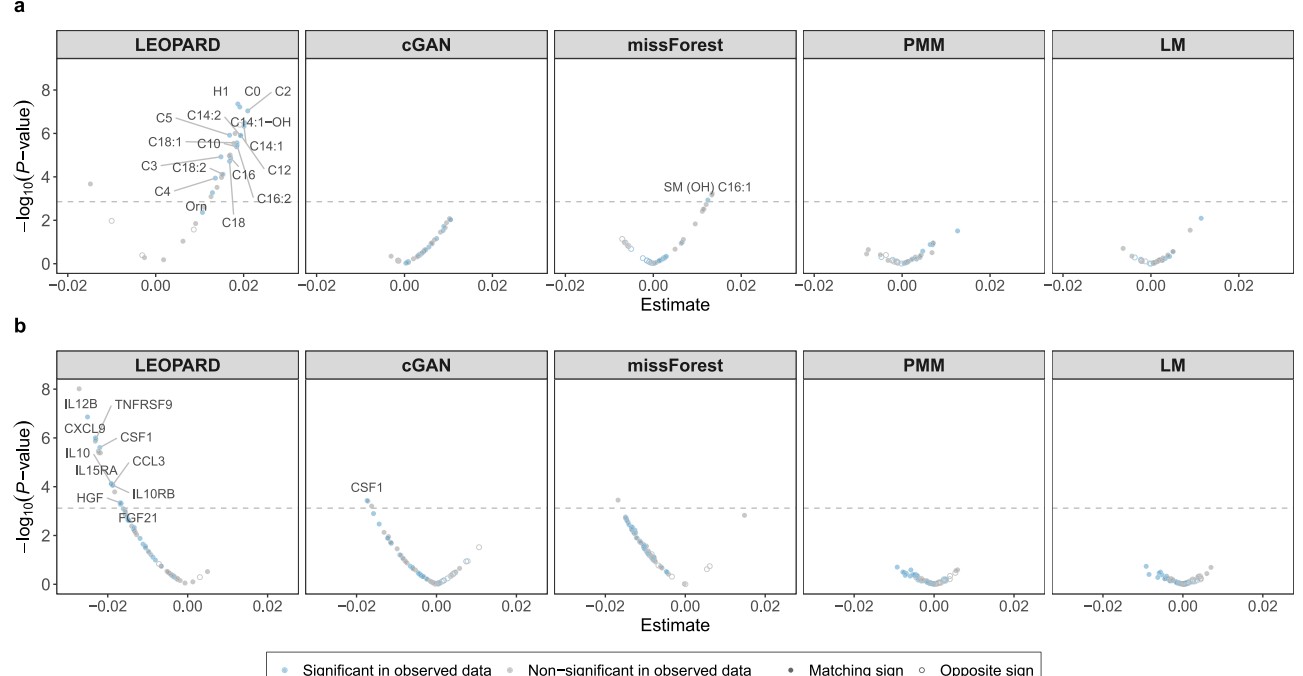

**Fig. 6 | Regression analyses with the data imputed by different methods.**
**a** Volcano plots display age-associated metabolites detected in the $\mathcal{D}^{\text{test}}_{v=v2,\,t=t2}$ and $\hat{\mathcal{D}}^{\text{test}}_{v=v2,\,t=t2}$ (obsNum = 0) of the KORA metabolomics dataset ($N = 417$). Associations are assessed using linear regression, with $P$-values adjusted for multiple comparisons via the Bonferroni method. 18 significant metabolites ($P < 0.05/36$) identified in the observed data are shown in blue. Replicated metabolites from the data imputed by different methods are marked with labels. Solid dots represent variables where the observed and imputed data have matching signs for the estimate, while hollow dots represent mismatched signs. **b** Volcano plots display eGFR-associated proteins detected in the $\mathcal{D}^{\text{test}}_{v=v2,\,t=t1}$ and $\hat{\mathcal{D}}^{\text{test}}_{v=v2,\,t=t2}$ (obsNum = 0) of the KORA multi-omics dataset ($N = 212$). Associations are also tested using linear regression with Bonferroni-adjusted $P$ values. 28 significant metabolites ($P < 0.05/66$) identified in the observed data are shown in blue. Replicated metabolites from the data imputed by different methods are marked with labels. Solid dots indicate sign matches between the observed and imputed data, while hollow dots indicate mismatches. Source data are provided as a Source Data file.

experiments. The model built on the observed data achieved an AUROC of 0.776 (Supplementary Fig. 13b), while the models trained on missForest- and PMM-imputed data outperform the observed data and achieve AUROC of 0.788 and 0.779, respectively.

The discrepancies between our findings and those of the original study might result from the limited sample size of our analysis. While the original study analyzed the complete dataset of 218 samples, our evaluation is restricted to a test set of only 43 samples. This substantial reduction in sample size can limit the statistical power, leading to non-significant results and inconsistent findings. Additionally, single imputations tend to underestimate the original variation in the data, potentially resulting in better predictive performance than observed data.

These findings highlight the challenges associated with performing and evaluating imputation, as well as the potential biases introduced by imputation methods. While these results should be interpreted with caution and may not reflect underlying biological reality, we include them to emphasize the complexities of imputation and to support reproducibility in scientific research.

**Minimum training samples for robust view completion**
To reveal LEOPARD's utility and adaptability in different analytical scenarios, we further explore how many training samples are required for LEOPARD to have robust view completion. The evaluation is performed on $\mathcal{D}^{\text{test}}_{v=v2,\,t=t2}$ of our three benchmark datasets by varying the number of training samples from 20 to 160 and obsNum from 0 to 50. Each condition is tested 10 times with different samples randomly selected from the training sets. The performance is evaluated by PB averaged across these repetitions. Figure 8 simplifies the boxplot and shows the median and the IQR of the averaged PB values calculated for the variables in the imputed data.

Across all datasets, average PB generally decreases with more training samples, indicating an improvement in view completion. Consistent with our previous evaluation, PB values decrease as obsNum increases. Additionally, we notice that the average PB steadily decreases for the MGH COVID proteomics dataset, which exhibits the smallest variation between the two timepoints in our UMAP plots (Fig. 4). In contrast, the average PB for the other two datasets shows some fluctuations, particularly for the KORA multi-omics dataset, which shows the most obvious variation between two timepoints. When obsNum = 0, the MGH COVID proteomics and the KORA metabolomics datasets require about 120 training samples to obtain stable results; the KORA multi-omics dataset, however, exhibits a wide range of PB under this condition. When we increase obsNum to 20, the performance stabilizes with approximately 50 to 80 samples used for training LEOPARD. Based on our evaluation, at least 80 training samples may be required for robust view completion.

**Arbitrary temporal knowledge transfer**
In computer vision, arbitrary style transfer enables the blending of any style with a content image. Here, we explore whether LEOPARD inherits this property and can complete views "arbitrarily". For this evaluation, $\mathcal{D}^{\text{test}}_{v=v2,\,t=t1}$ and $\mathcal{D}^{\text{test}}_{v=v1,\,t=t3}$ from the Extended KORA metabolomics dataset (S4 as t1, F4 as t2, and FF4 as t3) are masked separately—one to complete data at an earlier timepoint, and the other to complete the first view. For each completion, LEOPARD is trained using data from the other view but at the same timepoint ($\mathcal{D}^{\text{train}}_{v=v1,\,t=t1}$ or $\mathcal{D}^{\text{train}}_{v=v2,\,t=t3}$, respectively) along with varying obsNum and data from one or two additional timepoints (Methods).

In both tasks, some metabolites exhibit high PB values at obsNum = 0, particularly for completing $\mathcal{D}^{\text{test}}_{v=v1,\,t=t3}$ (Fig. 9) due to low metabolite concentrations. While different completions show

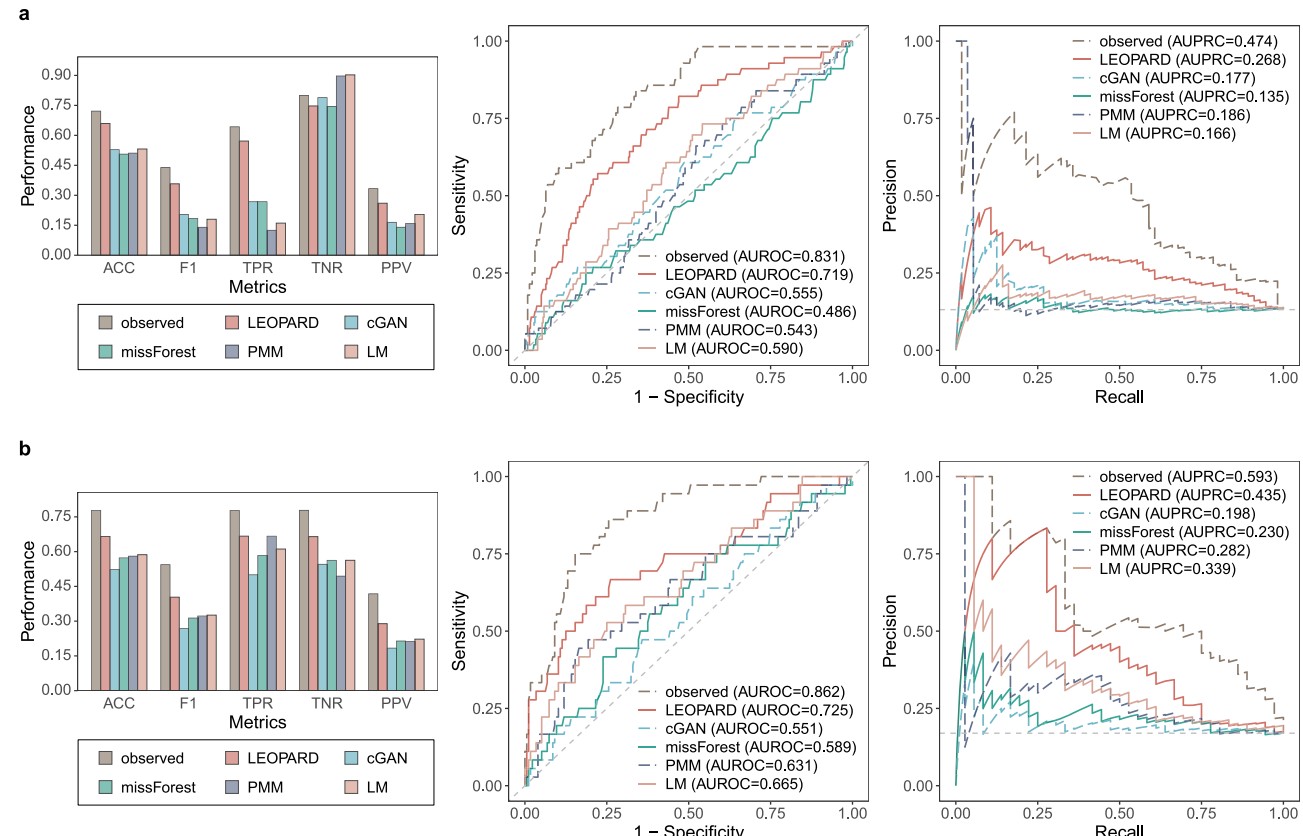

**Fig. 7 | Classification analyses with the data imputed by different methods.** Chronic kidney disease (CKD) classification evaluated using $\mathcal{D}_{v=v2,\,t=t2}^{\text{test}}$ and $\mathcal{D}_{v=v2,\,t=t2}^{\text{test}}$ (obsNum = 0) from (**a**) the KORA metabolomics dataset ($N = 416$, $N_{\text{positive}} = 56$, $N_{\text{negative}} = 360$) and (**b**) the KORA multi-omics dataset ($N = 212$, $N_{\text{positive}} = 36$, $N_{\text{negative}} = 176$). Models are trained using the balanced random forest (BRF) algorithm with identical hyperparameters and evaluated using leave-one-out-

cross-validation (LOOCV). Evaluation metrics in the bar plot include accuracy (ACC), F1 score, true positive rate (TPR, also known as sensitivity), true negative rate (TNR, also known as specificity), and positive predictive value (PPV, also known as precision). The dashed lines in the ROC and PR curves represent the performance of a hypothetical model with no predictive capability. Source data are provided as a Source Data file.

variability in their performances, PB generally decreases as obsNum increases. Our results also show that including data from additional timepoints into training can improve imputation, especially for obsNum = 0. We additionally use UMAP to visualize the content and temporal representations disentangled from the three-timepoint observed data (Supplementary Fig. 14). The UMAP illustrates LEOPARD's capability to process data spanning more than two timepoints.

Our evaluation demonstrates that LEOPARD can transfer extracted temporal knowledge to different content representations in a flexible and generalized way. Incorporating additional observations from the incomplete view or additional timepoints into training can contribute to robust results, particularly for metabolites with low concentrations.

## Discussion

In this study, we introduce LEOPARD, an architecture for missing view completion designed for multi-timepoint omics data. The performance of LEOPARD is comprehensively assessed through simulations and case studies using four real human omics datasets. Additional interesting findings emerged from our evaluation.

As illustrated in the UMAP plots, the MGH COVID proteomics data (Fig. 4 upper row) from D0 (t1, in red) and D3 (t2, in blue) show relatively low variation, while the KORA metabolomics data (Fig. 4 middle row) from F4 (t1, in red) and FF4 (t2, in blue) exhibit more substantial variations, potentially due to biological variations spanning seven years and technical variation from different analytical kits in the KORA data. In the MGH COVID dataset, we observed that LEOPARD is

outperformed by its competitors. This can be attributed to the inherent differences in the representation learning process of LEOPARD and its competitors. By directly learning mappings between views, the methods developed for cross-sectional data can exploit the input data to learn detailed, sample-specific patterns, while the representation learning in LEOPARD primarily focuses on more compact and generalized structures related to views and timepoints, potentially neglecting detailed information specific to individual samples. The MGH COVID dataset, having a high similarity between the data from D0 and D3, allows the cross-sectional imputation methods to effectively apply the mappings learned from one timepoint to another. As the data variation between two timepoints increases, LEOPARD's advantages become increasingly evident, while its competitors tend to overfit the training data and fail to generalize well to the new timepoint (Fig. 4, middle and lower rows).

For the KORA multi-omics dataset, we observed that the extremely high PB values are associated with low analyte abundances. For the same absolute error in imputed values, a variable with a low analyte abundance will have a higher absolute error ratio, leading to a higher PB than those variables with high abundances. Additionally, protein levels quantified using the Olink platform are represented as relative quantities. Data from different measurements also contain technical variations that arise from experimental factors and normalization methods used for relative quantification. When obsNum is 0, LEOPARD is trained without any data from v2 at t2, and thus cannot account for the technical variations exclusive to that part. By incorporating a few observed samples from the second timepoint into the training process, the model can better capture the data distribution and technical

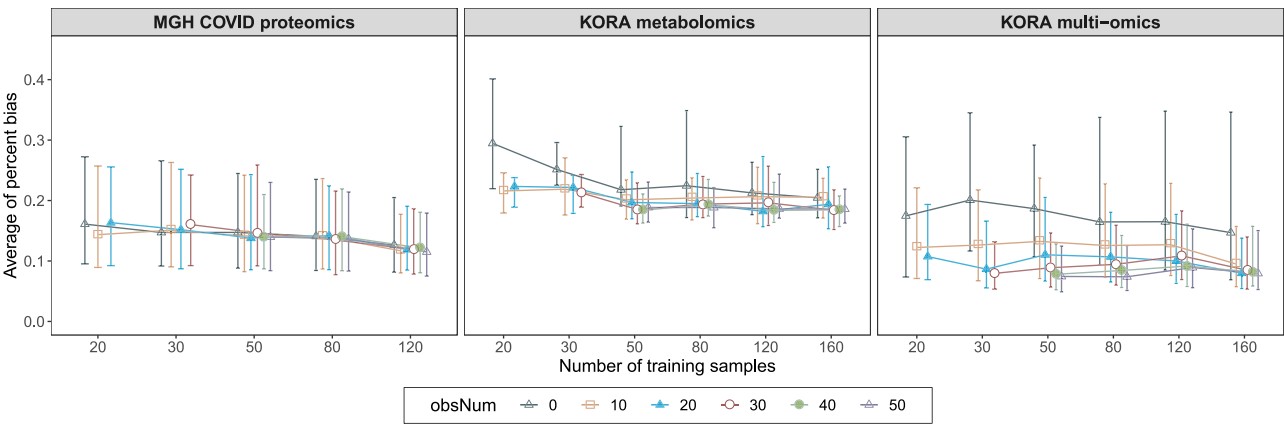

**Fig. 8 | Evaluation of minimum number of training samples required for LEO-PARD.** For each benchmark dataset, the average percent bias is evaluated on $\mathcal{D}^{\text{test}}_{v=\text{v2},\,t=\text{t2}}$ across 10 repeated completions for each combination of training sample sizes and numbers of training observations (obsNum). The bar indicates the median and the interquartile range (IQR) of the average percent bias values for different variables. In each repetition, the samples are selected randomly. Please note that the maximum obsNum cannot exceed the number of training samples, and the full training set of the MGH COVID proteomics dataset contains only 140 samples. Source data are provided as a Source Data file.

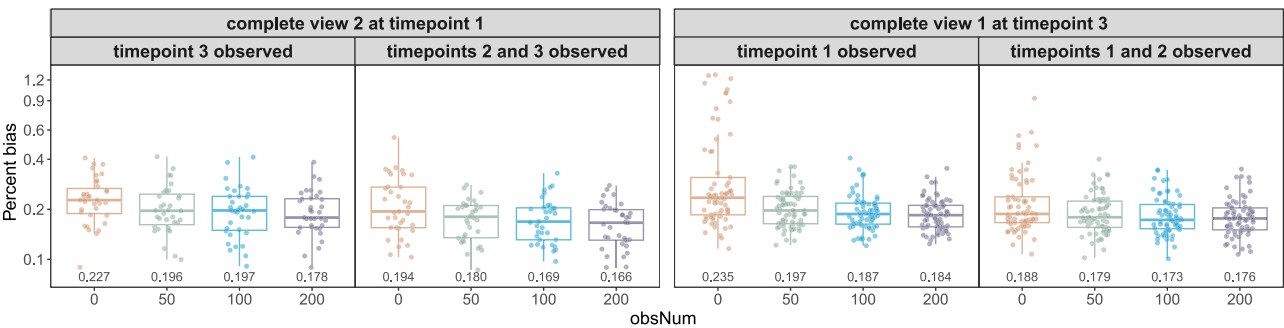

**Fig. 9 | Performance of arbitrary temporal knowledge transfer.** LEOPARD is evaluated on $\mathcal{D}^{\text{test}}_{v=\text{v2},\,t=\text{t1}}$ and $\mathcal{D}^{\text{test}}_{v=\text{v1},\,t=\text{t3}}$ from the Extended KORA metabolomics dataset. Timepoints t1, t2, and t3 correspond to the KORA S4, F4, and FF4 studies, respectively. For each completion, LEOPARD is trained with the data from the other view at the same timepoint ($\mathcal{D}^{\text{train}}_{v=\text{v1},\,t=\text{t1}}$ for $\mathcal{D}^{\text{test}}_{v=\text{v2},\,t=\text{t1}}$ and $\mathcal{D}^{\text{train}}_{v=\text{v2},\,t=\text{t3}}$ for $\mathcal{D}^{\text{test}}_{v=\text{v1},\,t=\text{t3}}$) along with varying obsNum and the data from one or two additional timepoints. For the same completion task, the evaluation shows that percent bias can be lowered by increasing obsNum or including additional timepoints into training. Each dot represents a percent bias value for a variable. Source data are provided as a Source Data file.

variation of the missing part, which contributes to a substantial reduction in high PB values.

As data imputation inevitably incurs a loss of information, we conducted case studies to assess the preservation of biological information in the imputed data. Despite all five imputation methods producing similar PB when obsNum is 0 (Fig. 3 lower row), the case studies showed that the data imputed by LEOPARD provided performance closest to the observed data, while the data imputed by cGAN, missForest, PMM, and LM showed a substantial loss of biological information (Figs. 6 and 7). This outcome highlights the importance of case studies for a reliable evaluation of imputed data.

Arbitrary style transfer, a concept from the computer vision field underpinning LEOPARD, allows the style of one image to be transferred to the content of another. This study demonstrates that LEOPARD inherits this capability and has the potential for arbitrary temporal knowledge transfer. While the minimum sample size for robust performance depends on specific data characteristics, our experiments demonstrate that LEOPARD yields robust results with approximately 60 to 80 samples on the benchmark datasets used in this study. Moreover, LEOPARD not only completes missing views for downstream analyses, but also facilitates the exploration of temporal dynamics. By analyzing the extracted temporal embeddings, LEOPARD could enable the inference of the temporal ordering of omics changes, which would be particularly valuable when there is a discrepancy between biological and chronological order. As the number

of data timepoints increases, LEOPARD is expected to offer new opportunities in predictive healthcare with multi-timepoint omics data.

While LEOPARD has been evaluated on real-world omics data with simulated missing views, it has not yet been applied to real-world studies with actual missing views. Additionally, it is important to consider the limitations and caveats of this study: (1) To align with real-world settings, we defined QC criteria based on existing studies when constructing our evaluation datasets, and consequently, only the most detectable proteins and metabolites were selected. This could inflate the metrics of both LEOPARD and other methods reported in this study. The performance on these selected variables may not accurately reflect that of the overall proteins and metabolites, especially those showing more variability in their abundance. (2) LEOPARD typically requires observed views to be complete so that temporal and content representations can be extracted. Considering the common occurrence of missing values in real-world omics data, LEOPARD is designed to tolerate a small proportion while maintaining optimal robustness. However, we observed that LEOPARD struggles to capture the full diversity present in the ground truth as maskObs increases to 20% (Supplementary Figs. 4 and 8). It is preferable for the input data for LEOPARD to contain less than 10% missing data points. Higher proportions of missing values are ideally addressed by generic imputation methods before processing with LEOPARD. Additionally, we assumed that the missing data were MCAR. Additional bias could be introduced

if data points are missing at random (MAR) or missing not at random (MNAR) in real-world scenarios. (3) Our experiments were restricted by data availability to three timepoints, but in principle LEOPARD can accommodate additional timepoints and is well-suited for cohort studies involving multiple omics with samples collected at shared discrete timepoints. As LEOPARD is not designed to learn varying temporal changes directly, it is not applicable to datasets with continuous or unaligned time intervals. (4) Due to data availability, this study does not include evaluations on large-scale transcriptomics or untargeted omics datasets with thousands of variables across multiple timepoints.

With advancements in omics measurement technology and the growing availability of longitudinal data, missing view in multi-view, multi-timepoint data is becoming a prominent issue. Our study demonstrates that established generic methods, originally developed for missing data points or cross-sectional data, do not produce robust results in this new context. This highlights the necessity for specialized methods, and our method, LEOPARD, represents an early attempt to address this issue. We anticipate further developments in imputation methods that exhibit high generalization ability, robustness to low analyte abundance, and preservation of biological variations.

## Methods

### Ethical compliance and approval
The MGH COVID study procedures were approved by the Mass General Brigham (formerly Partners) Human Research Committee, the governing institutional review board at Massachusetts General Hospital. A waiver of informed consent was approved in compliance with the Code of Federal Regulations (45CFR 46, 2018 Common Rule). Our study did not involve any direct interaction with study participants nor new data collection.

The KORA studies were all conducted in accordance with the Declaration of Helsinki. The participants provided written informed consent. The protocol of the KORA study was approved by the Ethics Committee of the Bavarian Chamber of Physicians.

### Terminology
The following outlines the terminology used throughout this study.
- Multi-view: different representations extracted from different data sources or several modalities collected in one cohort[2]. It is common for samples to be examined from multiple perspectives using different platforms or technologies, yielding different views of the data.
- Mapping: the transformation that a model learns to apply from an input space to an output space.
- Embeddings: low-dimensional representations of high-dimensional data.
- Arbitrary style transfer: a technique in computer vision field that allows the visual style from any source image to a target image, irrespective of their content differences. LEOPARD adapts this technique and has the potential to perform arbitrary temporal knowledge transfer.
- Representation disentanglement: the process of decomposing complex data into distinct underlying factors of variation, where each factor is represented independently of the others.

### Problem formulation
In our missing view completion problem, we are given a generalized dataset $\mathcal{X}_{V,T}^{N}$ that includes omics data from $N$ individuals across multiple views ($V$) and timepoints ($T$). For simplification, we consider data from two timepoints {t1, t2}, categorized into two view groups {v1, v2}. Either v1 or v2 includes complete views, with the other includes incomplete views. The dataset is split into training, validation, and test sets, denoted as $\mathcal{D}_{v,t}^{split}$. The task is to complete $\mathcal{D}_{v=v2,t=t2}^{test}$. To achieve this, we attempt to develop a model that captures the mappings

between v1 and v2, while simultaneously extracting temporal knowledge from t1 and t2 using all observed data in the training set. Further, the model should be capable of accommodating data from more than two timepoints.

### Data preprocessing
***MGH COVID dataset*** The MGH COVID study includes plasma proteins measured for patients at three timepoints: on day 0 (D0) for all patients on the day they were hospitalized for COVID, and on days 3 (D3) and 7 (D7) for patients still hospitalized then. We utilize proteomics data from D0 (t1) and D3 (t2), which have the largest sample sizes ($N = 218$, one duplicated sample removed), to construct the first mono-omics dataset.

The proteomics data from the MGH COVID study were obtained using plasma samples and the Olink® Explore 1536 platform (Olink Proteomics, Watertown, MA, USA), which consists of 1472 proteins across four Olink® Explore 384 panels: inflammation, oncology, cardiometabolic, and neurology[30]. The platform enables relative quantification of analyte concentrations in the form of $\log_2$-scaled normalized protein expression (NPX) values, where higher values correspond to higher protein levels. We selected proteins listed in the Cardiometabolic and Inflammation panels to construct the two views for the MGH COVID proteomics dataset, and the inflammation view was assumed to be incomplete. The proteins are processed based on the following quality control (QC) criteria: (1) no missing values; (2) at least 50% of measured sample values are equal to or above the limits of detection (LOD). After quality inspection (Supplementary Data 1), a total of 322 and 295 proteins from the Cardiometabolic (v1) and Inflammation (v2) panels were selected.

***KORA datasets*** Data from the KORA study are extracted from the baseline survey (S4, examined between 1999 and 2001), the first follow-up study (F4, 2006–2008), and the second follow-up study (FF4, 2013–2014)[53,54]. We use metabolomics data ($N = 2085$) from F4 (t1) and FF4 (t2) from the KORA cohort to construct the second mono-omics dataset. We additionally use metabolomics and proteomics data ($N = 1062$) from the KORA S4 (t1) and F4 (t2) to construct the multi-omics dataset.

The metabolite profiling of the KORA S4 (March–April 2011), F4 (August 2008–March 2009), and FF4 (February–October 2019) serum samples spans over a decade, during which analytical procedures have been upgraded several times. The targeted metabolomics data of F4 were measured with the analytical kit AbsoluteIDQ® p150, while S4 and FF4 data were quantified using the kit AbsoluteIDQ® p180 (Biocrates Life Sciences AG, Innsbruck, Austria). To assess the technical variation introduced by different kits, samples from 288 individuals from the F4 were remeasured (September–October 2019) using the p180 kit. Three manufacturer-provided QC samples were added to each plate to quantity the plate effect. Additionally, five external QC samples were added to each plate when measuring using the p180 kit. The p150 and p180 kits allow simultaneous quantification of 163 and 188 metabolites, respectively[55]. Only metabolites meeting the following QC criteria[56,57] were selected: (1) overlap between p150 and p180; (2) no missing values; (3) at least 50% of measured sample values are equal to or above the LOD of corresponding plates; (4) median relative standard deviation (RSD) of QC samples <25%; (5) Spearman correlation coefficients between the KORA F4 (remeasured, p180) and F4 (original, p150) > 0.5. After QC procedures, the metabolites values were further normalized using TIGER (Technical variation elImination with ensemble learninG architEctuRe)[55] with its default setting to remove the plate effects. For the multi-omics dataset, TIGER was also used to remove the technical variation introduced by different kits following our previous protocol[58].

The proteomics data from the KORA cohort are available at two timepoints, S4 and F4, and are measured using plasma (S4, February 2020) and serum (F4, December 2016–January 2017) samples with the

Olink® Target 96 Inflammation panel (Olink Proteomics, Uppsala, Sweden)[50]. The panel includes 92 proteins, and only proteins pass the following QC criteria were selected: (1) no missing values; (2) at least 75% of measured sample values are equal to or above the LOD. TIGER was then used to remove the technical variation introduced by different kits following our previous protocol[55,58].

For the KORA metabolomics dataset, 106 targeted metabolites satisfy all criteria (Supplementary Data 2) and are categorized into five analyte classes: acylcarnitine (AC), amino acid (AA), glycerophospholipid (GPL), sphingolipid (SL), and monosaccharide (MS). Two view groups are constructed by 70 metabolites from GPL (v1) and 36 metabolites from the other four classes (v2). For the KORA multi-omics dataset, 104 metabolites (v1) and 66 proteins (v2) satisfy all QC criteria (Supplementary Data 2 and 3) and are selected to construct two views.

To evaluate LEOPARD's capability for arbitrary temporal knowledge transfer, we further constructed the Extended KORA metabolomics dataset to include data from the baseline study (S4, as t1) and the second follow-up study (FF4, as t3), spanning approximately 14 years. We divided the metabolites data into two views using the same strategy as we used for the original KORA metabolomics dataset. Due to different QC results across the two analytical kits, two metabolites, specifically PC aa C38:1 in v1 and C16:2 in v2, were excluded (see Supplementary Data 2). This Extended KORA metabolomics dataset comprised 104 metabolites (69 in v1 and 35 in v2) with 614 individuals who have data at all timepoints. These samples were divided into training, validation, and test sets with a ratio of 64%, 16%, and 20%, respectively, corresponding to 393, 98, and 123 samples. The data in $\mathcal{D}^{\text{test}}_{v=v2,\,t=t1}$ and $\mathcal{D}^{\text{test}}_{v=v1,\,t=t3}$ are masked for performance evaluation.

## Training the cGAN model

***Architecture implementation*** The cGAN[59] extends the original GAN[60] model by introducing additional information into the generation process, thereby providing more control over the generated output. In our context, the completion of missing views is conditioned on the data from observed views. Moreover, we enhanced the baseline cGAN model with an auxiliary classifier[37] to ensure that the imputed view can be paired with the corresponding observed view.

Specifically, the generator $G_{\text{cGAN}}$ learns the complex mappings between $\mathcal{D}^{\text{train}}_{v=v1,\,t}$ and $\mathcal{D}^{\text{train}}_{v=v2,\,t}$, with $t = t1$ in our case. The reconstruction loss $\mathcal{L}_{\text{rec\_cGAN}}$ quantifies the differences between the actual and reconstructed data. The discriminator $D_{\text{cGAN}}$ computes the adversarial loss $\mathcal{L}_{\text{adv\_cGAN}}$ by distinguishing if data are real ($\mathcal{D}^{\text{train}}_{v=v2,\,t=t1}$) or generated ($\hat{\mathcal{D}}^{\text{train}}_{v=v2,\,t=t1}$). $D_{\text{cGAN}}$ also has an auxiliary classifier that computes the auxiliary loss $\mathcal{L}_{\text{aux\_cGAN}}$ by predicting whether the pairs of views are real, i.e., $\langle \mathcal{D}^{\text{train}}_{v=v1,\,t=t1}, \mathcal{D}^{\text{train}}_{v=v2,\,t=t1} \rangle$, or fake, i.e., $\langle \mathcal{D}^{\text{train}}_{v=v1,\,t=t1}, \hat{\mathcal{D}}^{\text{train}}_{v=v2,\,t=t1} \rangle$. The final loss is defined as:

$$\mathcal{L}_{\text{cGAN}} = w_{\text{rec\_cGAN}} \times \mathcal{L}_{\text{rec\_cGAN}} + w_{\text{adv\_cGAN}} \times \mathcal{L}_{\text{adv\_cGAN}} + w_{\text{aux\_cGAN}} \times \mathcal{L}_{\text{aux\_cGAN}} \tag{1}$$

where $w_{\text{rec\_cGAN}}$, $w_{\text{adv\_cGAN}}$, and $w_{\text{aux\_cGAN}}$ are weights for the corresponding losses. After training, the generator $G_{\text{cGAN}}$ is applied to $\mathcal{D}^{\text{test}}_{v=v1,\,t=t2}$ to generate $\hat{\mathcal{D}}^{\text{test}}_{v=v2,\,t=t2}$.

The generator of the cGAN model consists of several residual blocks[61] and uses the parametric rectified linear unit (PReLU)[62] as its activation function. Each residual block includes batch normalization[63] as necessary. MSE loss serves as $\mathcal{L}_{\text{rec\_cGAN}}$. Both the MSE loss and the binary cross-entropy (BCE) loss are considered for $\mathcal{L}_{\text{adv\_cGAN}}$ and $\mathcal{L}_{\text{aux\_cGAN}}$, determined by the hyperparameter tuning experiments. MSE and BCE losses are defined as:

$$\mathcal{L}_{\text{MSE}} = \frac{1}{N} \left\| x^i - \hat{x}^i \right\|_2^2 \tag{2}$$

$$\mathcal{L}_{\text{BCE}} = -\frac{1}{N} \sum_{i=1}^{N} \left[ y^i \log(\hat{y}^i) + (1 - y^i) \log(1 - \hat{y}^i) \right] \tag{3}$$

where $N$ is the number of samples, $x^i$ and $\hat{x}^i$ represents the true and estimated values for the $i$-th sample, while $y^i$ and $\hat{y}^i$ represents the true and predicted labels for the $i$-the sample.

The cGAN model is trained with the Adam optimizer[64], with a mini-batch size of 16 for the MGH COVID dataset and 32 for the two KORA-derived datasets. The model is implemented using PyTorch[65] (v1.11.0), PyTorch Lightning[66] (v1.6.4), and tensorboard[67] (v2.10.0), and run on a graphics processing unit (GPU) operating the compute unified device architecture (CUDA, v11.3.1).

***Hyperparameter optimization*** We utilized the training (64%) and validation (16%) sets from the KORA metabolomics dataset, which has the largest sample size among our evaluation datasets, to optimize the hyperparameters. We employed a grid search over various combinations of hidden layer size, numbers of residual blocks, batch normalization, and weights for different losses. We determined the number of training epochs based on early stopping triggered by the MSE reconstruction accuracy calculated on the observed data from the validation sets. Our experiments included variations in the number of hidden neurons for both the generator and discriminator, with options including 32, 64, 128, and 256. The number of residual blocks spanned from 2 to 6.

The final hyperparameters comprised five residual blocks of 64 neurons each for the generator and three hidden layers of 128 neurons each for the discriminator. Batch normalization was incorporated into the first four residual blocks of the generator and the last two layers of the discriminator to stabilize the learning process and accelerate convergence. A weight of 0.5 to $\mathcal{L}_{\text{rec\_cGAN}}$ and a weight of 0.25 to $\mathcal{L}_{\text{adv\_cGAN}}$ and $\mathcal{L}_{\text{aux\_cGAN}}$ were determined. The MSE loss was selected for both $\mathcal{L}_{\text{adv\_cGAN}}$ and $\mathcal{L}_{\text{aux\_cGAN}}$. The determined hyperparameters were fixed and used in all evaluations.

## Training the LEOPARD

***Architecture implementation*** View-specific pre-layers $E^v_{\text{pre}}$ are used to embed input data $x^i_{v,t} \in \mathcal{D}^{\text{train}}_{v,t}$ of different views into dimensionally uniform embeddings $z\_\text{pre}^i_{v,t}$. Here, $i$ represents the data from the $i$-th individual. The representation disentangler of LEOPARD comprises a content encoder $E_c$ and a temporal encoder $E_t$, both shared by input data across different views and timepoints. This module learns a timepoint-invariant content representation $z\_\text{content}^i_{v,t}$ and temporal feature $z\_\text{temporal}^i_{v,t}$ from $z\_\text{pre}^i_{v,t}$. Following the encoding process, the generator $G$ employs the AdaIN technique to re-entangle content representation and temporal knowledge:

$$\text{AdaIN}\left(z\_\text{content}^i, z\_\text{temporal}^i\right) = \sigma\left(z\_\text{temporal}^i\right) \\ \times \frac{z\_\text{content}^i - \mu(z\_\text{content}^i)}{\sigma(z\_\text{content}^i)} + \mu\left(z\_\text{temporal}^i\right) \tag{4}$$

where $\mu$ and $\sigma$ denote the mean and standard deviation operations, respectively. View-specific post-layers $E^v_{\text{post}}$ convert the re-entangled embeddings back to omics data $\hat{x}^i_{v,t}$. The discriminator $D$ is trained to classify whether an input is a real sample or a generated output coming from $G$ and $E^v_{\text{post}}$. $D$ produces the same number of outputs as the source classes of the observed data, each corresponding to one view at one timepoint. For a sample belonging to the source class $c_{v,t}$, we penalize $D$ during the update cycle of $D$ if its output incorrectly classifies a real data instance as false or a generated data instance as true for $c_{v,t}$; when updating $G$, we only penalize $G$ if $D$ correctly identifies the generated data instance as false for $c_{v,t}$.

In our study, we defined the contrastive loss $\mathcal{L}_{\text{con}}$ as the mean of the NT-Xent losses calculated separately for content and temporal

representations. The NT-Xent loss is formulated as follows:

$$\mathcal{L}_{\text{NT-Xent}}(z^i, z^j) = -\log \frac{\exp\left(\text{sim}(z^i, z^j)/\tau\right)}{\sum_{k=1}^{2N} \mathbb{1}_{[k \neq i]} \exp\left(\text{sim}(z^i, z^k)/\tau\right)} \quad (5)$$

where $z^i$ and $z^j$ are embeddings of a positive pair $(i, j)$, $\tau$ is a temperature factor that scales the similarities, and $\mathbb{1}_{[k \neq i]}$ is an indicator function that equals 1 when $k \neq i$ and 0 otherwise. And $\text{sim}(\cdot)$ denotes cosine similarity, defined as:

$$\text{sim}(a, b) = \frac{a \cdot b}{||a||_2 ||b||_2} \quad (6)$$

In each training iteration, we generate data for the missing view before calculating the loss. This allows the generated data to be factorized into corresponding content and temporal representations, further facilitating loss minimization. For the $\mathcal{L}_{\text{NT-Xent}}$ calculated on content representations, positive pairs are defined as $\langle z\_\text{content}^i_{v=v1, t=t1}, z\_\text{content}^i_{v=v1, t=t2}\rangle$ and $\langle z\_\text{content}^i_{v=v2, t=t1}, z\_\widehat{\text{content}}^i_{v=v2, t=t2}\rangle$, which are the same kind of content embeddings from the same individuals across different timepoints. Similarly, the positive pairs of temporal representations are $\langle z\_\text{temporal}^i_{v=v1, t=t1}, z\_\text{temporal}^i_{v=v2, t=t1}\rangle$ and $\langle z\_\text{temporal}^i_{v=v1, t=t2}, z\_\widehat{\text{temporal}}^i_{v=v2, t=t2}\rangle$. The representation loss $\mathcal{L}_{\text{rep}}$ is the mean of the MSE losses calculated for content and temporal representations. For each type of representation, LEOPARD measures the MSE between the representation factorized from the actual data and reconstructed data. The reconstruction loss $\mathcal{L}_{\text{rec}}$ quantifies the discrepancies between the actual and reconstructed data. Any missing values in the observed view are encoded as the mean values across each specific variable, and these mean-encoded values are excluded from the computation of $\mathcal{L}_{\text{rec}}$ during back-propagation. This strategy enhances the robustness of LEOPARD in scenarios where input data contain missing values. The generator can arbitrarily produce data for any source classes given the content and temporal representations. To ensure the representation disentangler can capture the highly structured data pattern, we only compute the $\mathcal{L}_{\text{rec}}$ on the data generated from content and temporal representations derived from different source classes. For instance, $\langle z\_\text{content}^i_{v=v1, t=t1}, z\_\text{temporal}^i_{v=v2, t=t1}\rangle$ or $\langle z\_\widehat{\text{content}}^i_{v=v2, t=t2}, z\_\text{temporal}^i_{v=v1, t=t1}\rangle$. Data generated from representation pairs of the same views and timepoints, such as $\langle z\_\text{content}^i_{v=v1, t=t2}, z\_\text{temporal}^i_{v=v1, t=t2}\rangle$ are not used for optimization. This design imposes additional restraints, and LEOPARD is tamed to learn more generalized representations, which helps prevent overfitting. Similar to the cGAN model described in the previous section, the adversarial loss $\mathcal{L}_{adv}$ is also computed based on the MSE. The final loss of LEOPARD is defined as:

$$\mathcal{L}_{\text{LEOPARD}} = w_{\text{con}} \times \mathcal{L}_{\text{con}} + w_{\text{rep}} \times \mathcal{L}_{\text{rep}} + w_{\text{rec}} \times \mathcal{L}_{\text{rec}} + w_{\text{adv}} \times \mathcal{L}_{\text{adv}} \quad (7)$$

where $w_{\text{con}}$, $w_{\text{rep}}$, $w_{\text{rec}}$, and $w_{\text{adv}}$ are the weights of the losses.

The encoders, generator, and discriminator of LEOPARD are built from blocks of layers without skip connections. Each block starts with a dense layer. An instance normalization layer[68] is added after the dense layer for the content encoder. The encoders and generator use the PReLU as their activation functions, while the discriminator uses the sigmoid function. A dropout layer[69] is incorporated after the activation layer, where necessary.

The LEOPARD model is trained with the Adam optimizer, with a mini-batch size of 64. The model is implemented under the same computational environment as the cGAN model.

***Ablation test*** We conducted a comprehensive ablation study to assess the individual contributions of the four distinct losses incorporated into our LEOPARD architecture. By excluding each loss, we benchmarked the performance against a baseline setting that only

utilizes reconstruction loss. The ablation test was performed with the training and validation sets from the KORA metabolomics dataset. Grid search was used to determine the optimal weights for the losses, and the median of PB computed from the validation set was used to quantify the performance. The network layer numbers and sizes were consistent during the evaluation. We used three hidden layers for the generator and encoders, with each layer containing 64 neurons. The weight for reconstruction loss was fixed at 1, while the weights for the other three losses varied across 0.01, 0.05, 0.1, 0.5, and 1. The number of training epochs was determined by the model's saturation point in learning, which is when the median of PB computed on the validation set ceased to decrease significantly. Our experiments show that all four losses contribute to lowering PB in the imputed data. The optimal weights include $w_{\text{rec}} = 1, w_{\text{con}} = 0.1, w_{\text{rep}} = 0.1$, and $w_{\text{adv}} = 1$. The performance of each loss combination at their optimal weights is summarized in Supplementary Fig. 15.

***Further hyperparameter optimization*** The training and validation sets from the KORA metabolomics dataset were further used to optimize LEOPARD's hyperparameters, aiming to effectively capture data structure for existing data reconstruction and achieve robust generalization for missing data imputation. The weights for different losses have been determined in the ablation test. We then conducted a grid search across various combinations of hidden layer size, hidden layer number, dropout rate, projection head, and temperature for contrastive loss.

The number of hidden neurons within the encoders, generator, and discriminator varied across 32, 64, 128, and 256, with the number of layers ranging from 2 to 4, and dropout rates of 0%, 30%, and 50%. Our findings show that higher numbers of hidden neurons and layers tended to yield worse performance in terms of median PB (Supplementary Fig. 16). LEOPARD was configured with three 64-neuron layers incorporated into both content and temporal encoders and the generator. The discriminator included two hidden layers, each having 128 neurons. Dropout was not used.

The projection head and temperature are two important hyperparameters that control the performance of contrastive learning. The projection head is a compact network consisting of one full connected hidden layer with the same layer size as the input dimension, a rectified linear unit (ReLU)[70] and one output layer. The temperature is a scalar that scales the similarities before the softmax operation. Some previous experiments performed on image datasets emphasized the importance of the projection head and reported different output sizes yielded similar results[38]. We evaluated the performance of LEOPARD both without a projection head and with a projection head of the output size varying across 16, 32, 64, 128, 256, and 512. The temperature is fined-tuned across 0.05, 0.1, 0.5, 1, 5, 10, and 30. Based on our experiments, LEOPARD is trained with a temperature of 0.05 and without using a projection head (Supplementary Fig. 17).

The determined hyperparameters, including loss weights, remained unchanged in all our performance evaluations.

## Representation disentanglement

The disentanglement of content and temporal representations was evaluated using the KORA multi-omics dataset. LEOPARD was trained for 600 epochs, for each of which the disentanglement progress was visualized with the following steps: First, content and temporal representations were factorized from the metabolomics (S4 and F4) and proteomics data (S4). Then the generator imputed the proteomics data (F4) by incorporating the temporal information from the metabolomics data (F4) into the content representation from the proteomics data (S4). The generated proteomic data (F4) were then fed to the content and temporal encoders to extract the corresponding representations. Subsequently, these content or temporal representations of both the observed and imputed data were standardized to ensure all latent variables had a mean of zero and a standard

deviation of one. Afterward, two separate UMAP models were built using the R package umap[71], each fitted to the content and temporal representations, with a configuration of *n_neighbors = 15* and *min_dist = 0.1*. Lastly, scatter plots were generated using the R packages ggplot2[72] and ggsci[73]. Each point in the plot represents an individual sample, and the color indicates the data sources. The visualization epochs were selected experimentally to illustrate the progress of representation disentanglement during the training process.

## Performance evaluation

The LEOPARD and cGAN models were trained using the hyperparameters previously described. For missForest, the imputation was performed using a 100-tree random forest[74] model, with the maximum number of iterations (maxit) set to 10. Multiple imputations by PMM, LM, and GLMM were performed using the R packages mice[44] and micemd[75]. Each method's imputations were performed five times ($m = 5$) with a maxit value set to five. The PMM model was built using the argument *method = "pmm"*. When obsNum = 0, data of v2 at t2 are assumed to be completely missing. In this scenario, the LM method was trained using *method = "norm"*. When obsNum is a non-zero value, the GLMM model was built using *method = "2l.glm.norm"*.

All methods used only the data in the training sets to build imputation models. Their performance was evaluated on $D^{\text{test}}_{v=v2, t=t2}$. Different imputation methods may require specific data structures for input: cGAN and LEOPARD first build imputation models using training data, then apply the built models to test set to complete missing views. In contrast, the input data for other methods can be an incomplete matrix with missing values coded as NA. We adapted the input data accordingly to accommodate these specific requirements:

- Method cGAN only learns from samples where both views are present. Therefore, its training data only included training data from the first timepoint ($D^{\text{train}}_{v=v1, t=t1}$ and $D^{\text{train}}_{v=v2, t=t1}$) and data of different obsNum from the second timepoint ($D^{\text{train}}_{v=v1, t=t2}$ and $D^{\text{train}}_{v=v2, t=t2}$).
- LEOPARD can additionally learn from data where only one view is available. In addition to $D^{\text{train}}_{v=v1, t=t1}$ and $D^{\text{train}}_{v=v2, t=t1}$, the entire $\mathcal{D}^{\text{train}}_{v=v1, t=t2}$ was included in its training. The variation of obsNum only affected the number of observed samples from $\mathcal{D}^{\text{train}}_{v=v2, t=t2}$.
- The input data for missForest combined training data (including $D^{\text{train}}_{v=v1, t=t1}$, $D^{\text{train}}_{v=v2, t=t1}$, $D^{\text{train}}_{v=v1, t=t2}$, and data of different obsNum from $D^{\text{train}}_{v=v2, t=t2}$) and test data ($D^{\text{test}}_{v=v1, t=t2}$), with NA filling the masked data in the matrix.
- For the multiple imputation methods in MICE family, the input data were constructed with training data (identical to that used for missForest) and test data (including $D^{\text{test}}_{v=v1, t=t1}$, $D^{\text{test}}_{v=v2, t=t1}$ and $D^{\text{test}}_{v=v1, t=t2}$). To ensure test data remained unused for model training, a logical vector with TRUE assigned to test samples was passed to the argument *ignore*. Masked values were filled with NA in the matrix.
- When building the GLMM model, the input data additionally contained sample IDs and timepoint labels. A constant residual error variance is assumed for all individuals. Building a GLMM model for a large dataset is extremely time-consuming; thus, for the MGH COVID dataset, we selected the top 100 highly Spearman-correlated proteins for each protein requiring imputation. The selected proteins were incorporated into the imputation process by passing to the argument *predictorMatrix*.

PB was selected as a performance metric as it quantifies the relative deviation of imputed values from actual observations, offering a more straightforward interpretation compared to metrics like RMSE and MAE. PB was calculated for each variable using the formula:

$$\text{PB}_i = \frac{1}{m} \sum_{\text{imp}=1}^{m} \text{median}\left( \frac{\left| \hat{x}^i_{(\text{imp})} - x^i \right|}{x^i} \right) \quad (8)$$

where $\hat{x}^i_{(\text{imp})}$ is the imputed value for the $i$-th variable from the imp-th imputation, while $m$ is the number of imputations. For single imputation methods, LEOPARD, cGAN, and missForest, $m = 1$. PB results for each imputation method were visualized using dot and box plots, with each dot representing a variable in the specific dataset. Bonferroni correction was used for multiple testing adjustment. The exact *P*-values of the comparison are provided in Supplementary Table 3.

For the evaluation using UMAP, we first fitted a UMAP model using the data of $D^{\text{train}}_{v=v2, t=t1}$ and $D^{\text{train}}_{v=v2, t=t2}$. We then used the fitted model to embed the data of $D^{\text{test}}_{v=v2, t=t1}$, $D^{\text{test}}_{v=v2, t=t2}$, and $\hat{D}^{\text{test}}_{v=v2, t=t2}$, where $\hat{D}^{\text{test}}_{v=v2, t=t2}$ represents the imputed data for $D^{\text{test}}_{v=v2, t=t2}$ produced by different imputation methods. Training UMAP models only with training data can improve the model's generalization and make it serve as a fixed reference. As observed data and different imputed data from test sets are transformed into embeddings in the same way, the structural similarities or discrepancies of their embeddings are more directly related to the data, rather than to variations of different dimension reduction processes. This approach reduces the possibility of obtaining similar-looking embedding plots from dissimilar datasets and guarantees robust evaluations.

For PMM, LM, and GLMM, $\hat{D}^{\text{test}}_{v=v2, t=t2}$ is the average of all estimates from their multiple imputations. An imputation method is considered effective if the distribution of $\hat{D}^{\text{test}}_{v=v2, t=t2}$ embeddings is highly similar to that of the $D^{\text{test}}_{v=v2, t=t2}$ embeddings. The UMAP models were fitted with the identical configurations described in the prior section.

## Regression analyses

We used observed and imputed test set data from the KORA metabolomics and multi-omics datasets for two regression analyses. We employed multivariate linear regression models for each of the observed or imputed data. The imputed data of LEOPARD, cGAN, missForest, PMM, and LM were obtained under the setting of obsNum = 0.

For the KORA metabolomics dataset, we used the concentration of each metabolite as the response variable and age as the predictor variable, while controlling for sex, to detect age-associated metabolites. For the KORA multi-omics dataset, we used NPX values of each protein as the response and eGFR as the predictor, controlling for sex and age, to detect eGFR-associated proteins. In both analyses, we applied a Bonferroni correction to adjust the *P*-value significance threshold to mitigate the risk of false positives in multiple testing.

## Classification analyses

We also used observed and imputed test set data from the KORA metabolomics and multi-omics datasets for CKD prediction. CKD cases were determined based on their eGFR values, which were computed from serum creatinine, sex, race, and age, using the Chronic Kidney Disease Epidemiology Collaboration (CKD-EPI) equation[76]. The data imputed by LEOPARD, cGAN, missForest, PMM, and LM were obtained under the setting of obsNum = 0.

For each type of raw or imputed data, we trained a BRF model using the Python library imbalanced-learn[77]. This model was specifically selected to address the dataset imbalance and reduce the risk of overfitting to the majority class. All models were trained with default hyperparameters (*criterion = "gini"*, *min_samples_split = 2*, *min_samples_leaf = 1*, *max_features = "sqrt"*, *bootstrap = True*), except for *n_estimators = 1000* and *class_weight = "balanced_subsample"*. Due to the limited sample size, we validated the performance using the LOOCV strategy, allowing maximal use of data for both training and validation. Performance metrics were calculated using the R package caret[78]. These metrics provided a comprehensive understanding of the predictive power of the observed and imputed data.

The ROC curves were plotted to illustrate the trade-off between sensitivity and 1-specificity at varying decision thresholds. Considering

the imbalance in our dataset, and with our primary interest in the positive class, which is also the minority, we further plotted PR curves to depict the trade-off between precision and recall at different thresholds for the classifiers trained with the different data. For PR curves, the baseline performance of a non-discriminative model was determined by the proportion of positive cases ($56/416 = 0.135$ for the KORA metabolomics dataset and $36/212 = 0.170$ for the KORA multi-omics dataset). Both the ROC and PR curves were plotted using the R package precrec[79].

### Case studies on the MGH COVID proteomics dataset

The neutralization level is a binary response indicating "LOW" or "HIGH" for each sample. The imputed proteomics data of LEOPARD, cGAN, missForest, PMM, and LM were obtained under the setting of obsNum = 0. Bonferroni correction is used to adjust the $P$-value significance threshold for logistic regression.

For the task of predicting neutralization level, the original study used NPX values from D0 to predict neutralization levels at D3. However, as our imputation evaluation involves only D3 data, we instead used D3 proteomics (from both observed and imputed data) to predict neutralization levels on the same day. This task is theoretically easier than the one in the original study, as both predictors and outcomes are from the same timepoint. The BRF models were trained using the same hyperparameters as in the previous experiments. For PR curves, the baseline performance of a non-discriminative model was determined by the proportion of positive cases ($30/43 = 0.698$).

### Arbitrary temporal knowledge transfer

In the previous evaluation, we assessed the performance of each method on $\mathcal{D}^{test}_{v=v2,\,t=t2}$ from the benchmark datasets. We then extended our analysis by evaluating LEOPARD's performance on individually masked test sets $\mathcal{D}^{test}_{v=v2,\,t=t1}$ and $\mathcal{D}^{test}_{v=v1,\,t=t3}$ from the Extended KORA metabolomics dataset with varying obsNum. This approach allows us to assess LEOPARD's capability to arbitrarily complete any views at any timepoints within this dataset. LEOPARD was trained using the same hyperparameters as we used in the previous experiments.

When completing $\mathcal{D}^{test}_{v=v2,\,t=t1}$, the training data initially include $\mathcal{D}^{train}_{v=v1,\,t=t1}$ and varying obsNum from $\mathcal{D}^{train}_{v=v2,\,t=t1}$. If timepoint 3 is observed, the training data also include $\mathcal{D}^{train}_{v=v1,\,t=t3}$ and $\mathcal{D}^{train}_{v=v2,\,t=t3}$, with $\mathcal{D}^{train}_{v=v1,\,t=t2}$ and $\mathcal{D}^{train}_{v=v2,\,t=t2}$ added if timepoint 2 is observed as well.

Similarly, when completing $\mathcal{D}^{test}_{v=v1,\,t=t3}$, the training data initially include $\mathcal{D}^{train}_{v=v2,\,t=t3}$ and varying obsNum from $\mathcal{D}^{train}_{v=v1,\,t=t3}$. If timepoint 1 is observed, $\mathcal{D}^{train}_{v=v1,\,t=t1}$ and $\mathcal{D}^{train}_{v=v2,\,t=t1}$ are added. As with the previous case, $\mathcal{D}^{train}_{v=v1,\,t=t2}$ and $\mathcal{D}^{train}_{v=v2,\,t=t2}$ are further included if timepoint 2 is also observed. For this three-timepoint scenario, we used UAMP to visualize the representations disentangled from the observed and imputed data during training. The UMAP model was configured with the same parameters previously employed to examine representation disentanglement using the KORA multi-omics dataset (see Representation disentanglement).

### Reporting summary

Further information on research design is available in the Nature Portfolio Reporting Summary linked to this article.

## Data availability

The proteomics data, published by the original authors, are freely available for investigators from Mendeley Data (https://doi.org/10.17632/nf853r8xsj). To protect the identity of individual subjects, public posting of patient-level demographic information is limited as required by the Mass General Brigham Human Research Committee. The MGH COVID proteomics dataset constructed in this study is available at our GitHub repository (https://github.com/HAN-Siyu/LEOPARD). The KORA data are governed by the General Data Protection Regulation (GDPR) and national data protection laws, with additional restrictions imposed by the Ethics Committee of the Bavarian Chamber of Physicians to ensure data privacy of the study participants. As a result, the data cannot be made freely available in a public repository. However, researchers with a legitimate interest in accessing the data may submit a request through an individual project agreement with KORA via the online portal (https://www.helmholtz-munich.de/en/epi/cohort/kora). Upon receipt of the request, the data access committee will review the application and, subject to approval, provide the researcher with a data usage agreement. The expected timeframe for processing requests and the duration of data access vary depending on the project and are determined by the data access committee. Researchers will receive this information upon submission of their request. Source data are provided with this paper.

## Code availability

The source code and implementation details of LEOPARD can be freely accessed on our GitHub repository (https://github.com/HAN-Siyu/LEOPARD) and are also available via Zenodo (https://zenodo.org/records/14927694)[80]. Detailed documentation and examples can be found in the Manual, which is also available at this repository.

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

## Acknowledgements

This project has received funding from the Innovative Medicines Initiative 2 Joint Undertaking (JU) under grant agreement No 821508 (CARDIATEAM). The JU receives support from the European Union's Horizon 2020 research and innovation programme and the European Federation of Pharmaceutical Industries and Associations (EFPIA). The German Diabetes Center is supported by the German Federal Ministry of Health (Berlin, Germany) and the Ministry of Science and Culture in North-Rhine Westphalia (Düsseldorf, Germany). This study was supported in part by a grant from the German Federal Ministry of Education and Research to the German Center for Diabetes Research (DZD). The KORA study was initiated and financed by the Helmholtz Zentrum München – German Research Center for Environmental Health, which is funded by the German Federal Ministry of Education and Research (BMBF) and by the State of Bavaria. Data collection in the KORA study is done in cooperation with the University Hospital of Augsburg. K.S. is supported by the Biomedical Research Program at Weill Cornell Medicine in Qatar, a program funded by the Qatar Foundation. K.S. is also supported by Qatar National Research Fund (QNRF) grants NPRP11C-0115-180010 and ARG01-0420-230007. We express our appreciation to all KORA study participants for their blood donation and time. We thank all participants for their long-term commitment to the KORA study, the staff for data collection and research data management and the members of the KORA Study Group (https://www.helmholtz-munich.de/en/epi/cohort/kora) who are responsible for the design and conduct of the KORA study. We also extend our gratitude to all participants of the MGH COVID study, as well as the dedicated staff who have participated in the study design, data collection, and project management. We also appreciate their efforts in making the data freely available. The authors are grateful to Prof. Dr. Barbara Thorand (Institute of Epidemiology, Helmholtz Zentrum München) for her contribution to the collection of KORA proteomics data and for her constructive suggestions to improve this study. The authors thank Yun-Hsiu Tai, Ming Cheng, Ruoyu Wang, Yuan Guo, Linrui Fan for their supports and suggestions during the development of LEOPARD.

## Author contributions

S.H. designed and carried out the analyses, implemented the algorithms, interpreted the result, and wrote the manuscript. S.Y. and Y.L. assisted in the development of the method. M.S., M.H., J.G, F.S., G.M. assisted in interpreting the results. J.L., C.P., A.P., J.A., K.S., C.G., S.M.H., C.H., M.R., A.P., and R.W-S. performed data acquisition and preparation of the KORA cohort data. N.C. and F.P.C. assisted in the analysis and revised the manuscript. R.W-S. supervised the analyses, interpreted the results, and revised the manuscript. All authors reviewed the final manuscript.

## Funding

## Competing interests

The authors declare no competing interests.

## Additional information

[1]Institute of Translational Genomics, Helmholtz Zentrum München, German Research Center for Environmental Health, Neuherberg, Germany. [2]TUM School of Medicine and Health, Technical University of Munich, Munich, Germany. [3]German Center for Diabetes Research (DZD), Partner Neuherberg, Neuherberg, Germany. [4]Institute of Epidemiology, Helmholtz Zentrum München, German Research Center for Environmental Health, Neuherberg, Germany. [5]Institute for Medical Information Processing, Biometry, and Epidemiology (IBE), Faculty of Medicine, Ludwig-Maximilians-Universität München, Pettenkofer School of Public Health, Munich, Germany. [6]Metabolomics and Proteomics Core, Helmholtz Zentrum München, German Research Center for Environmental Health, Neuherberg, Germany. [7]College of Computer Science and Technology, Key Laboratory of Symbol Computation and Knowledge Engineering of Ministry of Education, Jilin University, Changchun, China. [8]Eli Lilly and Company, Lilly Corporate Center, Indianapolis, IN, USA. [9]Whitaker Cardiovascular Institute, Boston University Chobanian & Avedisian School of Medicine, Boston, MA, USA. [10]Genomics Variation, Population Medicine and Complex Diseases Unit, Turin University, Turin, Italy. [11]Institute of Experimental Genetics, Helmholtz Zentrum München, German Research Center for Environmental Health, Neuherberg, Germany. [12]Department of Biochemistry, Yong Loo Lin School of Medicine, National University of Singapore, Singapore, Singapore. [13]Institute of Biochemistry, Faculty of Medicine, University of Ljubljana, Ljubljana, Slovenia. [14]Bioinformatics Core, Weill Cornell Medicine-Qatar, Education City, Doha, Qatar. [15]Englander Institute for Precision Medicine, Weill Cornell Medicine, New York, NY, USA. [16]Research Unit of Molecular Epidemiology, Helmholtz Zentrum München, German Research Center for Environmental Health, Neuherberg, Germany. [17]Institute for Clinical Diabetology, German Diabetes Center, Leibniz Center for Diabetes Research at Heinrich-Heine-University Düsseldorf, Düsseldorf, Germany. [18]German Center for Diabetes Research (DZD), Partner Düsseldorf, Neuherberg, Germany. [19]Department of Endocrinology and Diabetology, Medical Faculty and University Hospital Düsseldorf, Heinrich-Heine-University Düsseldorf, Düsseldorf, Germany. [20]Institute of AI for Health, Helmholtz Zentrum München, German Research Center for Environmental Health, Neuherberg, Germany. [21]Helmholtz Pioneer Campus, Helmholtz Zentrum München, German Research Center for Environmental Health, Neuherberg, Germany. [22]School of Computation, Information and Technology, Technical University of Munich, Garching, Germany. [23]Munich Heart Alliance, German Center for Cardiovascular Health (DZHK E.V., Partner-Site Munich), Munich, Germany. ✉e-mail: rui.wang-sattler@helmholtz-munich.de

