## [Transparent Peer Review file · Nature Communications]

LEOPARD: missing view completion for multi-timepoint omics data via representation disentanglement and temporal knowledge transfer

Corresponding Author: Dr Rui Wang-Sattler

Version 0:

Reviewer comments:

Reviewer #1

(Remarks to the Author)

Summary

The authors use GANs to learn and reconstruct signals across “views” & time points with the aim to impute a completely missed view at a timepoint. They further benchmark their method with three existing methods. The manuscript is well-written in language per se. But major concerns and shortcomings prevent a recommendation for further consideration by Nature Communications.

Major Concerns

Below are major concerns on their manuscript.

1. The authors provide a solution to a problem that unlikely exists in real biological research. It is very rare that a data modal completely misses at a timepoint. It is more common that some samples miss some data at some timepoints. The authors completely ignore the more common scenario in their manuscript. If the authors think their approach is valuable, they should provide examples in which 1) a data modal is completely missed at a timepoint in published studies and 2) their approach provide new insights that are missing in the original studies.
2. The authors benchmark their method against three methods that are NOT designed to handle the scenario they describe. It might be worth re-reading the motivations and problem statements for references 8-10. The main goal for these methods is ‘data completeness’ and ensures standard statistical approaches work (e.g., clustering), not necessarily addresses incomplete views or ‘extrapolation’.
3. The authors claim that their method is valuable for longitudinal studies but only provide a framework and examples on pairwise studies. Is their method applicable to longitudinal studies with 3 or more timepoints? Can their method handle time as a continuous variable instead of a categorical variable?
4. The manuscript is well-written in language per se. But it is structured more suitable for an audience of Compute Scientists and/or Bioinformaticians than for the general audience of the Nature Communications. For example, most of the Data Preparation section should be described in Methods while the Results section is very light on content.

Additional Concerns

1. Generalized linear mixed effect models (GLMM) is a well-established, widely used method for handling missing data in longitudinal studies. The authors completely ignore GLMM in their manuscript. There are other imputation methods designed around GLMM that would be good to benchmark against.
 - a. Huque, M.H., Carlin, J.B., Simpson, J.A. et al. A comparison of multiple imputation methods for missing data in longitudinal studies. *BMC Med Res Methodol* 18, 168 (2018). <https://doi.org/10.1186/s12874-018-0615-6>.
 2. Figure 3: I am not convinced that their method is better. Please add p values, comparing their method with others.
 3. Figures 2-4: It is hard to read. Consider using different colors instead of different shades of same colors.
 4. As the authors correctly point out in lines 68-69, missing data can occur in different patterns - can the authors comment on how Leopard adjusts or accounts for differences in missingness.
 - a. Missing completely at random (MCAR).
 - b. Missing at random (MAR).
 - c. Missing not at random (MNAR).
 5. Lines 362-364: can the authors provide evidence of how Leopard facilitates analyses of temporal dynamics or suggest to

users how they can do so?

6. I have a slight concern with overfitting. Some comments on overfitting can be valuable to the users.

Reviewer #2

(Remarks to the Author)

Reviewer #3

(Remarks to the Author)

In this manuscript Han et al. present a method to impute missing data, when the data setup includes multiple views/modalities and multiple time points. The noteworthy result lies in the ability of LEOPARD to learn the temporal mapping of the data, when in one modality data is (completely) missing at a later time point.

I also want to commend the authors on the code availability on github and the clear instructions on obtaining the data. After some tinkering (see minor comments below), I was able to reproduce the figures in the manuscript. Well done!

LEOPARD is abiding by the state-of-the-art in computer sciences when it comes to multi-modal learning and prediction. In addition, they comply with standards of training/testing dataset separation, use of proper performance metrics, and employment of statistical necessities such as multiple testing correction, if necessary.

The performance of LEOPARD when it comes to predict/impute missing data at a later time point (t_2), when all data of one view is missing (v_2), is astonishing and could be an important contribution to the field of biomedical research, as the expanse of different data types is increasing and also societies are realising the importance of collecting health data over time.

For the work to fully support the claims made by the authors, I would like to see some additional results/comments on the following points:

1.) While it is fair to "simplify" the setting to t_1/t_2 and v_1/v_2 , I think the authors should also test what happens, if they use D_{t_1/v_2} (say, the first time point) to test. While it is generally the case that in studies the baseline is complete and follow ups are more likely to suffer from missing measurements/modalities, the claim is that LEOPARD is able to extract the mapping over time. Can it also do it backwards?

2.) The case studies are using two rather simple methods (linear regression, random forest) to establish some "biological information", however, the selected metabolites/proteins are "just" the ones which are significant/passed a threshold. They are not necessarily actual biological information. I would like to hear the authors opinion on this minor aspect of the case studies. A major problem with the case studies is their potential bias because you selected just one dataset per method. I would like to see the linear regression also on the MGH and KORA multi-omics data, as well as the random forest on the MGH and KORA metabolomics. This would help establish the claims made by the authors that LEOPARD preserves biological signals in general and not just on the datasets selected for a specific method.

3.) While mice and missForest are some of the heavy hitters in imputation, they serve a different purpose than the one the authors are addressing - mice is built for multiple imputation, missForest is built for mixed-type data - both amongst other things. This raises the question, whether these two methods are the right benchmarking partners. This is also clear, when we are playing with $obsNum=0$, where these two cannot play along, as they are not able to infer mapping along time or in between modalities. In addition, cGAN and LEOPARD have multiple pages reserved on how their hyperparameters have been trained for the benchmark, while both mice and missForest get a sentence or two on how they are run more or less in default mode. Again, not sure whether we are comparing apples and oranges here. The authors should consider either selecting other methods for comparison, design a more sophisticated benchmarking setup (including cGAN, mice and missForest in their natural habitats and showing how LEOPARD outperforms them on their soil), or helping mice and missForest to be tweaked to the data in a way, they might be able to "understand" the temporal and multi-view setup of the data.

If the authors can address the above points in a satisfactory manner, I fully support the publication of this manuscript.

Here are some minor points, I noticed while reviewing the manuscript:

- Why are you training LEOPARD for 600 epochs?
- What is a BRF model (line 625), please introduce acronyms which are not obvious.
- There is some rendering-related error due to using $\$$ in the plot.ipynb in [3]. Once remedied the code runs smoothly. The authors should address this such that the representation in github is directly usable.
- Somehow the authors are not referring to Figure 3 in the Benchmarking section starting at line 246, while they do refer to Figure 4. While clear after some moments, it would facilitate the reading of this section, if references to figures are complete.
- The illustrations in Fig 1b are suboptimal, the one in Fig 1a is practically useless. Either revise these to support with the

actual explanation you want to provide or expand on the text/legend to make it clear.

Reviewer #4

(Remarks to the Author)

The authors present a new method, LEOPARD, for imputing missing data in a longitudinal data construct. In general, I found the manuscript to be well-written. Overall, the method may be promising. However, I have concerns how widely applicable the method can be, and in its current form I have concerns with the manuscript.

While I understand that it might be a necessary assumption for some of the data scenarios, to make comparisons, the scenario of having all data at time point 1 is simply unrealistic, and the authors make this assumption with all datasets and offer no limitation statement in the Discussion.

On line 178-179, the authors state “In the training phase, the generator of the cGAN is trained on $\mathcal{D}_{v=v1,t=t1}$ train and $v=v2,t=t1$ train capture the mappings between two views.” and then say on Line 182 “In the inference phase, the generator utilizes the mappings it has learned from $t1$ to impute the missing view at $t2$.” This was extremely misleading to me, as it made it sound like cGAN was magically learning temporal dependence while only trained on the first time point. A clearer description of what is happening and the limitations with regards to temporal dependence is needed here since it comes right after your premise of longitudinal data.

Perhaps the biggest issue to me is that the authors use datasets with sample sizes of $N = 218$, $N = 2,085$, $N = 1,062$. These types of sample sizes are not typical of most omics studies. There are handfuls of large consortiums that get hundreds of samples, but a majority of research in the field operates at total numbers of samples far smaller than these sample sizes. Further, the authors’ datasets utilized have very small numbers of features/biomolecules used relative to the sample size. Most omics studies will have more features than samples, even if you have hundreds of samples. I found myself questioning how LEOPARD would hold-up with studies that are much less rich with samples throughout the manuscript. At an absolute minimum, the authors need to state this as a limitation, but to be truly valuable to the research community the authors should evaluate with how many samples their model can actually still perform well. What if I come in with 20 samples, or 40 samples, etc? Further, at least one of the datasets used should have a number of features that is more representative of real data, even if there isn’t ground truth for all proteins.

When comparing performance metrics such as PB, actual significance of model performance compared to other models should be done rather than just reporting the median value and stating it’s lowest.

Line 391: “We select the proteins based on the following QC criteria: (1) no missing or negative values; (2) at least 75% of 392 measured sample values are equal to or above the limits of detection (LOD)” – You went from 1,472 proteins down to 72 which in itself could be introducing bias into the results presented here. Similar measures were done in the other datasets as well. It should be noted in limitations that you are only using the most abundant proteins so any metrics of performance (from any model) may be inflated.

On lines 583-589, while I understand the additional advantage of LEOPARD being able to process observations where only one view is present, I do not understand why the authors chose to explicitly construct a different training scenario for LEOPARD compared to the other methods. It makes it impossible to disentangle differences in performance and what the true difference is results would be under the same training scenario.

The authors chose to implement MICE with only a single imputation; this is nonsensical to me, as the algorithm is specifically designed to do multiple imputation and have those multiple imputations be leveraged.

Again, I understand that simplifications and choices needed to be made, but I think other scenarios and guidance for model users is lacking and any potential user of the model is going to be left with unanswered questions. For example, does LEOPARD work when you have some missing values for each sample and are not just in the complete view missing scenario? Do you have to have one complete time point of data for it to work? Is there some level of missingness where a sample is not viable to be in the dataset? A brief discussion of practical issues when using the model and a frank list of caveats/limitations, in addition to the already present list of advantages, is needed.

Minor edit:

Line 229: “UMPA” should be “UMAP”

Version 1:

Reviewer comments:

Reviewer #1

(Remarks to the Author)

Summary

The authors have made substantial improvements on their manuscript. It reads much better. Nonetheless, 2 out of the 4 major concerns raised in our previous review remained unaddressed and require major clarification. Below we list our *previous concerns in italic* and our remaining concerns in normal text.

Major Concerns

Below are major concerns on their manuscript.

1. The authors provide a solution to a problem that unlikely exists in real biological research. It is very rare that a data modal completely misses at a timepoint. It is more common that some samples miss some data at some timepoints. The authors completely ignore the more common scenario in their manuscript. If the authors think their approach is valuable, they should provide examples in which 1) a data modal is completely missed at a timepoint in published studies and 2) their approach provide new insights that are missing in the original studies.

The authors have addressed the issue of “some samples miss some data at some timepoints”. But the original concern remains valid:

a) They cited 6 references in their lengthy response, including a review and five bioinformatics/technic papers. They only cited two of the six references in the revised manuscript and claimed on the two cited references: “However, the typically limited number of timepoints in current human cohorts can restrict the effectiveness of these longitudinal methods.” In other words, they have not established the need of LEOPARD in **real, biological/clinical studies** from their literature review in the manuscript.

b) So far the authors have shown potential values of LEOPARD in a very specific scenario: two timepoints, two views at each timepoint, one view completely missing at one timepoint, >80 samples in each timepoint. If this scenario is “increasingly common” as claimed, the authors should not have any problems citing 3 real, biological/clinical studies in the literature that fall under this scenario. While complete missing view is not uncommon in studies with many (far >2) timepoints, the authors unfortunately have not shown LEOPARD can handle data beyond two timepoints (see also the next concern).

c) We believe that more users will be encouraged to use LEOPARD if the authors can show that new insights are obtained when LEOPARD is applied to a real-world study having an actual (not an artificially designated) missing view. The two case studies in the manuscript are good for benchmarking but not for showcasing utility. If the authors prefer not to take this suggestion, they should at least state in “limitations” that LEOPARD has not been applied to real-world data so that users won't have unrealistic expectations.

3. The authors claim that their method is valuable for longitudinal studies but only provide a framework and examples on pairwise studies. Is their method applicable to longitudinal studies with 3 or more timepoints? Can their method handle time as a continuous variable instead of a categorical variable?

This concern has not been addressed in the revised manuscript.

a) The authors have at least two datasets (the MGH COVID dataset and the KORA metabolomics dataset) on which they can demonstrate LEOPARD's capability of handling data with three timepoints. Instead, they treated the three timepoints in the KORA metabolomics dataset in pairwise analyses (Extended Figure 3). Why?

b) The authors claim in their response: “data from additional timepoints (or views) can be accommodated by just expanding the one-hot embeddings, without modifications to the architecture.” This is not obvious. Using the approach they described, will the equivalence to Fig 2c show 2 temporal clusters or 3? We guess it can go either way. If it is the former, the approach fails.

c) It is generally accepted that more data points allow for better imputation on missing data. Will LEOPARD obtain better percent bias (PB) values from three-timepoint analysis than those from pairwise analysis as in the last panel of Extended Figure 3?

d) As a minor point, the authors should make a comment somewhere in the manuscript that LEOPARD cannot handle longitudinal data when time acts as a continuous variable.

Additional Concerns

2. Figure 3: I am not convinced that their method is better. Please add p values, comparing their method with others.

The authors added p values to the plot, which is very helpful. Unfortunately, the results from different methods are too close to tell which method is better. Can the authors add the corresponding medians in the plot so that readers don't need to refer to the text to know which method is better? The same comments apply to Fig. S1, Extended Fig. 1 & Extended Fig. 3.

New Minor Points

It may be more thorough to provide the corresponding results of Extended Fig. 1 & 2 on the other 2 datasets.

Please describe in Fig. 6 whether the biomarkers have the same sign on “Estimate” from real & imputed data.

It may be more informative to combine the two rows in Fig. 8. Change the panel arrangement from 2x3 to 3x1 if needed.

Make a comment in Discussion that the minimum sample size likely depends on data characteristics.

The legend of Extended Fig. 2 is truncated.

(Remarks on code availability)

Reviewer #2

(Remarks to the Author)

(Remarks on code availability)

Reviewer #3

(Remarks to the Author)

The authors have comprehensively changed and improved their manuscript. They have also managed to dispel most of my concerns, but not all of them.

When in my first review, I was referring to methods not being able to "play along", I was also meaning that you could in theory build an additional script around a method like missForest making it capable of handling a temporal setting. Similarly, this should be possible for GLMM. However, I do find the expansion of the number and types of benchmarking methods convincing, as well as the reasoning that even if not any of them is specifically intended to address the missing time point/view setting, they represent a lay of the land of imputation methods.

However, I would like the authors to reconsider not adding the COVID results - even if it remains unclear how some of the methods achieve better than groundtruth results (probably due to the fact that single imputations tend to underestimate the original variation in the data) - these are results that do show the difficulty of imputation (and reproducing scientific work, fwiw). My decision towards accepting this manuscript review is not depending on this.

Minor comment:

- please choose a different name than groundtruth, it is misleading
- in methods comparisons you perform Wilcoxon tests, please adjust test for multiple testing using an appropriate method (by the way, in this part of your study a Bonferroni correction would also not be inappropriate) and ensure that your interpretation of the results are updated accordingly
- in figures with/containing box plots (especially when illustrating PB) consider using a log scale on the y-axis, to make the box plots more "discernible"

I support the publication of this manuscript on the condition that the results are fully reproducible (see my remarks on code review) and my above comments have been addressed.

Daniel Stekhoven

(Remarks on code availability)

The code is well written and uses established tools of ensuring reproducibility. However, there are still some issues, when it comes to reproducing the results. The authors will need to address these before the manuscript is published to ensure that their results are indeed reproducible.

- some data objects seem to have wrong names (e.g. `~/data/MGH/COVID_imputed/obsNum = 50/LEOPARD.csv`, iso `~/data_LEOPARD.csv`)
- avoid using whitespaces in filenames
- it still requires some hacking to get all going, the documentation should include a link to a jupyter instance to allow for a straightforward running of the method (something that was actually possible in my first review but seemingly not available anymore or I did not find it)

Reviewer #4

(Remarks to the Author)

The authors have addressed many of the reviewers' comments, and the quality of the manuscript has been improved. However, I still have issues with several of the choices made.

1. Per several reviewers' comments, which were glossed over by the authors by stating "we did not focus on missing value imputation, as existing methods already address this effectively". This simply isn't true (at least in fields like untargeted proteomics and transcriptomics) and simulating data, as the authors did, with values going missing completely at random (MCAR) is not reflective of the mixed missing value mechanisms in several omics, such as untargeted proteomics.
2. The use of dimension reduction to assess performance is of limited utility. Because it is based on the entire omics profile there are multiple ways to arrive at similar looking plots.
3. In their response to Reviewer 1, the authors claim that they use imputation methods like MICE, missForest, KNNimpute because methods for handling complete missing views in longitudinal studies don't exist. While they did include one new method, all of the other methods presented are designed for single datasets and don't really take advantage of the multi-omics data structure, creating an inherently unfair comparison. This is pointed out by multiple reviewers. Why not use Multiple Imputation Multiple Factor Analysis (MI-MFA) by Voillet et al, 2016, or Late Fusion Incomplete View Multi-View Clustering (LF-IMVC) by Liu et al 2019, or any other of the multiple methods for multi-view data?
4. The authors were asked "The authors claim that their method is valuable for longitudinal studies but only provide a framework and examples on pairwise studies. Is their method applicable to longitudinal studies with 3 or more timepoints?"

Can their method handle time as a continuous variable instead of a categorical variable?" While I understand that there may not be readily available datasets for a real-world application, they also have a number of simulated datasets in the manuscript and did not attempt to simulate anything to address these questions.

5. In response to Reviewer 4, while the authors make some concessions to extend the number of features to as many as 322, this is still far smaller than a typical proteomics or transcriptomics dataset, as two examples. In practice, there is nothing in this manuscript to give me faith that these methods would withstand the more typical cases in biological data.

6. Further, the question about minimal sample sizes was answered, but it showed a required sample size of 60 to 80 which, in practice, is rarely a luxury that researchers have. I believe that this makes this method much less compelling to a wide range audience.

(Remarks on code availability)

Version 2:

Reviewer comments:

Reviewer #1

(Remarks to the Author)

We thank the authors for their additional work and efforts to improve their manuscript. The authors have either done additional work to address our concerns or admitted limitations in Discussion. While we strongly believe that it is a major weakness of the manuscript not to demonstrate the value of LEOPARD in real-world OMICS studies, it is time for a final decision by the Editor.

Minor Comments for improvement.

Figure 6:

The use of hollow points to show sign similarity is difficult to highlight global behavior. A better representation of showing whether the sign (effect size) is correlated between the imputed and the real dataset is to do a scatter plot of the effect size per dataset on each axis. Then you can quantify the correlation and the % of features with the same directionality.

Figure 3 / Results (lines 314-318):

The IQR is not a very descriptive statistic on its own when describing PB because it provides information on 'consistency' rather than accuracy. For example, if a method has a high, yet consistent bias, it will still achieve a low IQR (albeit being consistently wrong). We recommend the authors either remove the IQR or provide the median PB in tandem to showcase both stability (IQR) and accuracy (median PB) for each method.

(Remarks on code availability)

Reviewer #2

(Remarks to the Author)

(Remarks on code availability)

N/A

Reviewer #3

(Remarks to the Author)

I thank the authors for further addressing my comments. The manuscript has matured and reached a state where it presents a nice method for a specific task in a very comprehensive way. While some comments from other reviewers will still need some attention - among which was an earlier comment also raised by me regarding the ability to compare LEOPARD with other methods essentially doing something else - I stand by my previous assessment that for an assessment of the performance of LEOPARD, the chosen methods and the discussions thereof are sufficient.

Adding the MGH Covid data set has further improved the manuscript. Thank you.

Regarding reproducibility: I was still able to reproduce your plots and results using the github repository - well done (one cannot point that out often enough!). However, your colab does not contain the data folder, therefore, when reaching chunk 5 where the MGH COVID dataset is supposed to be processed, the fun stops... please fix this.

(Remarks on code availability)

The code and data on github is allowing for the reproduction of the authors figures and results. The code is written according

to the state of the art in R coding containing a sufficient amount of documentation.

A provided colab Jupyter notebook link was missing the data to be functional, however, contained the results in its cache (I know that's not reproduction, but alas...).

Reviewer #5

(Remarks to the Author)

Upon review of the manuscript and the authors' responses, I have found Reviewer #4's questions/comments to be sufficiently addressed.

(Remarks on code availability)

Reviewer #1 (Remarks to the Author):

Summary

The authors use GANs to learn and reconstruct signals across “views” & time points with the aim to impute a completely missed view at a timepoint. They further benchmark their method with three existing methods. The manuscript is well-written in language per se. But major concerns and shortcomings prevent a recommendation for further consideration by Nature Communications.

Reply: Thank you for your detailed review and valuable feedback. We are committed to addressing the concerns raised and improving our manuscript accordingly.

Major Concerns

Below are major concerns on their manuscript.

1. The authors provide a solution to a problem that unlikely exists in real biological research. It is very rare that a data modal completely misses at a timepoint. It is more common that some samples miss some data at some timepoints. The authors completely ignore the more common scenario in their manuscript. If the authors think their approach is valuable, they should provide examples in which 1) a data modal is completely missed at a timepoint in published studies and 2) their approach provide new insights that are missing in the original studies.

Reply: Thank you for highlighting concerns about the applicability of our method, LEOPARD, for completing missing views in multi-timepoint omics data.

We acknowledge that completely missing data points are perceived as uncommon; however, they are increasingly present in multi-omics studies, especially in longitudinal contexts. The literature¹ provides examples where entire modalities are missed due to high costs. Furthermore, methodologies such as spline interpolation and Gaussian process have been used to impute unobserved time points with completely missing data in multi-omics analyses²⁻⁵. A recent review⁶ introduces that Gaussian processes can be beneficial in the analysis of incomplete omics profiles in longitudinal health studies.

The datasets used in existing studies are mainly constructed from microbiome data, where data are collected at numerous timepoints. For example, the microbiome dataset used by MEFISTO⁵ includes 24 timepoints. However, the omics data in human cohorts are often available at a limited number of timepoints. For instance, data collection of the KORA cohort, one of the earliest and largest human cohorts, is performed at regular intervals. Its targeted metabolomics data are available for three timepoints, and its proteomics data only for two timepoints. The limited number of timepoints has greatly limits the effectiveness of existing methods, such as spline interpolation and Gaussian process. Thus, we propose LEOPARD in this study aiming to facilitate multi-omics research and overcome the obstacles to biomarker discovery and casual inference caused by missing views. In this revision, we briefly introduced existing studies involving missing timepoints imputation in the Introduction section from lines 92 to 98:

To address the complexities of longitudinal data, numerous effective imputation methods have been developed based on the generalized linear mixed effect model (GLMM)⁷. Existing studies have also explored the use of spline interpolation and Gaussian processes to extrapolate or interpolate missing timepoints^{5,6}. However, the typically limited number of timepoints in current human cohorts can restrict the

effectiveness of these longitudinal methods. Given these challenges, there is a growing need for view completion methods that are specifically designed for multi-timepoint omics data.

Inspired by your feedback and suggestions from Reviewer #4, in this revision we have tested LEOPARD under more realistic scenarios where observed views also contain missing values. We evaluated how missing data in observed view impact the performance of LEOPARD and other methods for missing view completion by varying the proportions of missing values to 1%, 3%, 5%, 10%, and 20%. As LEOPARD is tailored for missing view completion, we did not focus on missing value imputation, as existing methods already address this effectively. The strategy of our simulations is detailed in the revised main text, from lines 347 to 371:

Apart from missing views, the presence of missing data points in observed view is also very common in omics analysis. Therefore, LEOPARD is designed to tolerate a small number of missing data points in the observed views (Methods). We further use KORA metabolomics dataset, which has the largest sample size, to evaluate the performance of different methods when observed views contain missing values. We simulate missing values by randomly masking 1%, 3%, 5%, 10%, and 20% of the data in the observed views (*maskObs*) under the assumption that data points are missing completely at random (MCAR). The experiment is repeated 10 times for each specified proportion, and the results are evaluated by PB and UMAP.

The results are summarized in Extended Figs.1 and 2 and the revised main text, from lines 459 to 472:

Our findings indicate that all the methods experienced an increase in PB when the observed views contain missing values (Extended Fig. 1). However, LEOPARD and missForest are robust to the missing data points in terms of PB. In contrast, cGAN and GLMM exhibit high sensitivity to missing values. Method cGAN does not show similar improvement with the increase of *obsNum* as it performs in Fig. 3 (middle row) and is gradually surpassed by missForest. GLMM overall exhibits higher PB than the other methods.

The UMAP plots (Extended Fig. 2) further demonstrate that LEOPARD's performance remains comparable to scenarios with no missing data in the observed views (Fig. 4, middle row), unlike the other methods which display overfitting or a great loss of data variation. Although LEOPARD outperforms other methods, we observed a change in the distribution of the imputed data (blue dots): as *maskObs* increases, these blue dots begin to shrink toward their center and become more concentrated. This leads to a reduced coverage of the outer areas of the ground truth embeddings (green dots) and suggests that the imputed data might not capture the full variability of the data when the proportion of missing data is high.

We additionally discussed the results in Discussion section from lines 656 to 663:

Considering the common occurrence of missing values in real-world omics data, LEOPARD is designed to tolerate a small proportion without compromising robustness. However, we observed that LEOPARD struggles to capture the full diversity present in the ground truth as *maskObs* increases to 20% (Extended Fig. 2). It is preferable for the input data for LEOPARD to contain less than 10% missing data points. Higher proportions of missing values are ideally addressed by generic imputation methods before

processing with LEOPARD. Additionally, we assumed that the missing data were MCAR. Additional bias could be introduced if data points are missing at random (MAR) or missing not at random (MNAR) in real-world scenarios.

In our evaluation, LEOPARD shows enhanced robustness by including a few samples from the incomplete views in the training process ($obsNum > 0$). This aspect of LEOPARD could be particularly beneficial in omics studies, whereas the cost of omics profiling can be higher than \$400 per sample. Researchers might use LEOPARD to estimate unmeasured data for preliminary analyses before undertaking the financially and logistically intensive process of comprehensive data measurement.

We believe that these additional experiments and clarification sufficiently address your concerns and highlight the potential impact of our study.

Extended Fig. 1. Performance of missing view completion when observed views have missing values.

The average PB are computed for $\mathcal{D}_{v=v2, t=t2}^{test}$ of the KORA metabolomics dataset across 10 repeated completions, under each proportion of masked data points in the observed views ($maskObs$). In each repetition, the data points are masked randomly. Each dot represents a PB value for a variable. Please note that LM is used instead of GLMM when $obsNum = 0$.

Extended Fig. 2. UMAP visualizes the imputed data obtained when observed views have missing values.

UMAP models are initially fitted with the training data from the KORA multi-omics dataset (c, t_1 : S4, t_2 : F4). Subsequently, the trained models are applied to the corresponding observed data (represented by red and blue dots for t_1 and t_2) and the data imputed by different methods (represented by green dots) under $obsNum = 200$ and varying $maskObs$. Please note that, for each $maskObs$, only the repetition that exhibits the lowest median of PB are visualized. The distributions of red and blue dots illustrate the variation across the two timepoints, while the similarity between the distributions of blue and green dots indicates the quality of the imputed data. A high degree of similarity suggests a strong resemblance between the imputed and observed data.

2. The authors benchmark their method against three methods that are NOT designed to handle the scenario they describe. It might be worth re-reading the motivations and problem statements for references 8-10. The main goal for these methods is ‘data completeness’ and ensures standard statistical approaches work (e.g., clustering), not necessarily addresses incomplete views or ‘extrapolation’.

Reply: Thank you for the recommendation to revisit the motivations and problem statements of references 8-10. We acknowledge that tools like MICE⁸, missForest⁹, and others including KNNimpute¹⁰ were not originally developed to handle complete missing views in longitudinal studies. As currently there is a lack of methods specifically designed for missing view completion in multi-timepoint omics data, we benchmarked LEOPARD against well-recognized methods that can represent different algorithms. Our

findings suggest that these generic methods do not provide robust results for missing view completion, which highlights the need for tailored approaches.

To have a more comprehensive evaluation in our longitudinal scenario, we additionally evaluated a GLMM-based method in this revision. We also added a brief explanation to clarify our choice of benchmarking partner in our revision from lines 320 to 331:

Due to the lack of established methods specifically designed for missing view completion in multi-timepoint omics datasets, we benchmarked LEOPARD against three widely recognized generic imputation methods: missForest, PMM (predictive mean matching), and GLMM (generalized linear mixed effect model), as well as a cGAN model designed for this study. The cGAN serves as a reference model to demonstrate how existing neural network approaches, typically suited for cross-sectional data, perform in longitudinal scenarios. MissForest, as a representative non-parametric method, was chosen for its robustness and ability to handle complex, non-linear relationships among variables. PMM and GLMM, both implemented within the MICE framework, represent established multiple imputation methods that not only address missing values but also allow for the assessment of imputation uncertainty. GLMM, with its ability to capture temporal patterns inherent in longitudinal data, is particularly advantageous for data imputation in longitudinal scenarios.

We hope this clarification can better explain the rationale behind our benchmark partner selection.

3. The authors claim that their method is valuable for longitudinal studies but only provide a framework and examples on pairwise studies. Is their method applicable to longitudinal studies with 3 or more timepoints? Can their method handle time as a continuous variable instead of a categorical variable?

Reply: Thank you for the insightful questions. In our study, LEOPARD has been evaluated using data from only two timepoints, largely due to limitations related to data availability. However, the architecture of LEOPARD can theoretically be extended to three or more timepoints. During the training process, the content and temporal encoders of LEOPARD are shared by samples from all different views and timepoints to extract content and temporal representations. The view and timepoint categories are indicated with one-hot embeddings. Thus, data from additional timepoints (or views) can be accommodated by just expanding the one-hot embeddings, without modifications to the architecture.

During the development of LEOPARD, we also explored more intricate architectures inspired by StarGAN¹¹, a framework designed for multi-domain style transfer. However, our initial experiments suggested that more complex models did not yield better performance, given by limited sample sizes. As datasets including more timepoints become available, we expect to further evaluate LEOPARD's effectiveness in these expanded contexts.

Regarding your question about handling time as a continuous variable, LEOPARD currently treats timepoints as categorical variables. This design is primarily tailored for multi-timepoint omics studies where data acquisition at multiple, distinct timepoints is common due to high costs and technical challenges. As the field rapidly advances and data become more abundant, we look forward to enhancing LEOPARD's applicability and usability for more complex scenarios.

4. The manuscript is well-written in language per se. But it is structured more suitable for an audience of Compute Scientists and/or Bioinformaticians than for the general audience of the Nature Communications. For example, most of the Data Preparation section should be described in Methods while the Results section is very light on content.

Reply: We greatly appreciate your feedback regarding the structure of our manuscript. We recognize the importance of making our findings accessible to the diverse readership of Nature Communications, which includes researchers from a variety of scientific backgrounds beyond computational sciences and bioinformatics. In response to your suggestions, we have made several adjustments to ensure the content is more relevant to all readers:

- 1) **Adjustment of technical details:** We have relocated the in-depth technical details and equations from the Results section to the Methods section. This helps keep the Results section focused on our findings, making it more accessible to a diverse audience.
- 2) **Introduction of a terminology section:** We have included a terminology section at the beginning of the Methods section. This section introduces key terms and concepts in a clear and concise way, ensuring that all readers can fully understand the applied methodologies and the significance of our findings.
- 3) **Revision of the Data Preparation section:** We have relocated the detailed description of each benchmark dataset to the Methods section. Additionally, we included a “Characterization of benchmark datasets” section as the first paragraph in the Results section to provide a concise overview of our benchmark datasets. Detailed revisions are described in lines 160 to 175 and are briefly outlined below:

Characterization of benchmark datasets

We evaluate LEOPARD using three real longitudinal omics datasets. These distinct datasets are designed based on data variations, time spans, and sample sizes (Table 1, Methods). The first two are mono-omics datasets, constructed with the proteomics data from MGH COVID study and the metabolomics from the KORA cohort, respectively. Views in both datasets correspond to panels or biochemical classes. Missingness in these datasets exemplifies a common issue encountered in longitudinal studies where data from certain panels or biochemical classes are incomplete in some but not all timepoints. The third dataset is a multi-omics dataset consisting of both metabolomics and proteomics data from the KORA cohort. In this dataset, views correspond to different omics. This dataset exemplifies the situation where data of a type of omics is incomplete. These three datasets comprise data of two views ($v1$ and $v2$) from two timepoints ($t1$ and $t2$). The samples from each dataset are divided into training, validation, and test sets in a 64%, 16% and 20% ratio, respectively. We use $\mathcal{D}_{v,t}^{\text{split}}$ to denote data split from different views and timepoints. The test data in $v2$ at $t2$ (i.e. $\mathcal{D}_{v=v2,t=t2}^{\text{test}}$) are masked for performance evaluation. To further evaluate LEOPARD’s applicability, the KORA metabolomics dataset is extended to span three timepoints, and $\mathcal{D}_{v=v2,t=t3}^{\text{test}}$ is masked for evaluation (Methods).

These modifications aim to align our work with the expectations of a general scientific audience. We are grateful for your guidance in refining our manuscript.

Additional Concerns

1. Generalized linear mixed effect models (GLMM) is a well-established, widely used method for handling missing data in longitudinal studies. The authors completely ignore GLMM in their manuscript. There are other imputation methods designed around GLMM that would be good to benchmark against.

(Huque, M.H., Carlin, J.B., Simpson, J.A. et al. A comparison of multiple imputation methods for missing data in longitudinal studies. BMC Med Res Methodol 18, 168 (2018). <https://doi.org/10.1186/s12874-018-0615-6>)

Reply: Thank you for your constructive feedback. Inspired by your suggestions, we have incorporated a GLMM model, which better suits our scenario, into our evaluation. The selected GLMM model assumes a constant residual error variance for all individuals, as subject-specific residual variances may not be well estimated from longitudinal data with few repeated measurements. When *obsNum* is zero, data for *v2* at *t2* are assumed to be completely missing, and GLMM cannot be trained due to limited longitudinal information. In this scenario, we trained a linear model (LM) to complete missing view, which allows us to further evaluate how the performance of GLMM compares to that of the simpler LM method.

Furthermore, in response to the concerns from Reviewer #4 about the performance and efficiency for datasets having more variables than samples, we reconstructed the MGH COVID dataset to contain 322 and 295 proteins for *v1* and *v2*, respectively. In our evaluation, we found GLMM to be extremely time-consuming on this dataset—it took more than two weeks to complete a single imputation on our device with an Intel(R) Core(TM) i9-10885H CPU @ 2.40GHz processor and 32 GB of memory, running Windows 11 (64 bit) OS. Therefore, we only selected the top 100 highly Spearman-correlated variables for each variable that requires imputation. We described this in the Methods section, lines 980 to 986 and lines 1023 to 1033:

Multiple imputations by PMM, LM, and GLMM were performed using the R packages *mice*⁸ and *micemd*¹². Each method's imputations were performed five times ($m = 5$) with a *maxit* value set to five. The PMM model was built using argument *method* = "pmm". When *obsNum* = 0, data of *v2* at *t2* are assumed as completely missing. In this scenario, the LM method was employed using *method* = "norm". When *obsNum* is a non-zero value, the GLMM model was built using *method* = "2l.glm.norm".

Different imputation methods may require specific data structures for input. ... We adapted the input data accordingly to accommodate these specific requirements:

- For the multiple imputation methods in MICE family, the input data were constructed with training data (identical to that used for *missForest*) and test data (including $D_{v=v1,t=t1}^{\text{test}}$, $D_{v=v2,t=t1}^{\text{test}}$ and $D_{v=v1,t=t2}^{\text{test}}$). To ensure test data remained unused for model training, a logical vector with TRUE assigned to test samples was passed to the *ignore* argument. Masked values were filled with NA in the matrix.
- When building the GLMM model, the input data additionally contained sample IDs and timepoint labels. A constant residual error variance is assumed for all individuals. Building GLMM model for a large dataset is extremely time consuming; thus, for the MGH COVID dataset, we selected the top 100 highly Spearman-correlated proteins for each protein requiring imputation. The selected proteins were incorporated into the imputation process by passing to the argument *predictorMatrix*.

The updated performance of LEOPARD and other methods including PMM and LM/GLMM is shown in Figs. 3 and 4. The performance of each individual imputation generated by the multiple imputation methods is shown in Figs. S1 and S2.

Fig. 3. PB of imputed results for the test sets of three benchmark datasets.

PB evaluated on $D_{v=v_2, t=t_2}^{\text{test}}$ of the MGH COVID proteomics dataset (upper row), KORA metabolomics dataset (middle row), and KORA multi-omics dataset (lower row). Each dot represents a PB value for a variable. Please note that LM is used for imputation instead of GLMM when $obsNum = 0$. Significance level: not significant (ns), $P < 0.05$ (*), $P < 0.01$ (**), and $P < 0.001$ (***)

Fig. 4. UMAP representations of the imputed values and corresponding observed data from the benchmark datasets.

UMAP models are initially fitted with the training data from the MGH COVID proteomics dataset (upper row, t_1 : D0, t_2 : D3), KORA metabolomics dataset (middle row, t_1 : F4, t_2 : FF4), and KORA multi-omics dataset (lower row, t_1 : S4, t_2 : F4). Subsequently, the trained models are applied to the corresponding observed data (represented by red and blue dots for t_1 and t_2) and the data imputed by different methods (represented by green dots) under the setting of $obsNum = 100$ for the MGH COVID dataset and $obsNum = 200$ for the two KORA-derived datasets. The distributions of red and blue dots illustrate the variation across the two timepoints, while the similarity between the distributions of blue and green dots indicates the quality of the imputed data. A high degree of similarity suggests a strong resemblance between the imputed and observed data.

2. Figure 3: I am not convinced that their method is better. Please add p values, comparing their method with others.

Reply: Thank you for your feedback. We have updated Fig. 3 to include significance levels, allowing for a clearer statistical comparison between LEOPARD and others.

3. Figures 2-4: It is hard to read. Consider using different colors instead of different shades of same colors.

Reply: Thank you for your suggestions. Different colors have been used in Figs. 2 - 4.

4. As the authors correctly point out in lines 68-69, missing data can occur in different patterns - can the authors comment on how Leopard adjusts or accounts for differences in missingness.

a. Missing completely at random (MCAR).

b. Missing at random (MAR).

c. Missing not at random (MNAR).

Reply: Thank you for your insightful comments. Tailored for missing views in multi-omics scenarios, LEOPARD does not make explicit assumptions about the nature of the missingness. LEOPARD only assumes that generalized patterns specific to each view and timepoint can be extracted from observed data and re-entangled to reconstruct the observed data and impute missing data. Although LEOPARD does not make explicit assumptions about the missingness mechanism, our evaluation provides insights into how LEOPARD account for these differences:

MNAR or MAR: Missing views or missing modalities in multi-omics studies can largely depend on budget constraints, resource limitations, and decisions made by healthcare specialists and/or study participants, which makes values tend to be missing simultaneously across an entire view. The missingness in this case is at least MAR, or even MNAR if the missingness is related to the unobserved data themselves. In our study, the evaluation under $obsNum = 0$ simulates this scenario (Fig. 3). When one view at one timepoint is entirely missing, building direct mapping between variables can be difficult. LEOPARD learns generalized patterns by minimizing the loss computed on the observed and reconstructed data.

MAR: When we evaluate LEOPARD under $obsNum > 0$, one view at one timepoint is incomplete, with a few samples observed. The observed samples are selected randomly, and the missingness is unrelated to the unobserved data and could be MCAR. In this scenario, LEOPARD is able to captures the content and temporal representations specific to that incomplete data block. Compared to the condition $obsNum = 0$, the model training is easier, and the PB is much lower (Fig. 3).

MCAR: Missing views have structured patterns and are unlikely to be MCAR. However, inspired by your comments and suggestions from Reviewer #4, we evaluated the case where observed views contain MCAR data points. In this case, the missing data points are excluded from the computation of the reconstruction loss during back-propagation. As previously discussed in our response to your first concern and illustrated in Extended Figs. 1 and 2, LEOPARD's capability for missing view completion is robust to a few MCAR data points. But a high proportion of missing data points should ideally be imputed using established imputations methods before using LEOPARD for missing view completion.

Determining the missingness mechanism for missing views is challenging, especially for multi-omics data, due to the complexity of multi-omics data acquisition. Many existing methods¹³⁻¹⁵ that address missing views/modalities adopt the strategy of avoiding “featurizing missingness” to mitigate the risk of making invalid assumptions and resulting in bias results. We noticed that there is currently a lack of studies on evaluating how existing methods for missing views/modalities perform under different missingness categories. Inspired by your comments, our future research may systematically investigate this.

5. Lines 362-364: can the authors provide evidence of how Leopard facilitates analyses of temporal dynamics or suggest to users how they can do so?

Reply: Thank you for the insightful comment. Even with evaluations performed at only two timepoints in our paper, the temporal representations extracted by LEOPARD reveal interesting aspects. Fig. RL1 displays the UMAP embeddings of the temporal representations factorized from the KORA multi-omics dataset at epoch 600. Dark colors represent data from the first timepoint, while light colors represent data from the second. We notice that most points of dark and light colors cluster within their respective groups, indicating the temporal information from the same timepoints is clustered together. But we also notice a few light-colored points (indicated with arrows) mixed into the dark-colored group, and vice versa. Without further investigation and additional phenotypes, we cannot determine whether this phenomenon is caused by model bias or, more interestingly, by the biological signals of these samples differing from others (perhaps the donors of these blood samples are biologically younger than other individuals in the cohort, despite having passed the same number of years and having similar chronological age). This could provide valuable insights for analyses of aging and the order of omic events. We have now included a brief description about this in the Discussion section, from lines 634 to 638:

By analyzing the extracted temporal embeddings, LEOPARD could enable the inference of the temporal ordering of omics changes, which would be particularly valuable when there is a discrepancy between biological and chronological order. As the number of data timepoints increases, LEOPARD is expected to offer new opportunities in predictive healthcare with multi-timepoint omics data.

Fig. RL1. UMAP embeddings of temporal representations factorized from the KORA multi-omics dataset.

The embeddings from the same timepoint cluster together. One cluster is formed by the data at t_1 (dark blue and red dots, regardless of v_1 or v_2), and the other is formed by the data at t_2 (light blue and red dots, regardless of v_1 or v_2). However, some data points are mixed into the clusters of other timepoints (indicated with arrows), which implies that the biological signals of these samples might differ from others in terms of temporal patterns.

6. I have a slight concern with overfitting. Some comments on overfitting can be valuable to the users.

Reply: Thank you for raising the important issue of potential overfitting in LEOPARD. We address this concern from two perspectives to demonstrate the measures we have taken to mitigate overfitting:

Theoretical approach: Unlike methods that learn direct mappings between variables, LEOPARD employs representation disentanglement and temporal knowledge transfer, which inherently enables LEOPARD to learn more generalized and structured representations specific to different views and timepoints. This mechanism makes LEOPARD less likely to overfit the training data. Additionally, during training, we added further constraint when LEOPARD reconstructs observed views with its learned content and temporal representations. For example, LEOPARD uses the content information of $v1$ (extracted from $\mathcal{D}_{v=v1,t=t1}^{\text{train}}$) and the temporal information of $t2$ (extracted from $\hat{\mathcal{D}}_{v=v2,t=t2}^{\text{train}}$) to reconstruct $\mathcal{D}_{v=v1,t=t2}^{\text{train}}$. Then it gradually optimizes its model by minimizing the difference between this reconstructed $\hat{\mathcal{D}}_{v=v1,t=t2}^{\text{train}}$ and the observed $\mathcal{D}_{v=v1,t=t2}^{\text{train}}$. We avoid using the $v1$ and $t2$ information extracted from the same source of data ($\mathcal{D}_{v=v1,t=t2}^{\text{train}}$ in this case) to recover itself. Entangling content and temporal information from other observed views helps LEOPARD learn highly generalized features within the data. Moreover, in this example, the $t2$ information is extracted from $\hat{\mathcal{D}}_{v=v2,t=t2}^{\text{train}}$, which is the imputed data of the unobserved view. If the temporal information from this imputed data can help LEOPARD reconstruct reliable views, it suggests that the unobserved view has been effectively imputed. We briefly mentioned this process in the Methods section of this revision, lines 896 to 903:

To ensure the representation disentangler can capture the highly structured data pattern, we only compute the \mathcal{L}_{rec} on the data generated from content and temporal representations derived from different source classes. For instance, $\langle z_content_{v=v1,t=t1}^i, z_temporal_{v=v2,t=t1}^i \rangle$ or $\langle z_content_{v=v2,t=t2}^i, z_temporal_{v=v1,t=t1}^i \rangle$. Data generated from representation pairs of the same views and timepoints, such as $\langle z_content_{v=v1,t=t2}^i, z_temporal_{v=v1,t=t2}^i \rangle$ are not used for optimization. This design imposes additional restraints, and LEOPARD is tamed to learn more generalized representations, which helps prevent overfitting.

Practical Measures: LEOPARD employs the toolkit tensorboard¹⁶, which allows the training process to be monitored in real-time. After initiating the visualization window, users can monitor different loss metrics to decide the optimal point to stop training. Early stopping can also be toggled to further prevent overfitting. Details on these settings can be found in our manual, which provides examples of how to train LEOPARD effectively. We have also updated our Code availability section (lines 1133 to 1136) accordingly:

Code availability
The source code and implementation details of LEOPARD are freely available at our GitHub repository (<https://github.com/HAN-Siyu/LEOPARD>). Detailed documentation and examples can be found in the Manual, which is also available at this repository.

We hope the updates in this revision have addressed your concerns and made the study more thorough and robust. We thank you again for your constructive feedback and detailed suggestions. Other modifications to the manuscript are highlighted in the text.

References

1. Yu, G. & Hou, S. Integrative nearest neighbor classifier for block-missing multi-modality data. *Stat. Methods Med. Res.* **31**, 1242–1262 (2022).
2. Ruiz-Perez, D. *et al.* Dynamic Bayesian Networks for Integrating Multi-omics Time Series Microbiome Data. *mSystems* **6**, (2021).
3. Teo, G., Bin Zhang, Y., Vogel, C. & Choi, H. PECAPlus: statistical analysis of time-dependent regulatory changes in dynamic single-omics and dual-omics experiments. *npj Syst. Biol. Appl.* **4**, 3 (2017).
4. Bodein, A., Chapleur, O., Droit, A. & Lê Cao, K.-A. A Generic Multivariate Framework for the Integration of Microbiome Longitudinal Studies With Other Data Types. *Front. Genet.* **10**, (2019).
5. Velten, B. *et al.* Identifying temporal and spatial patterns of variation from multimodal data using MEFISTO. *Nat. Methods* **19**, 179–186 (2022).
6. Velten, B. & Stegle, O. Principles and challenges of modeling temporal and spatial omics data. *Nat. Methods* **20**, 1462–1474 (2023).
7. Huque, M. H., Carlin, J. B., Simpson, J. A. & Lee, K. J. A comparison of multiple imputation methods for missing data in longitudinal studies. *BMC Med. Res. Methodol.* **18**, 168 (2018).
8. Buuren, S. van & Groothuis-Oudshoorn, K. **mice** : Multivariate Imputation by Chained Equations in R. *J. Stat. Softw.* **45**, (2011).
9. Stekhoven, D. J. & Bühlmann, P. MissForest--non-parametric missing value imputation for mixed-type data. *Bioinformatics* **28**, 112–8 (2012).
10. Troyanskaya, O. *et al.* Missing value estimation methods for DNA microarrays. *Bioinformatics* **17**, 520–5 (2001).
11. Choi, Y., Uh, Y., Yoo, J. & Ha, J.-W. StarGAN v2: Diverse Image Synthesis for Multiple Domains. in *2020 IEEE/CVF Conference on Computer Vision and Pattern Recognition (CVPR)* 8185–8194 (IEEE, 2020). doi:10.1109/CVPR42600.2020.00821.
12. Audigier, V. & Resche-Rigon, M. micemd: Multiple Imputation by Chained Equations with Multilevel Data. at <https://cran.r-project.org/package=micemd> (2023).
13. Swamy, Vinitra, *et al.* MultiModN—Multimodal, Multi-Task, Interpretable Modular Networks. *Advances in Neural Information Processing Systems (NeurIPS)* **36** (2024).
14. Cai, L., Wang, Z., Gao, H., Shen, D. & Ji, S. Deep Adversarial Learning for Multi-Modality Missing Data Completion. in *Proceedings of the 24th ACM SIGKDD International Conference on Knowledge Discovery & Data Mining* 1158–1166 (ACM, New York, NY, USA, 2018). doi:10.1145/3219819.3219963.
15. Tran, L., Liu, X., Zhou, J. & Jin, R. Missing Modalities Imputation via Cascaded Residual Autoencoder. in *2017 IEEE Conference on Computer Vision and Pattern Recognition (CVPR)* 4971–4980 (IEEE, 2017). doi:10.1109/CVPR.2017.528.
16. Abadi, M. *et al.* TensorFlow: Large-Scale Machine Learning on Heterogeneous Distributed Systems. (2016).

Reviewer #2 (Remarks to the Author):

Reply: Thank you for your valuable feedback, which has significantly contributed to the improvement of our manuscript.

Reviewer #3 (Remarks to the Author):

In this manuscript Han et al. present a method to impute missing data, when the data setup includes multiple views/modalities and multiple time points. The noteworthy result lies in the ability of LEOPARD to learn the temporal mapping of the data, when in one modality data is (completely) missing at a later time point. I also want to commend the authors on the code availability on github and the clear instructions on obtaining the data. After some tinkering (see minor comments below), I was able to reproduce the figures in the manuscript. Well done!

LEOPARD is abiding by the state-of-the-art in computer sciences when it comes to multi-modal learning and prediction. In addition, they comply with standards of training/testing dataset separation, use of proper performance metrics, and employment of statistical necessities such as multiple testing correction, if necessary.

The performance of LEOPARD when it comes to predict/impute missing data at a later time point (t_2), when all data of one view is missing (v_2), is astonishing and could be an important contribution to the field of biomedical research, as the expanse of different data types is increasing and also societies are realising the importance of collecting health data over time.

For the work to fully support the claims made by the authors, I would like to see some additional results/comments on the following points.

Reply: Thank you for your thorough and helpful feedback on our manuscript. We greatly appreciate your efforts in evaluating our code availability and reproducibility.

1.) While it is fair to "simplify" the setting to t_1/t_2 and v_1/v_2 , I think the authors should also test what happens, if they use D_{t_1/v_2} (say, the first time point) to test. While it is generally the case that in studies the baseline is complete and follow ups are more likely to suffer from missing measurements/modalities, the claim is that LEOPARD is able to extract the mapping over time. Can it also do it backwards?

Reply: Thank you for your constructive feedback. Indeed, LEOPARD is able to perform "backwards" imputation. This capability is derived from "arbitrary style transfer", a concept leveraged from the computer vision field and underpinning LEOPARD, which allows a style of any image to be transferred to the content of another.

Inspired by your suggestion, we have expanded our tests to include not only D_{t_1/v_2} but also D_{t_1/v_1} and D_{t_2/v_1} , covering all possible combinations (See Extended Fig. 3). Furthermore, we have incorporated the second follow-up study from KORA cohort, spanning approximately 14 years, to test LEOPARD on D_{t_3/v_2} (See Extended Fig. 3).

We have added a new section in our revision that explores the applicability of LEOPARD, which includes using the KORA metabolomics dataset to assess if LEOPARD can arbitrarily transfer temporal knowledge. Details of the experiments are described in the Methods section, lines 782 to 791, as well as 1108 to 1117:

Data preprocessing

To evaluate LEOPARD's capability for arbitrary temporal knowledge transfer, we further expanded the KORA metabolomics dataset to include data from the baseline study (S4, as t_1) and the second follow-up

study (FF4, as t_3), spanning approximately 14 years. We divided the metabolites data into two views using the same strategy as we used for the original KORA metabolomics dataset. Due to different QC results across the two analytical kits, two metabolites, specifically PC aa C38:1 in v_1 and C16:2 in v_2 , were excluded (see Table S2). The final dataset comprised 102 metabolites with 614 individuals who have data at both timepoints. These samples were divided into training, validation, and test sets with a ratio of 64%, 16%, and 20% respectively, corresponding to 393, 98, and 123 samples. The data in $\mathcal{D}_{v=v_2, t=t_3}^{\text{test}}$ are masked for performance evaluation.

Arbitrary temporal knowledge transfer

In the previous evaluation, we assessed the performance of each method on $\mathcal{D}_{v=v_2, t=t_2}^{\text{test}}$ across the benchmark datasets. We then extended our analysis by evaluating LEOPARD’s performance on individually masked test sets: $\mathcal{D}_{v=v_1, t=t_1}^{\text{test}}$, $\mathcal{D}_{v=v_1, t=t_2}^{\text{test}}$, and $\mathcal{D}_{v=v_2, t=t_1}^{\text{test}}$ from the KORA metabolomics dataset. This approach allows us to assess LEOPARD’s capability to arbitrarily complete any views at any timepoints within this dataset. Moreover, LEOPARD was evaluated using the expanded KORA metabolomics dataset to complete the masked test set $\mathcal{D}_{v=v_2, t=t_3}^{\text{test}}$, using the training data from $\mathcal{D}_{v=v_1, t=t_1}^{\text{train}}$, $\mathcal{D}_{v=v_2, t=t_1}^{\text{train}}$, and $\mathcal{D}_{v=v_1, t=t_3}^{\text{train}}$. This analysis further enables us to explore LEOPARD’s capability to complete views across a long time span. LEOPARD was trained using the same hyperparameters as we used in the previous experiments.

We have included a new paragraph in the Results section, detailed in lines 585 to 590. Below is a summary of our findings:

Our results reveal that some metabolites exhibit high PB values at $obsNum = 0$ (Extended Fig. 3) due to their low concentrations. While different completions show variability in their performances, PB generally decreases as $obsNum$ increases. This evaluation demonstrates that LEOPARD can transfer extracted temporal knowledge to different content representations in a flexible and generalized way. However, additional observations from the incomplete view may be necessary to ensure robust results, particularly for metabolites with low concentrations.

Extended Fig. 3. Performance of arbitrary temporal knowledge transfer.

The first three panels illustrate LEOPARD’s performance evaluated on $\mathcal{D}_{v=v_1, t=t_1}^{\text{test}}$, $\mathcal{D}_{v=v_1, t=t_2}^{\text{test}}$, and $\mathcal{D}_{v=v_2, t=t_1}^{\text{test}}$ of the KORA metabolomics dataset, with varying $obsNum$. The fourth panel displays LEOPARD’s performance on the extended KORA metabolomics dataset, where it completes the missing view $\mathcal{D}_{v=v_2, t=t_3}^{\text{test}}$ using the training data at t_1 . Each dot represents a PB value for a variable.

2.) The case studies are using two rather simple methods (linear regression, random forest) to establish some "biological information", however, the selected metabolites/proteins are "just" the ones which are significant/passed a threshold. They are not necessarily actual biological information. I would like to hear the authors' opinion on this minor aspect of the case studies. A major problem with the case studies is their potential bias because you selected just one dataset per method. I would like to see the linear regression also on the MGH and KORA multi-omics data, as well as the random forest on the MGH and KORA metabolomics. This would help establish the claims made by the authors that LEOPARD preserves biological signals in general and not just on the datasets selected for a specific method.

Reply: Thank you for your insightful comment, which has prompted further enhancements to our analysis. In this revision, we have expanded our case studies to include both the KORA metabolomics and multi-omics datasets, as well as the MGH COVID study for regression and classification tasks:

- 1) Updated the age-association analysis on the KORA metabolomics dataset by including pooled imputation from PMM and LM (see Fig. 6a).
- 2) Newly performed eGFR-association analysis on the KORA multi-omics dataset (see below for details and Fig. 6b).
- 3) Newly added regression analysis performed on each individual imputation of PMM and LM (see Figs. S3 and S4).
- 4) Newly performed analysis for CKD prediction performed on the KORA metabolomics dataset (see Fig. 7a, Extended Table 1); as well as updated the CKD prediction analysis on the KORA multi-omics dataset (see Fig. 7b, Extended Table 2).
- 5) Newly added CKD prediction performed on each individual imputation of PMM and LM (see Figs. S5 and S6).
- 6) Newly performed regression and classification case studies performed on the MGH COVID proteomics dataset (see Fig. RL 2).

Due to the restriction of the KORA Data Agreement, we cannot disclose new biological discoveries in this publication. Instead, this study investigates and replicates questions that have been previously explored with the KORA cohort. This approach allows us to use previous discoveries as references to validate the findings in our case studies.

We fully agree with your concerns that the significant metabolites/proteins reported in our case studies are the ones passing a threshold, which might not indicate true biological information. However, the agreement between the results from LEOPARD-imputed data and the observed data can validate if LEOPARD has preserved the signals within the data, though the signals may not represent true biological information. To enhance our analyses and in response to your valuable feedback, we further compared the significant metabolites/proteins detected in LEOPARD-imputed data with the findings from previous studies, which can demonstrate whether those signals are actual biological information.

To clarify the results of our new case studies, we would also like to summarize related enhancements that have been incorporated into our performance evaluation. Originally, MICE was configured to perform single imputation to align with the other imputation methods evaluated in this study, which are all single imputation methods. Following suggestions from other reviewers, we now evaluate MICE family methods under a multiple imputation setting. The imputation was performed five times for each method. We renamed

the method “MICE” in the original manuscript to “PMM” (predictive mean matching), the default algorithm used by MICE, and included the method “GLMM” (generalized linear mixed model) from the MICE family in our evaluation. When $obsNum = 0$, GLMM cannot be fitted as longitudinal information is completely missing. The method LM (linear model), also from the MICE family, is used in this case. This provides additional insights into how GLMM compares with a simpler LM method.

We have described the updated results for case studies in the main text, lines 476 to 536:

Regression analysis

The regression models are fitted using the observed data and different imputed data corresponding to $\mathcal{D}_{v=v_2, t=t_2}^{\text{test}}$. The performance of LEOPARD, cGAN, and missForest are evaluated on their imputed data directly, while two multiple imputation methods, PMM and LM, are evaluated by pooling their multiple estimates using Rubin’s rules.

We use the KORA metabolomics dataset to identify metabolites associated with age, controlling for sex. The models are fitted separately to each of the 36 metabolites ($N = 417$). Of the 18 metabolites significantly associated with age after a Bonferroni correction for multiple testing ($P < 0.05/36$) in the observed data, 17 are also significant in the data imputed by LEOPARD (see Fig. 6a). Among these 17 metabolites, several, including C14:1 (Tetradecenoylcarnitine), C18 (Octadecenoylcarnitine), C18:1 (Octadecenoylcarnitine), and Orn (Ornithine), have been validated by previous research¹⁻⁴ showing that they might be particularly relevant in aging and age-related metabolic conditions. In contrast, only one metabolite is significantly associated with age in the data imputed by missForest. No metabolite is identified as significant in the data imputed by cGAN, PMM, and LM. The results on each imputation of PMM and LM are shown in Fig. S3.

We then use the KORA multi-omics dataset to identify proteins associated with the estimated glomerular filtration rate (eGFR), controlling for age and sex. Each model is individually fitted to one of the 66 proteins from $\mathcal{D}_{v=v_2, t=t_2}^{\text{test}}$ ($N = 212$). In the observed data, 28 proteins are significantly associated with eGFR after a Bonferroni correction ($P < 0.05/66$). Of these 28 proteins, 10 proteins remain significant in the data imputed by LEOPARD (see Fig. 6b), while one is significant in the data from cGAN, and none is identified as significant in the data from missForest, PMM, and LM. Among the 10 proteins detected from the LEOPARD-imputed data, eight (TNFRSF9, IL10RB, CSF1, FGF21, HGF, IL10, CXCL9, IL12B) have been validated by prior research⁵. The results on each imputation of PMM and LM are shown in Fig. S4.

Fig. 6. Regression analyses with the data imputed by different methods.

a, Volcano plots display age-associated metabolites detected in the $\mathcal{D}_{v=v_2, t=t_2}^{\text{test}}$ and $\hat{\mathcal{D}}_{v=v_2, t=t_2}^{\text{test}}$ ($obsNum = 0$) of the KORA metabolomics dataset ($N = 417$). 18 significant metabolites ($P < 0.05/36$) identified in the observed data are shown in blue. Replicated metabolites from the data imputed by different methods are marked with labels. **b**, Volcano plots display eGFR-associated proteins detected in the $\mathcal{D}_{v=v_2, t=t_2}^{\text{test}}$ and $\hat{\mathcal{D}}_{v=v_2, t=t_2}^{\text{test}}$ ($obsNum = 0$) of the KORA multi-omics dataset ($N = 212$). 28 significant metabolites ($P < 0.05/66$) identified in the observed data are shown in blue. Replicated metabolites from the data imputed by different methods are marked with labels.

Fig. S3. The age-associate metabolites identified from each individual imputation of PMM and LM.

The eGFR-associated proteins identified from each individual imputation (Imputation 1 to 5) of methods PMM and LM. The evaluation is performed on $\mathcal{D}_{v=v_2, t=t_2}^{\text{test}}$ ($N = 417$) of the KORA metabolomics dataset, under $obsNum = 0$. 18 significant metabolites ($P < 0.05/36$) identified from the observed data are shown in blue. Replicated metabolites from the imputed data ($obsNum = 0$) are marked with labels.

Fig. S4. The eGFR-associate proteins identified from each individual imputation of PMM and LM.

The eGFR-associated proteins identified from each individual imputation (Imputation 1 to 5) of methods PMM and LM. The evaluation is performed on $\mathcal{D}_{v=v_2, t=t_2}^{\text{test}}$ ($N = 212$) of the KORA multi-omics dataset, under $obsNum = 0$. 28 significant metabolites ($P < 0.05/66$) identified from the observed data are shown in blue. Replicated metabolites from the imputed data ($obsNum = 0$) are marked with labels.

Classification analysis

We use balanced random forest (BRF)⁶ classifiers to predict chronic kidney disease (CKD) using the KORA metabolomics and multi-omics datasets. The classifiers are individually fitted using the observed and different imputed data of $\mathcal{D}_{v=v_2, t=t_2}^{\text{test}}$, corresponding to 36 metabolites from the KORA metabolomics dataset ($N = 416$, one sample removed due to a missing CKD label) and 66 proteins from the KORA multi-omics dataset ($N = 212$). CKD cases are defined as having an eGFR < 60 mL/min/1.73m²⁷. In the two datasets, 56 and 36 individuals are identified as CKD cases, respectively. We train the classifiers using identical hyperparameters and use leave-one-out-cross-validation (LOOCV) to evaluate their performance. The models for LEOPARD, cGAN, and missForest are trained using their respective imputed data, while the models for PMM and LM are trained on the average estimates across their multiple imputations (Methods).

For the KORA metabolomics dataset, the observed data obtain an F1 Score of 0.439, and the data imputed by LEOPARD achieves the closest performance with an F1 Score of 0.358 (Fig. 7 a, Extended Table 1). LEOPARD also outperforms its competitors in terms of accuracy, sensitivity, precision, AUROC (area under the receiver operating characteristic curve) and AUPRC (area under the precision-recall curve). The proteins from the KORA multi-omics dataset perform better than the metabolites from the metabolomics dataset for this task. The F1 Score increases to 0.544 for the observed data of the KORA multi-omics dataset. LEOPARD outperforms its competitors with an F1 Score of 0.403, an AUROC of 0.725, and an AUPRC of 0.435 (Fig. 7b, Extended Table 2). The prediction results on each individual imputation of PMM and LMM are displayed in Fig. S5 and S6.

Fig. 7. Classification analyses with the data imputed by different methods.

CKD classification evaluated using $\mathcal{D}_{v=v_2, t=t_2}^{\text{test}}$ and $\hat{\mathcal{D}}_{v=v_2, t=t_2}^{\text{test}}$ ($obsNum = 0$) from (a) the KORA metabolomics dataset ($N = 416$, $N_{\text{positive}} = 56$, $N_{\text{negative}} = 360$) and (b) the KORA multi-omics dataset ($N = 212$, $N_{\text{positive}} = 36$, $N_{\text{negative}} = 176$). Models are trained using the balanced random forest (BRF) algorithm with identical hyperparameters and evaluated using LOOCV. Evaluation metrics in the bar plot include true positive rate (TPR, also known as sensitivity), true negative rate (TNR, also known as specificity), positive predictive value (PPV, also known as precision), accuracy (ACC), and F1 score. The dashed lines in the ROC and PR curves represent the performance of a hypothetical model with no predictive capability.

Extended Table 1. Method performance of CKD prediction on KORA metabolomics dataset

Method	Sensitivity	Specificity	Precision	Balanced Accuracy	F1 Score	AUROC	PRRPC
groundtruth	0.643	0.800	0.333	0.721	0.439	0.831	0.474
PMM	0.125	0.897	0.159	0.511	0.140	0.543	0.186
LM	0.161	0.903	0.205	0.532	0.180	0.590	0.166
missForest	0.268	0.744	0.140	0.506	0.184	0.486	0.135
cGAN	0.268	0.789	0.165	0.528	0.204	0.555	0.177
LEOPARD	0.571	0.747	0.260	0.659	0.358	0.719	0.268

Bold indicates the highest performance among imputation methods.

**Fig. S5. CKD prediction using the KORA metabolomics dataset on each individual imputation of PMM and LM**

The performance of CKD prediction on the KORA metabolomics dataset using each of the 5 multiple imputations produced by methods PMM (a) and LM (b). Methods are evaluated on $\mathcal{D}_{v=v_2, t=t_2}^{\text{test}}$ of the KORA metabolomics dataset ($N = 416$, $N_{\text{positive}} = 56$, $N_{\text{negative}} = 360$), under $\text{obsNum} = 0$. The CKD prediction models are trained with each individual imputation of the 5 multiple imputations using the balanced random forest (BRF) algorithm with identical hyperparameters and evaluated using LOOCV. The barplot (left) shows multi-metric performance. The dashed lines in the ROC (middle) and PR (right) curves represent the performance of a hypothetical model with no predictive capability.

Extended Table 2. Method performance of CKD prediction on KORA multi-omics dataset

Method	Sensitivity	Specificity	Precision	Balanced Accuracy	F1 Score	AUROC	PRRPC
groundtruth	0.778	0.778	0.418	0.778	0.544	0.862	0.593
PMM	0.667	0.494	0.212	0.580	0.322	0.631	0.282
LM	0.611	0.562	0.222	0.587	0.326	0.665	0.339
missForest	0.583	0.562	0.214	0.573	0.313	0.589	0.230
cGAN	0.500	0.545	0.184	0.523	0.269	0.551	0.198
LEOPARD	0.667	0.665	0.289	0.666	0.403	0.725	0.435

Bold indicates the highest performance among imputation methods.

Fig. S6. CKD prediction using the KORA multi-omics dataset on each individual imputation of PMM and LM.

The performance of CKD prediction on the KORA multi-omics dataset using each of the 5 multiple imputations produced by methods PMM (a) and LM (b). Methods are evaluated on $\mathcal{D}_{v=v_2, t=t_2}^{\text{test}}$ ($N = 212$, $N_{\text{positive}} = 36$, $N_{\text{negative}} = 176$), under $\text{obsNum} = 0$. Models are trained using the balanced random forest (BRF) algorithm with identical hyperparameters and evaluated using LOOCV. The barplot (left) shows multi-metric performance. The dashed lines in the ROC (middle) and PR (right) curves represent the performance of a hypothetical model with no predictive capability.

Inspired by your feedback, we also performed regression and classification analyses on the MGH COVID dataset. This benchmark dataset has been reconstructed with 322 and 295 proteins selected from the Olink Explore 384 cardiometabolic ($v1$) and inflammation ($v2$) panels, respectively. We performed two analyses similar to those in the original study⁸ of this dataset: identifying proteins associated with neutralization level and predicting neutralization level, using both observed and imputed data corresponding to $\mathcal{D}_{v=v2, t=t2}^{\text{test}}$ ($N = 43$). As virus neutralization activity is highly correlated with inflammatory markers, both analyses showed promising results in the original paper.

For each protein, we fitted a logistic regression model using NPX values as the predictor and neutralization levels as the response. We applied a Bonferroni correction for multiple tests ($P < 0.05/295$). Surprisingly, no significant proteins were identified, even in the observed data (Fig. RL2a), which contradicts the results from the original study. We highlighted the 10 proteins with the lowest P -values in the volcano plots. We noticed the patterns in the data imputed by missForest were more similar to those in the observed data, compared with the data imputed by other methods.

For the classification task, the original study used Day 0 NPX values to predict Day 3 neutralization levels. In our evaluation, as the NPX values in the test set are from Day 3, we used Day 3 NPX values from both observed and imputed data to predict the same day's neutralization levels. We used the same settings as in our previous case studies to build BRF models. This task should theoretically be easier, as both predictors and outcomes are from the same day. However, while the original study achieved an AUC of 0.830, our model built on the observed data only obtained an AUC of 0.776 (Fig. RL2b). Moreover, we observed that the data imputed by missForest and PMM outperformed the ground truth in terms of accuracy and AUROC, which was unexpected and suggests potential biases with the experiments.

The discrepancies between the findings of our analysis and the original study may be due to the limited sample size of our analysis. The original study used the full dataset of 218 samples, while our analysis was performed on a test set with only 43 samples, which could greatly reduce the predictive power and give misleading results. As the case studies performed on the MGH COVID dataset yielded unusual results, we chose not to include them in the main manuscript to avoid potential misinterpretation.

We hope these additional comparisons and experiments can provide a more reliable evaluation and validate LEOPARD's capability in missing view completion.

Fig. RL 2. Case studies performed on the MGH COVID proteomics dataset.

a. Volcano plots show neutralization level-associated proteins detected from observed and imputed data ($obsNum = 0$) corresponding to $\mathcal{D}_{v=v_2, t=t_2}^{test}$ of the MGH COVID proteomics ($N = 43$). No significant proteins ($P < 0.05/295$) identified in the observed data and imputed data. The labels on the plots indicate the top 10 proteins with the lowest P -values. **b.** Neutralization level prediction evaluated on $\mathcal{D}_{v=v_2, t=t_2}^{test}$ and $\hat{\mathcal{D}}_{v=v_2, t=t_2}^{test}$ ($obsNum = 0$) from the MGH COVID proteomics dataset ($N = 43$, $N_{positive:HighLevel} = 30$, $N_{negative:LowLevel} = 13$). Models are trained using the balanced random forest (BRF) algorithm with identical hyperparameters and evaluated using LOOCV.

3.) While mice and missForest are some of the heavy hitters in imputation, they serve a different purpose than the one the authors are addressing - mice is built for multiple imputation, missForest is built for mixed-type data - both amongst other things. This raises the question, whether these two methods are the right benchmarking partners. This is also clear, when we are playing with obsNum=0, where these two cannot play along, as they are not able to infer mapping along time or in between modalities. In addition, cGAN and LEOPARD have multiple pages reserved on how their hyperparameters have been trained for the benchmark, while both mice and missForest get a sentence or two on how they are run more or less in default mode. Again, not sure whether we are comparing apples and oranges here. The authors should consider either selecting other methods for comparison, design a more sophisticated benchmarking setup (including cGAN, mice and missForest in their natural habitats and showing how LEOPARD outperforms them on their soil), or helping mice and missForest to be tweaked to the data in a way, they might be able to "understand" the temporal and multi-view setup of the data.

If the authors can address the above points in a satisfactory manner, I fully support the publication of this manuscript.

Reply: Thank you for your insightful observations regarding the choice of benchmarking partners for LEOPARD. To create a more sophisticated benchmarking setup and ensure that different methods can be evaluated in their natural habitats, we have included the following improvements in this revision:

- 1) As suggested by Reviewer #1, we added GLMM, a method tailored for longitudinal datasets, in our performance evaluation (see updated Figs. 3 and 4).
- 2) Imputations using multiple imputation methods were performed five times to leverage their advantage of uncertainty estimation (see updated Figs. 3-4 and 6-7, as well as newly added Figs. S1 to S6).
- 3) We explained the rationale for selecting cGAN, missForest, PMM, and GLMM, as benchmarking partners (see lines 320 to 331, and details below):

Due to the lack of established methods specifically designed for missing view completion in multi-timepoint omics datasets, we benchmarked LEOPARD against three widely recognized generic imputation methods: missForest, PMM (predictive mean matching), and GLMM (generalized linear mixed effect model), as well as a cGAN model designed for this study. The cGAN serves as a reference model to demonstrate how existing neural network approaches, typically suited for cross-sectional data, perform in longitudinal scenarios. MissForest, as a representative non-parametric method, was chosen for its robustness and ability to handle complex, non-linear relationships among variables. PMM and GLMM, both implemented within the MICE (Multivariate Imputation by Chained Equations)⁹ framework, represent established multiple imputation methods that not only address missing values but also allow for the assessment of imputation uncertainty. GLMM, with its ability to capture temporal patterns inherent in longitudinal data, is particularly advantageous for data imputation in longitudinal scenarios.

Currently, omics data are generally available at only a limited number of timepoints, which makes it difficult for most existing longitudinal imputation methods to capture temporal changes effectively. LEOPARD uses a different strategy by transferring temporal knowledge to content representation, without directly capturing temporal changes. This allows LEOPARD to be applied to datasets with data from only two timepoints, and to samples where only one view is available. To the best of our knowledge, LEOPARD

is the first method specifically developed for missing view completion in multi-timepoint omics data. By comparing LEOPARD with imputation methods that use various algorithms, we demonstrate that generic imputation methods cannot provide a robust solution for this emerging challenge, highlighting the necessity for tailored methods.

Regarding your concern about using the default settings for MICE and missForest, we appreciate the opportunity to clarify this matter. The default hyperparameters of established methods have been empirically tested to provide robust results across various datasets^{10,11}. In contrast, the cGAN and LEOPARD models designed in this study, which involve more hyperparameters for controlling their neural networks, lack prior knowledge of proper settings. This requires us to calibrate their hyperparameters. To ensure a fair comparison, we tuned the hyperparameters of cGAN and LEOPARD just once, using the KORA metabolomics dataset under $obsNum = 0$. We maintained these hyperparameters consistently across all scenarios, regardless of changes in datasets or settings such as $obsNum$ and $maskObs$. Additionally, we also exposed the same training data to LEOPARD, missForest, and the methods from MICE family. While methods like missForest, PMM, and LM cannot learn temporal information from samples measured at multiple timepoints, they can leverage additional data to calculate imputation errors in their model training iterations, which can further stabilize their imputation processes. These efforts are made to ensure our evaluation is rigorous and fair.

We greatly value your feedback and have made every effort in this revision to strengthen our analysis. We hope our evaluation strategies have provided a fair and informative assessment for each method.

Here are some minor points, I noticed while reviewing the manuscript:

- Why are you training LEOPARD for 600 epochs?

Reply: Thank you for your question, which has prompted us to clarify the model training process. When evaluating LEOPARD's performance on representation disentanglement, LEOPARD is trained until the contrastive losses computed on content and temporal representations stabilize, which occurs at around 450 epochs, as shown in Fig. 2a. To ensure full saturation of the training process, we extended the training process to 600 epochs, which also helps better visualize the separation of content and temporal embeddings (Fig. 2b and Fig. 2c). We have clarified this in the main text (lines 290 to 291):

In this analysis, the model is trained for 600 epochs to ensure that the contrastive loss stabilizes and reaches full saturation.

For the performance evaluation on the benchmark datasets, the number of training epochs was determined by the point at which PB values on the validation set stabilized (described in Methods, lines 924 to 926).

- What is a BRF model (line 625), please introduce acronyms which are not obvious.

Reply: Thank you for pointing this out. It should be "balanced random forest (BRF)". We have added clarification of this acronym in this revision.

- There is some rendering-related error due to using \$ in the plot.ipynb in [3]. Once remedied the code runs smoothly. The authors should address this such that the representation in github is directly usable.

Reply: Thank you for your valuable feedback regarding the script issues. This rendering error has been fixed. We have also updated the scripts to include the latest results and figures.

- Somehow the authors are not referring to Figure 3 in the Benchmarking section starting at line 246, while they do refer to Figure 4. While clear after some moments, it would facilitate the reading of this section, if references to figures are complete.

Reply: Thank you for highlighting this oversight. We have added references to the figures in this revision.

- The illustrations in Fig 1b are suboptimal, the one in Fig 1a is practically useless. Either revise these to support with the actual explanation you want to provide or expand on the text/legend to make it clear.

Reply: Thank you for the detailed suggestions regarding Fig. 1. During the development of LEOPARD, we had discussions with researchers in the fields of epidemiology and biomedicine, who are potential users of LEOPARD. We found that the concept of missing view is currently still primarily associated with the computational field. Therefore, we aim to retain Fig. 1a in the manuscript to help a broad scientific audience better understand the question this study addresses. To enhance the clarity of Fig. 1a, we have adjusted the text in the Introduction section (lines 73 to 74) accordingly:

Unlike missing data points that may be scattered across the entire dataset, a missing view refers to the complete missingness of all features from a certain view, as shown in Fig. 1a.

We have also revised the legend to support Fig. 1b:

An example of data density calculated from a variable in raw data (Timepoint 1 and Timepoint 2) and imputed data. The data density indicates a distribution shift across the two timepoints. Imputation methods developed for cross-sectional data cannot account for the temporal changes within the data, and their imputation models built with data from one timepoint, such as Timepoint 1, might not be appropriate for inferring data from another timepoint, such as Timepoint 2.

Thank you again for your positive feedback, valuable insights, and efforts in evaluating our code and scripts. We remain open to further suggestions to refine our work. Other modifications to the manuscript are highlighted in the text.

Reference

1. Chak, C. M. *et al.* Ageing Investigation Using Two-Time-Point Metabolomics Data from KORA and CARLA Studies. *Metabolites* **9**, (2019).
2. Darst, B. F., Kosciak, R. L., Hogan, K. J., Johnson, S. C. & Engelman, C. D. Longitudinal plasma metabolomics of aging and sex. *Aging (Albany. NY)*. **11**, 1262–1282 (2019).
3. Liu, W. *et al.* Metabolic Biomarkers of Aging and Aging-related Diseases in Chinese Middle-Aged and Elderly Men. *J. Nutr. Heal. aging* **22**, 1189–1197 (2018).

4. Pararasa, C. *et al.* Age-associated changes in long-chain fatty acid profile during healthy aging promote pro-inflammatory monocyte polarization via PPAR γ . *Aging Cell* **15**, 128–39 (2016).
5. Herder, C. *et al.* A Systemic Inflammatory Signature Reflecting Cross Talk Between Innate and Adaptive Immunity Is Associated With Incident Polyneuropathy: KORA F4/FF4 Study. *Diabetes* **67**, 2434–2442 (2018).
6. Pedregosa, F. *et al.* Scikit-learn: Machine Learning in Python. *J. Mach. Learn. Res.* **12**, 2825–2830 (2011).
7. Nano, J. *et al.* Novel biomarkers of inflammation, kidney function and chronic kidney disease in the general population. *Nephrol. Dial. Transplant.* **37**, 1916–1926 (2022).
8. Filbin, M. R. *et al.* Longitudinal proteomic analysis of severe COVID-19 reveals survival-associated signatures, tissue-specific cell death, and cell-cell interactions. *Cell reports. Med.* **2**, 100287 (2021).
9. Buuren, S. van & Groothuis-Oudshoorn, K. **mice** : Multivariate Imputation by Chained Equations in R. *J. Stat. Softw.* **45**, (2011).
10. Morris, T. P., White, I. R. & Royston, P. Tuning multiple imputation by predictive mean matching and local residual draws. *BMC Med. Res. Methodol.* **14**, 75 (2014).
11. Stekhoven, D. J. & Bühlmann, P. MissForest--non-parametric missing value imputation for mixed-type data. *Bioinformatics* **28**, 112–8 (2012).

Reviewer #4 (Remarks to the Author):

The authors present a new method, LEOPARD, for imputing missing data in a longitudinal data construct. In general, I found the manuscript to be well-written. Overall, the method may be promising. However, I have concerns how widely applicable the method can be, and in its current form I have concerns with the manuscript.

Reply: Thank you for all your valuable suggestions regarding the applicability of LEOPARD which have given us the opportunity to improve our manuscript. We have revised our manuscript and added experiments to address your thoughtful concerns.

While I understand that it might be a necessary assumption for some of the data scenarios, to make comparisons, the scenario of having all data at time point 1 is simply unrealistic, and the authors make this assumption with all datasets and offer no limitation statement in the Discussion.

Reply: Thank you for your insightful comments. LEOPARD is designed to complete missing views by transferring temporal knowledge to view-specific information. To complete $\mathcal{D}_{v=v2,t=t2}^{\text{test}}$, LEOPARD needs to extract temporal knowledge from $\mathcal{D}_{v=v1,t=t2}^{\text{train}}$ and content information from $\mathcal{D}_{v=v2,t=t1}^{\text{train}}$. This approach requires the data from these observed views to be complete.

We recognize that in real-world scenarios, data from observed views are very likely to contain missing values. Although many established methods can easily impute missing values, understanding LEOPARD's performance under this condition will greatly benefit this study. To address your concern, as well as major concern from Reviewer #1, we updated the architecture of LEOPARD to enable it to tolerate some missing values in the observed views. The details are below and in the manuscript from lines 891 to 894:

Any missing values in the observed view are encoded as the mean values across each specific variable, and these mean-encoded values are excluded from the computation of the reconstruction loss during back-propagation. This strategy enhances the robustness of LEOPARD in scenarios where input data contain missing values.

We explored how LEOPARD and its competitors perform in this scenario using the KORA metabolomics dataset (lines 368 to 371):

We simulate missing values by randomly masking 1%, 3%, 5%, 10%, and 20% of the data in the observed views (*maskObs*) under the assumption that data points are missing completely at random (MCAR). The experiment is repeated 10 times for each specified proportion, and the results are evaluated by PB and UMAP.

In this revision, we included a GLMM (generalized linear mixed effect model) method in the evaluation. The method previously referred to as MICE is now called PMM (predictive mean matching), which is the default method in MICE package. When *obsNum* is zero, data of *v2* at *t2* are assumed as completely missing, and GLMM cannot be trained due to limited longitudinal information. In this case, we trained a linear model (LM) to complete missing views. This can additionally evaluate how GLMM, a method widely used for longitudinal datasets, compares to a simpler linear model. PMM, LM, and GLMM are all multiple

imputation methods within the MICE framework. The results are shown below as well as in Extend Figs. 1 and 2, and main text from lines 459 to 472:

Our findings indicate that all the methods experienced an increase in PB when the observed views contain missing values (Extended Fig. 1). However, LEOPARD and missForest are robust to the missing data points in terms of PB. In contrast, cGAN and GLMM exhibit high sensitivity to missing values. Method cGAN does not show similar improvement with the increase of *obsNum* as it performs in Fig. 3 (middle row) and is gradually surpassed by missForest. GLMM overall exhibits higher PB than the other methods.

The UMAP plots (Extended Fig. 2) further demonstrate that LEOPARD’s performance remains comparable to scenarios with no missing data in the observed views (Fig. 4, second row), unlike the other methods which display overfitting or a great loss of data variation. Although LEOPARD outperforms other methods, we observed a change in the distribution of the imputed data (blue dots): as *maskObs* increases, these blue dots begin to shrink toward their center and become more concentrated. This leads to a reduced coverage

Extended Fig. 1. Performance of missing view completion when observed views have missing values.

The average PB are computed for $\mathcal{D}_{v=v_2, t=t_2}^{\text{test}}$ of the KORA metabolomics dataset across 10 repeated completions, under each proportion of masked data points in the observed views (*maskObs*). In each repetition, the data points are masked randomly. Each dot represents a PB value for a variable. Please note that LM is used instead of GLMM when *obsNum* = 0.

of the outer areas of the ground truth embeddings (green dots) and suggests that the imputed data might not capture the full variability of the data when the proportion of missing data is high.

This evaluation assumes missing values in the observed views are MCAR, which may introduce bias if the data points are missing at random (MAR) or missing not at random (MNAR). We summarized this limitation in the Discussion section from lines 654 to 663. We also discussed the issue of variation loss as *maskObs* increases:

Extended Fig. 2. UMAP visualizes the imputed data obtained when observed views have missing values.

UMAP models are initially fitted with the training data from the KORA multi-omics dataset (c, *t1*: S4, *t2*: F4). Subsequently, the trained models are applied to the corresponding observed data (represented by red and blue dots for *t1* and *t2*) and the data imputed by different methods (represented by green dots) under *obsNum* = 200 and varying *maskObs*. Please note that, for each *maskObs*, only the repetition that exhibits the lowest median of PB are visualized. The distributions of red and blue dots illustrate the variation across the two timepoints, while the similarity between the distributions of blue and green dots indicates the quality of the imputed data. A high degree of similarity suggests a strong resemblance between the imputed and observed data.

LEOPARD typically requires observed views to be complete so that temporal and content representations can be extracted. Considering the common occurrence of missing values in real-world omics data, LEOPARD is designed to tolerate a small proportion without compromising robustness. However, we observed that LEOPARD struggles to capture the full diversity present in the ground truth as *maskObs* increases to 20% (Extended Fig. 2). It is preferable for the input data for LEOPARD to contain less than 10% missing data points. Higher proportions of missing values are ideally addressed by generic imputation methods before processing with LEOPARD. Additionally, we assumed that the missing data were MCAR. Additional bias could be introduced if data points are missing at random (MAR) or missing not at random (MNAR) in real-world scenarios.

On line 178-179, the authors state “In the training phase, the generator of the cGAN is trained on $\mathcal{D}_{v=v1,t=t1}$ train and $\mathcal{D}_{v=v2,t=t1}$ train capture the mappings between two views.” and then say on Line 182 “In the inference phase, the generator utilizes the mappings it has learned from $t1$ to impute the missing view at $t2$.” This was extremely misleading to me, as it made it sound like cGAN was magically learning temporal dependence while only trained on the first time point. A clearer description of what is happening and the limitations with regards to temporal dependence is needed here since it comes right after your premise of longitudinal data.

Reply: Thank you for highlighting this ambiguity in our manuscript. After realizing the original description may imply the cGAN model can learn temporal information, we have rewritten this sentence to make it clearer and avoid ambiguity (lines 188 to 196 in the main text):

In the training phase, the generator of the cGAN is trained on observed data from the training set to capture the mappings between two views. ...In the inference phase, the generator utilizes the mappings it has learned from the observed data to impute the missing view in the test set. Compared to methods PMM and missForest, our cGAN model has the potential to learn more complex mappings between views. However, these three methods are not able to capture temporal dependence within longitudinal data and can only learn from samples where both views are observed.

Methods like missForest and cGAN are not designed for longitudinal datasets and, thus, are unable to capture temporal information within longitudinal data. As we also evaluated the methods under various *obsNum*, and the training data also include samples from $t2$ when *obsNum* is non-zero, in this revision we used “observed data from the training set” instead of $\mathcal{D}_{v=v1,t=t1}^{\text{train}}$ and $\mathcal{D}_{v=v2,t=t1}^{\text{train}}$ to have a more accurate description. We also explicitly state that cGAN, PMM, and missForest do not capture temporal information from longitudinal data. We believe this revised version better explains the imputation process of cGAN.

Perhaps the biggest issue to me is that the authors use datasets with sample sizes of $N = 218$, $N = 2,085$, $N = 1,062$. These types of sample sizes are not typical of most omics studies. There are handfuls of large consortiums that get hundreds of samples, but a majority of research in the field operates at total numbers of samples far smaller than these sample sizes. Further, the authors’ datasets utilized have very small numbers of features/biomolecules used relative to the sample size. Most omics studies will have more features than samples, even if you have hundreds of samples. I found myself questioning how LEOPARD

would hold-up with studies that are much less rich with samples throughout the manuscript. At an absolute minimum, the authors need to state this as a limitation, but to be truly valuable to the research community the authors should evaluate with how many samples their model can actually still perform well. What if I come in with 20 samples, or 40 samples, etc? Further, at least one of the datasets used should have a number of features that is more representative of real data, even if there isn't ground truth for all proteins.

Reply: Thank you for your valuable suggestion. To ensure our benchmark datasets cover a broad spectrum, we constructed our longitudinal datasets using data from different time spans, different panels, biochemical classes, and omics types. We fully agree that further incorporating a dataset in a wide format is essential for a comprehensive comparison and to effectively evaluate LEOPARD and other imputation methods.

In this study, the proteomics data in the MGH COVID dataset were measured using the Olink Explore 1536 platform, which comprises proteins from four Olink Explore 384 panels: cardiometabolic, inflammation, neurology, and oncology. In the original manuscript, we constructed the MGH COVID dataset by selecting proteins listed in the Olink Target 96 cardiometabolic and inflammation panels, resulting in 72 and 62 proteins for two views, respectively. This selection was made to facilitate performance comparison with the KORA multi-omics dataset, which also uses the Olink Target 96 platform for proteomics measurement. In this revision, we decided that it makes more sense to use proteins from the original Olink Explore 384 panels, and this MGH COVID dataset can complement the KORA multi-omics dataset. After the QC process (see Methods in the main text from lines 722 to 739), we selected 322 and 295 proteins from the Explore 384 cardiometabolic (*v1*) and inflammation (*v2*) panels to construct the two views. The training set of MGH COVID includes 140 samples, and this new dataset is used to evaluate the different methods in the scenario of more variables than samples. All analyses in the manuscript have been re-performed with this new dataset.

From the PB values (Fig. 3, upper row), we notice the performances of different methods are similar to those obtained from the original dataset (constructed with the Olink Target 96 panels): missForest overall achieves the lowest percent bias; LEOPARD yields similar performance to cGAN, and the performance gap between LEOPARD and missForest decreases with an increase in *obsNum*.

Additionally, we fully agree with your comments that evaluating how LEOPARD performs on datasets with limited samples will greatly benefit this study and provide valuable insights into LEOPARD's applicability. In this revision, we added a new section to evaluate the minimum number of training samples required for LEOPARD to achieve robust view completion. The evaluation is still performed on the three benchmark datasets, allowing direct comparison with the results obtained from using the full datasets. The evaluation strategy can be found below and in the manuscript from lines 563 to 568:

The evaluation is performed on $\mathcal{D}_{v=v2, t=t2}^{\text{test}}$ of our three benchmark datasets by varying the number of training samples from 20 to 160 and *obsNum* from 0 to 50. Each condition is tested 10 times with different samples randomly selected from the training sets. The performance is evaluated by PB averaged across these repetitions. Fig. 8 simplifies the boxplot and shows the median and the IQR of the averaged PB values calculated for the variables in the imputed data.

Fig. 3. PB of imputed results for the test sets of three benchmark datasets.

a-c, PB evaluated on $\mathcal{D}_{v=v_2, t=t_2}^{\text{test}}$ of the MGH COVID proteomics dataset (a), KORA metabolomics dataset (b), and KORA multi-omics dataset (c). Each dot represents a PB value for a variable. Please note that LM is used for imputation instead of GLMM when $obsNum = 0$.

With reduced training samples, this evaluation can also reflect how LEOPARD performs on datasets with more variable than samples. The results are summarized in Fig. 8 and in the manuscript (lines 569 to 580) as follows:

Across all datasets, average PB generally decreases with more training samples, indicating an improvement in view completion (Fig. 8). Consistent with our previous evaluation, PB values decrease as $obsNum$ increases. Additionally, we notice that the average PB steadily decreases for the MGH COVID proteomics dataset, which exhibits the smallest variation between the two timepoints in our UMAP plots (Fig. 4). In contrast, the average PB for the other two datasets shows some fluctuations, particularly for the KORA multi-omics dataset, which shows the most obvious variation between two timepoints. When $obsNum = 0$, the MGH COVID proteomics and the KORA metabolomics datasets require 120 training samples to obtain stable results; the KORA multi-omics dataset, however, exhibits a wide range of PB under this condition. When we increase $obsNum$ to 20, the performance stabilizes with approximately 60 to 80 samples used for training LEOPARD. Based on our evaluation, at least 80 training samples may be required for robust view completion.

We hope these new experiments can provide a more realistic representation of real-world scenarios and make our manuscript more relevant to the field.

Fig. 8. Evaluation of minimum number of training samples required for LEOPARD.

For each benchmark dataset, the average PB is evaluated on $\mathcal{D}_{v=v_2, t=t_2}^{\text{test}}$ across 10 repeated completions for each combination of training sample sizes and *obsNum*. The bar indicates the median and the IQR of the average PB values for different variables. In each repetition, the samples are selected randomly. Please note that the maximum *obsNum* cannot exceed the number of training samples, and the full training set of the MGH COVID proteomics dataset contains only 140 samples.

When comparing performance metrics such as PB, actual significance of model performance compared to other models should be done rather than just reporting the median value and stating it's lowest.

Reply: Thank you for your valuable suggestion, which helps make our methods comparison more informative. In line with your recommendation, as well as feedback from Reviewer #1, we have added significance levels in Fig. 3 to provide a clearer and more statistically robust performance evaluation.

Line 391: “We select the proteins based on the following QC criteria: (1) no missing or negative values; (2) at least 75% of 392 measured sample values are equal to or above the limits of detection (LOD)” – You went from 1,472 proteins down to 72 which in itself could be introducing bias into the results presented here. Similar measures were done in the other datasets as well. It should be noted in limitations that you are only using the most abundant proteins so any metrics of performance (from any model) may be inflated.

Reply: Thank you for your insightful suggestion. In the original manuscript, we selected proteins listed in the Olink Target 96 panels, each containing 92 proteins. After QC, 72 and 62 proteins were selected for the two views. For this revision, the MGH COVID dataset was updated to include 322 and 295 proteins selected from the Olink Explore 384 panels, with each panel originally including around 360 proteins.

We defined our QC criteria based on the procedures used in existing studies to ensure our experiments align with real-world settings. However, we recognize that removing less detectable variables could simplify the imputation and potentially skew our evaluation. We have addressed these considerations in the Discussion section from lines 649 to 654, where we outline the limitations of our study:

To align with real-world settings, we defined QC criteria based on existing studies when constructing our benchmark datasets, and consequently, only the most detectable proteins and metabolites were selected. This could inflate the metrics of both LEOPARD and other methods reported in this study. The performance on these selected variables may not accurately reflect that of the overall proteins and metabolites, especially those showing more variability in their abundance.

On lines 583-589, while I understand the additional advantage of LEOPARD being able to process observations where only one view is present, I do not understand why the authors chose to explicitly construct a different training scenario for LEOPARD compared to the other methods. It makes it impossible to disentangle differences in performance and what the true difference is results would be under the same training scenario.

Reply: Thank you for your concern. Unlike cGAN, missForest, and MICE-family methods, which impute missing values by learning direct mappings between variables, LEOPARD uses a different strategy that treats data of different views separately, not trying to find mappings, but to factorize the data into temporal and content representations. Therefore, LEOPARD needs to “see” all observed data ($D_{v=v1,t=t1}^{\text{train}}$, $D_{v=v1,t=t2}^{\text{train}}$, and $D_{v=v2,t=t1}^{\text{train}}$) to extract representations specific to each view and timepoints.

For missForest and MICE-family methods, although training samples with incomplete views are not directly used to build their imputation models, these samples are utilized to estimate the stopping criterion, which helps to tune and stabilize their model training. In this study, we aim to make the evaluation conditions as similar as possible wherever feasible. Recognizing the ambiguity in the original text, we have included more details to clarify how we trained and evaluated each method (in the Methods section, lines 987 to 1027):

All methods used only the data in the training sets to build imputation models. Their performance was evaluated on $D_{v=v2,t=t2}^{\text{test}}$. Different imputation methods may require specific data structures for input: cGAN and LEOPARD first build imputation models using training data, then apply the built models to test set to complete missing views. In contrast, the input data for other methods can be an incomplete matrix with missing values coded as NA. We adapted the input data accordingly to accommodate these specific requirements:

- Method cGAN only learns from samples where both views are present. Therefore, its training data only included training data from the first timepoint ($D_{v=v1,t=t1}^{\text{train}}$ and $D_{v=v2,t=t1}^{\text{train}}$) and data of different *obsNum* from the second timepoint ($D_{v=v1,t=t2}^{\text{train}}$ and $D_{v=v2,t=t2}^{\text{train}}$).
- LEOPARD can additionally learn from data where only one view is available. In addition to $D_{v=v1,t=t1}^{\text{train}}$ and $D_{v=v2,t=t1}^{\text{train}}$, the entire $D_{v=v1,t=t2}^{\text{train}}$ was included in its training. The variation of *obsNum* only affected the number of observed samples from $D_{v=v2,t=t2}^{\text{train}}$.
- The input data for missForest combined training data (including $D_{v=v1,t=t1}^{\text{train}}$, $D_{v=v2,t=t1}^{\text{train}}$, $D_{v=v1,t=t2}^{\text{train}}$, and data of different *obsNum* from $D_{v=v2,t=t2}^{\text{train}}$) and test data ($D_{v=v1,t=t2}^{\text{test}}$), with NA filling the masked data in the matrix.

- For the multiple imputation methods in MICE family, the input data were constructed with training data (identical to that used for missForest) and test data (including $D_{v=v1,t=t1}^{\text{test}}$, $D_{v=v2,t=t1}^{\text{test}}$ and $D_{v=v1,t=t2}^{\text{test}}$). To ensure test data remained unused for model training, a logical vector with TRUE assigned to test samples was passed to the *ignore* argument. Masked values were filled with NA in the matrix.

PB was selected as performance metric as it quantifies the relative deviation of imputed values from actual observations, offering a more straightforward interpretation compared to metrics like RMSE and MAE.

The cGAN model only supports paired data (where both views are available), thus the data where only one view is present are excluded from its input data. However, cGAN serves as a reference model and represents LEOPARD without the feature disentanglement and temporal knowledge transfer capabilities. The comparison between cGAN and LEOPARD and the ablation test on LEOPARD can demonstrate the improvement brought by LEOPARD’s unique features.

For missForest, PMM, and GLMM, the entire data in $D_{v=v1,t=t2}^{\text{train}}$ is included in the input data, identical to that for LEOPARD. Additionally, MICE-family methods (PMM and GLMM) have functionality to split training and test data (via argument *ignore*). Therefore, we also included $D_{v=v1,t=t1}^{\text{test}}$, $D_{v=v2,t=t1}^{\text{test}}$ into their input data. These test data are not included for missForest, which does not provide the function of train-test split, to ensure missForest does not use them to build the imputation model. In our evaluation, the same training data are used for training LEOPARD, missForest, and MICE-family methods. Due to their inherent algorithmic differences, the way they use the data can vary.

The authors chose to implement MICE with only a single imputation; this is nonsensical to me, as the algorithm is specifically designed to do multiple imputation and have those multiple imputations be leveraged.

Reply: Thank you for your valuable suggestions. In the original version, MICE was employed with its default algorithm (PMM), and the imputation was only performed once (single imputation). We were doing that as we attempted to apply the same evaluation condition for all methods, as missForest, cGAN, and LEOPARD are all single imputation methods. In this revision, the imputations for PMM, LM, and GLMM were performed five times with $m = 5$ to leverage their multiple imputation advantages. To compare with other single imputation methods, we calculate their PB values using this formula:

$$PB_i = \frac{1}{m} \sum_{imp=1}^m \text{median} \left(\frac{|\hat{x}_{(imp)}^i - x^i|}{x^i} \right),$$

where $\hat{x}_{(imp)}^i$ is the imputed value for the i -th variable from the imp -th imputation, while m is the number of imputations. The UMAP plotted the average of all estimates from their multiple imputations. In our case studies, results from multiple imputation methods were pooled with Rubin’s rule. The new evaluation reflects how multiple imputation methods compare to single imputation methods in the context of missing view completion for multiple timepoints omics data. Additionally, we also evaluated the performance of each individual imputation of the multiple imputation methods to assess their robustness and variability, as shown in Figs. S1 to S6.

Again, I understand that simplifications and choices needed to be made, but I think other scenarios and guidance for model users is lacking and any potential user of the model is going to be left with unanswered questions. For example, does LEOPARD work when you have some missing values for each sample and are not just in the complete view missing scenario? Do you have to have one complete time point of data for it to work? Is there some level of missingness where a sample is not viable to be in the dataset? A brief discussion of practical issues when using the model and a frank list of caveats/limitations, in addition to the already present list of advantages, is needed.

Reply: Thank you for your thoughtful feedback on LEOPARD's practical aspects. Based on our extensive new experiments in this revision, we have expanded the discussion to address the limitations and caveats of using LEOPARD, which can be found below and in the revision from lines 629 to 665:

Arbitrary style transfer, a concept from the computer vision field underpinning LEOPARD, allows the style of one image to be transferred to the content of another. This study demonstrates that LEOPARD inherits this capability and has the potential for arbitrary temporal knowledge transfer. Our experiments also demonstrate that LEOPARD can yield robust results with approximately 80 samples.

...

While LEOPARD demonstrates superior performance over existing generic imputation methods on missing view completion, it is important to consider the limitations and caveats of this study. To align with real-world settings, we defined QC criteria based on existing studies when constructing our benchmark datasets, and consequently, only the most detectable proteins and metabolites were selected. This could inflate the metrics of both LEOPARD and other methods reported in this study. The performance on these selected variables may not accurately reflect that of the overall proteins and metabolites, especially those showing more variability in their abundance. LEOPARD typically requires observed views to be complete so that temporal and content representations can be extracted. Considering the common occurrence of missing values in real-world omics data, LEOPARD is designed to tolerate a small proportion without compromising robustness. However, we observed that LEOPARD struggles to capture the full diversity present in the ground truth as *maskObs* increases to 20% (Extended Fig. 2). It is preferable for the input data for LEOPARD to contain less than 10% missing data points. Higher proportions of missing values are ideally addressed by generic imputation methods before processing with LEOPARD. Additionally, we assumed that the missing data were MCAR. Additional bias could be introduced if data points are missing at random (MAR) or missing not at random (MNAR) in real-world scenarios. Finally, our experiments were restricted by data availability to three timepoints, but in principle LEOPARD can accommodate additional timepoints and is well-suited for analyses involving multiple omics per timepoint.

Specifically, we have explored the scenarios where missing values may occur across samples. We have also clarified the threshold of missingness, and the minimum number of training samples required for LEOPARD to function effectively. These additions aim to provide potential users with clear guidance on the usage of LEOPARD. We hope these experiments and the revised text will comprehensively address your concerns and strengthen the study.

Minor edit:

Line 229: "UMPA" should be "UMAP"

Reply: Thank you for pointing out this error. It has been corrected. We have also thoroughly checked our manuscript for other potential typos.

Thank you again for helping us improve our research. We believe this revision addresses your concerns and provides a more comprehensive understanding of the theoretical and practical aspects of LEOPARD. Other modifications to the manuscript have been highlighted in the text.

REVIEWER COMMENTS

Reviewer #1 (Remarks to the Author):

Summary

The authors have made substantial improvements on their manuscript. It reads much better. Nonetheless, 2 out of the 4 major concerns raised in our previous review remained unaddressed and require major clarification. Below we list our previous concerns in italic and our remaining concerns in normal text.

Reply: Thank you very much for recognizing the substantial improvement in our revised manuscript and for taking the time to review it thoroughly. We appreciate your detailed feedback which are highly valuable and have helped us further refine our study. Below, we provide a point-by-point response to each of your concerns and comments.

Major Concerns

Below are major concerns on their manuscript.

1. The authors provide a solution to a problem that unlikely exists in real biological research. It is very rare that a data modal completely misses at a timepoint. It is more common that some samples miss some data at some timepoints. The authors completely ignore the more common scenario in their manuscript. If the authors think their approach is valuable, they should provide examples in which 1) a data modal is completely missed at a timepoint in published studies and 2) their approach provide new insights that are missing in the original studies.

The authors have addressed the issue of “some samples miss some data at some timepoints”. But the original concern remains valid:

Reply: Thank you for recognizing that we have addressed the issue of “some samples miss some data at some timepoints”. We apologize if we misunderstood your original concerns and appreciate this opportunity to address them further.

a) They cited 6 references in their lengthy response, including a review and five bioinformatics/technic papers. They only cited two of the six references in the revised manuscript and claimed on the two cited references: “However, the typically limited number of timepoints in current human cohorts can restrict the effectiveness of these longitudinal methods.” In other words, they have not established the need of LEOPARD in real, biological/clinical studies from their literature review in the manuscript.

Reply: Thank you for your detailed comments and for carefully evaluating the citations. In the 1st revision, we cited two representative studies focusing on method development, aligning with the scope of our study. For clarity, we have now included all six mentioned studies in this 2nd revision (see references 7 and 19-23).

We also appreciate the opportunity to further strengthen our claim that incomplete views are common in real-world biomedical and clinical studies by providing examples from a biological discovery context:

1) Incomplete data in real-world omics studies

In many omics-related studies, samples with incomplete phenotype or omics data are often excluded from analyses. While this practice is common, it is not always explicitly described. However, some studies detail their exclusion criteria:

- In Figure 1 of study¹, participants without adequate follow-up or metabolomics data were removed, yielding 2,059 samples for statistical analyses.
- In *Methods* section of study², the authors describe limiting their analyses to participants with complete datasets: “To compare the associations between BMI and host phenotypes across different omics, we limited the original cohort to the participants whose datasets contained (1) all main omic measurements (metabolomics, proteomics and clinical laboratory tests) from the same first blood draw”. After applying all criteria, 702 participants with two or more timepoints were included in their longitudinal analysis.

As samples with incomplete data are often excluded, studies with entirely missing modalities are rarely published, as there is no available data for downstream analyses.

2) Prevalence of incomplete data in multi-omics studies

While studies with completely missing modalities may not reach publication, reviews highlight the frequent occurrence of incomplete and missing omics profiles in real-world cohort studies:

- **Incomplete data in cross-sectional studies:**

“A more frequent and disruptive category of missing values, however, arises when a complete set of measurements is unavailable for some of the samples in the multi-omics dataset. This is typically encountered in cohort studies, in which not all individuals will have all omics data types collected, and thus the number patients with complete records can be a small fraction of the total dataset.”³

- **Practical constraints causing incomplete data:**

“Multi-omics is essentially open-ended, so it is not surprising that most papers, excluding the PGP-UK pilot study of ten individuals, did not present a ‘complete’ multi-omics dataset for all samples included in the study. Sample availability, budget limitations or simply experimental constraints alone, frequently result in datasets with missing data for some omics.”⁴

- **Longitudinal studies and incomplete/missing views:**

“Complete multi-omics profiles are often not available or feasible to get for all samples/study participants. For instance, the Alzheimer’s Disease Neuroimaging Initiative (ADNI) is a longitudinal, multicenter study launched in 2004...While large-scale metabolomics and lipidomics profiling is available for the study phases ADNI-1 and -GO/2, up to now proteomics profiling was only applied to a subset of ADNI-1 participants and gene expression profiling is only available for ADNI-GO/2.”⁵

These studies highlight the challenges of incomplete omics profile in real-world settings, especially in longitudinal studies.

Existing methods for handling missing data in multi-omics studies are primarily designed for cross-sectional datasets and are not well-suited for multi-timepoint data. This gap motivated the

development of LEOPARD, a method specifically designed for multi-timepoint omics data. We hope this additional context and the inclusion of relevant citations address your concerns. We are happy to provide further clarifications if needed.

b) So far the authors have shown potential values of LEOPARD in a very specific scenario: two timepoints, two views at each timepoint, one view completely missing at one timepoint, >80 samples in each timepoint. If this scenario is “increasingly common” as claimed, the authors should not have any problems citing 3 real, biological/clinical studies in the literature that fall under this scenario. While complete missing view is not uncommon in studies with many (far >2) timepoints, the authors unfortunately have not shown LEOPARD can handle data beyond two timepoints (see also the next concern).

Reply: Thank you for your detailed comment. Regarding your request for representative studies, in this 2nd revision, we have cited the previously mentioned examples that highlight the common occurrence of incomplete views in real-world biological and clinical studies (see references 5, 6, 8, and 10-11). We would like to further clarify the scenarios where LEOPARD is applicable:

1. **Datasets with more than two timepoints:**

In this revision, we extend our evaluation to a three-timepoint dataset. This demonstrates LEOPARD’s applicability to datasets with more than two timepoints (see our response to point 2 for details).

2. **Datasets with multiple views at each timepoint:**

In our study, $v1$ and $v2$ are used to distinguish complete views and incomplete views. Each of $v1$ and $v2$ can consist of a single view or a group of views. For example, in the KORA metabolomics dataset evaluation, $v1$ consists of one view (glycerophospholipids), while $v2$ consists of four views (acylcarnitines, sphingolipids, amino acids, and monosaccharide).

The evaluation in *Arbitrary temporal knowledge transfer* section demonstrates that LEOPARD can be trained on multiple views and can also impute multiple views. To avoid ambiguity, we have clarified this in the *Characterization of benchmark datasets* section (page 8, line 162 in the main text) and the *Problem formulation* section (page 28, line 632 in the main text) in this revision.

3. **$v2$ at specific timepoints can be incomplete or entirely missing:**

Our evaluation includes the most extreme scenario, where $v2$ is completely missing ($obsNum = 0$). By varying $obsNum$, we also demonstrate that LEOPARD achieves reliable imputation for partially incomplete views.

We hope this additional clarification and the expanded evaluation demonstrate LEOPARD’s flexibility and applicability to a wide range of scenarios, including datasets with multiple timepoints, multiple views, and both fully missing and partially incomplete views.

c) We believe that more users will be encouraged to use LEOPARD if the authors can show that new insights are obtained when LEOPARD is applied to a real-world study having an actual (not an artificially designated) missing view. The two case studies in the manuscript are good for

benchmarking but not for showcasing utility. If the authors prefer not to take this suggestion, they should at least state in “limitations” that LEOPARD has not been applied to real-world data so that users won’t have unrealistic expectations.

Reply: Thank you for your constructive feedback. We appreciate your suggestion to showcase LEOPARD's utility by applying it to a real-world study with an actual missing view. However, due to data agreement restrictions, we do not have access to additional data from the KORA cohort for use in this study. Furthermore, such datasets are rarely publicly available or included in published studies. To address your concern and set appropriate expectations, we have explicitly noted this in the *Discussion* section (page 26, line 573 in the main text):

While LEOPARD has been evaluated on real-world omics data with simulated missing views, it has not yet been applied to real-world studies with actual missing views.

This clarification ensures that users are fully informed of this limitation and prevent unrealistic expectations. We hope this addresses your concern and can provide transparency regarding LEOPARD's current scope and applications.

3. The authors claim that their method is valuable for longitudinal studies but only provide a framework and examples on pairwise studies. Is their method applicable to longitudinal studies with 3 or more timepoints? Can their method handle time as a continuous variable instead of a categorical variable?

This concern has not been addressed in the revised manuscript.

Reply: Thank you for highlighting this important concern and allowing us to expand LEOPARD’s evaluation using a dataset with three timepoints. Below are our specific responses to each part.

a) The authors have at least two datasets (the MGH COVID dataset and the KORA metabolomics dataset) on which they can demonstrate LEOPARD’s capability of handling data with three timepoints. Instead, they treated the three timepoints in the KORA metabolomics dataset in pairwise analyses (Extended Figure 3). Why?

Reply: In this revision, we have redesigned the experiments in the *Arbitrary Temporal Knowledge Transfer* section (page 24 in the main text) to incorporate evaluations with data spanning three timepoints. Specifically, in the updated results, we demonstrate LEOPARD’s ability to handle three-timepoint data using the fourth dataset, Extended KORA metabolomics dataset (Table 1 in the main text), with S4 as $t1$, F4 as $t2$, and FF4 as $t3$.

For this evaluation, we individually masked data at two of the three timepoints to test LEOPARD’s ability to complete views at earlier or later timepoints under various conditions:

- To complete $v2$ at $t1$, using data from $t3$ and $t2 + t3$ for training.
- To complete $v1$ at $t3$, using data from $t1$ and $t1 + t2$ for training.

The results (Extended Fig. 3, next page) show that percent bias (PB) generally decreases as the number of observed samples ($obsNum$) increases.

Extended Fig. 3. Performance of arbitrary temporal knowledge transfer.

LEOPARD is evaluated on $\mathcal{D}_{v=v_2, t=t_1}^{\text{test}}$ and $\mathcal{D}_{v=v_1, t=t_3}^{\text{test}}$ from the Extended KORA metabolomics dataset. Timepoints t_1 , t_2 , and t_3 correspond to the KORA S4, F4, and FF4 studies, respectively. For each completion, LEOPARD is trained with the data from the other view at the same timepoint ($\mathcal{D}_{v=v_1, t=t_1}^{\text{train}}$ for $\mathcal{D}_{v=v_2, t=t_1}^{\text{test}}$ and $\mathcal{D}_{v=v_2, t=t_3}^{\text{train}}$ for $\mathcal{D}_{v=v_1, t=t_3}^{\text{test}}$) along with varying $obsNum$ and the data from one or two additional timepoints. For the same completion task, the evaluation shows that PB can be lowered by increasing $obsNum$ or including additional timepoints into training. Each dot represents a PB value for a variable.

Moreover, we used UMAP to visualize the disentangled content and temporal representations obtained from the three-timepoint data (Extended Fig. 4, next page). These visualizations confirm that LEOPARD can process data spanning three or more timepoints, demonstrating its applicability beyond pairwise analyses.

We believe this three-timepoint evaluation has provided evidence of LEOPARD's capability to handle longitudinal datasets with multiple timepoints.

Extended Fig. 4. UMAP embeddings illustrate the representation disentanglement process of LEOPARD on the three-timepoint data from the Extended KORA metabolomics dataset.

a-b, UMAP embeddings of content (**a**) and temporal (**b**) representations at various training epochs for the Extended KORA metabolomics dataset's validation set. Representations encoded from data of $v1$ and $v2$ (shown as dark- and light-colored dots) at timepoints $t1$, $t2$, and $t3$ (depicted by blue, green, and red dots) are visualized. The data of $v1$ at $t3$ are imputed data produced at each training epoch, while the other data correspond to observed samples in the validation set. The plot shows clusters of similar representations factorized from different data sources.

b) The authors claim in their response: “data from additional timepoints (or views) can be accommodated by just expanding the one-hot embeddings, without modifications to the architecture.” This is not obvious. Using the approach they described, will the equivalence to Fig 2c show 2 temporal clusters or 3? We guess it can go either way. If it is the former, the approach fails.

Reply: Thank you for this question. For datasets with three timepoints, LEOPARD will extract three distinct temporal representations, each corresponding to a specific timepoint. In this revision, we have explicitly evaluated this using the extended KORA metabolomics dataset and visualized the temporal representations using UMAP (Extended Fig. 4 b). The results demonstrate that the temporal representations for each timepoint are well-separated, showing effective representation disentanglement process achieved by LEOPARD.

c) It is generally accepted that more data points allow for better imputation on missing data. Will LEOPARD obtain better percent bias (PB) values from three-timepoint analysis than those from pairwise analysis as in the last panel of Extended Figure 3?

Reply: Yes, incorporating data from an additional timepoint can indeed improve imputation performance. As shown in our updated results (Extended Fig. 3), adding data from an additional timepoint reduces PB for the imputations of both $v1$ and $v2$, especially when the number of observed samples ($obsNum$) is low (e.g., $obsNum = 0$). This improvement highlights the advantage of leveraging data from multiple timepoints to enhance the robustness of imputation, which is also consistent with the general understanding that more data points improve the reliability of missing data imputations. We have included this finding in the revised manuscript, emphasizing data from additional timepoints contribute to robust imputation performance (page 24 in the main text).

d) As a minor point, the authors should make a comment somewhere in the manuscript that LEOPARD cannot handle longitudinal data when time acts as a continuous variable.

Reply: Thank you for the suggestion. We have explicitly stated in the *Discussion* section (page 26, line 591 in the main text) that LEOPARD is not applicable to datasets where time is treated as a continuous variable:

LEOPARD is well-suited for cohort studies involving multiple omics with samples collected at shared discrete timepoints. However, as LEOPARD is not designed to learn varying temporal changes directly, it is not applicable to datasets with continuous or unaligned time intervals.
--

Additional Concerns

2. *Figure 3: I am not convinced that their method is better. Please add p values, comparing their method with others.*

The authors added p values to the plot, which is very helpful. Unfortunately, the results from different methods are too close to tell which method is better. Can the authors add the corresponding medians in the plot so that readers don't need to refer to the text to know which method is better? The same comments apply to Fig. S1, Extended Fig. 1 & Extended Fig. 3.

Fig. 3. PB of imputed results for the test sets of three benchmark datasets.

PB evaluated on $\mathcal{D}_{v=v_2, t=t_2}^{\text{test}}$ of the MGH COVID proteomics dataset (upper row), KORA metabolomics dataset (middle row), and KORA multi-omics dataset (lower row). Each dot represents a PB value for a variable. The value below each box represents the median. Please note that LM is used for imputation instead of GLMM when $obsNum = 0$. Significance level: not significant (ns), $P < 0.05$ (*), $P < 0.01$ (**), and $P < 0.001$ (***)

Reply: We appreciate the helpful suggestion. The P -values and medians have been added to Figures 3 (see above). The same changes have also been applied to Extended Figures 1 and 3, Supplementary Figures S1, S3, and S5. Please refer to the manuscript for the updated figures.

New Minor Points

It may be more thorough to provide the corresponding results of Extended Fig. 1 & 2 on the other 2 datasets.

Reply: We have extended our evaluations to the MGH COVID proteomics (Figs. S3 and S4 in the supplementary file) and KORA multi-omics datasets (Figs. S5 and S6 in the supplementary file). To ensure robustness, we applied random masking with $maskObs$ percentages of [1%, 3%, 5%, 10%, 20%] across four $obsNum$ scenarios. To account for the stochastic mechanism of the random masking, each experiment is repeated 10 times with different masked training data points, resulting in a total of 200 evaluations per method. However, due to the large number of variables in the

MGH COVID dataset, the computation of PMM and GLMM was very time-consuming, with an estimated runtime exceeding four months on our high-performance cluster (15 cores in parallel). As a result, we only managed to evaluate LEOPARD, cGAN, and missForest on this dataset. We believe the current results provide a reliable representation of performance under varying proportions of missing data.

Please describe in Fig. 6 whether the biomarkers have the same sign on “Estimate” from real & imputed data.

Reply: In this revision, we use solid and hollow dots to indicate whether the sign of the estimate from observed and imputed data matches or differs. In addition to Fig. 6 (see below), this update has also been applied to Figs. S7, S8, and S11 (see our 2nd revision).

Fig. 6. Regression analyses with the data imputed by different methods.

a, Volcano plots display age-associated metabolites detected in the $\mathcal{D}_{v=v_2, t=t_2}^{\text{test}}$ and $\hat{\mathcal{D}}_{v=v_2, t=t_2}^{\text{test}}$ ($obsNum = 0$) of the KORA metabolomics dataset ($N = 417$). 18 significant metabolites ($P < 0.05/36$) identified in the observed data are shown in blue. Replicated metabolites from the data imputed by different methods are marked with labels. Solid dots represent variables where the observed and imputed data have matching signs for the estimate, while hollow dots represent mismatched signs.

b, Volcano plots display eGFR-associated proteins detected in the $\mathcal{D}_{v=v_2, t=t_2}^{\text{test}}$ and $\hat{\mathcal{D}}_{v=v_2, t=t_2}^{\text{test}}$ ($obsNum = 0$) of the KORA multi-omics dataset ($N = 212$). 28 significant metabolites ($P < 0.05/66$) identified in the observed data are shown in blue. Replicated metabolites from the data imputed by different methods are marked with labels. Solid dots indicate sign matches between the observed and imputed data, while hollow dots indicate mismatches.

It may be more informative to combine the two rows in Fig. 8. Change the panel arrangement from 2x3 to 3x1 if needed. Make a comment in Discussion that the minimum sample size likely depends on data characteristics.

Reply: Thank you for your valuable suggestion. For Fig.8, we initially created two versions with panel arrangements of 2x3 and 3x1. Based on your input, we agree that the revised arrangement improves clearly. In this 2nd revision, we have also updated the tringles for the six conditions by replacing them with square and circle, varied further by using unfilled and filled shapes in addition to color distinctions (see updated Fig. 8 below).

We have also addressed the comment regarding the minimum sample size in the *Discussion* section (page 25, line 563 in the main text):

While the minimum sample size for robust performance depends on specific data characteristics, our experiments demonstrate that LEOPARD yields robust results with approximately 60 to 80 samples on the benchmark datasets used in this study.

The legend of Extended Fig. 2 is truncated.

Reply: We apologize for this oversight and thank you for bringing it to our attention. The legend of Extended Fig. 2 has been corrected in this revision (may need to zoom in to notice the difference).

We sincerely appreciate your constructive feedback and insights. We hope that our revisions have addressed your concerns. Additional modifications to the manuscript have been highlighted in the text for your convenience.

Reference

1. Li, Y. *et al.* Metabolomic Profiles, Ideal Cardiovascular Health, and Risk of Heart Failure and Atrial Fibrillation: Insights From the Framingham Heart Study. *J. Am. Heart Assoc.* **12**, (2023).

2. Watanabe, K. *et al.* Multiomic signatures of body mass index identify heterogeneous health phenotypes and responses to a lifestyle intervention. *Nat. Med.* **29**, 996–1008 (2023).
3. Tarazona, S., Arzalluz-Luque, A. & Conesa, A. Undisclosed, unmet and neglected challenges in multi-omics studies. *Nat. Comput. Sci.* **1**, 395–402 (2021).
4. Conesa, A. & Beck, S. Making multi-omics data accessible to researchers. *Sci. Data* **6**, 251 (2019).
5. Wörheide, M. A., Krumsiek, J., Kastenmüller, G. & Arnold, M. Multi-omics integration in biomedical research – A metabolomics-centric review. *Anal. Chim. Acta* **1141**, 144–162 (2021).
6. Filbin, M. R. *et al.* Longitudinal proteomic analysis of severe COVID-19 reveals survival-associated signatures, tissue-specific cell death, and cell-cell interactions. *Cell reports. Med.* **2**, 100287 (2021).

Reviewer #2 (Remarks to the Author):

Reply: Thank you for your comment and for participating in the review process. Please refer to our detailed responses provided to Reviewer #1.

Reviewer #3 (Remarks to the Author):

The authors have comprehensively changed and improved their manuscript. They have also managed to dispel most of my concerns, but not all of them.

When in my first review, I was referring to methods not being able to "play along", I was also meaning that you could in theory build an additional script around a method like missForest making it capable of handling a temporal setting. Similarly, this should be possible for GLMM. However, I do find the expansion of the number and types of benchmarking methods convincing, as well as the reasoning that even if not any of them is specifically intended to address the missing time point/view setting, they represent a lay of the land of imputation methods.

However, I would like the authors to reconsider not adding the COVID results - even if it remains unclear how some of the methods achieve better than groundtruth results (probably due to the fact that single imputations tend to underestimate the original variation in the data) - these are results that due show the difficulty of imputation (and reproducing scientific work, fwiw). My decision towards accepting this manuscript review is not depending on this.

Reply: Thank you very much for your valuable suggestion. We agree that including the results from the published MGH COVID dataset can provide additional insights into imputation evaluation and support reproducibility. Furthermore, as noted by Reviewers #1 & 2, including examples from published studies is highly encouraged.

In this 2nd revision, we have added a new section to report our findings on this dataset (page 20 in the main text):

Case studies on the MGH COVID proteomics dataset We additionally use the observed and imputed data corresponding to $\mathcal{D}_{v=v_2, t=t_2}^{\text{test}}$ ($N = 43$) of the MGH COVID dataset to identify proteins associated with neutralization levels and to predict neutralization levels. These analyses are adapted from those conducted in the original study of this dataset⁶.

For each protein, we fit a logistic regression model using the proteomics data as the predictor and neutralization levels as the response. Bonferroni correction is used for multiple tests ($P < 0.05/295$). Surprisingly, no proteins reach statistical significance even in the observed data (Fig. S11 a), which contradicts the results of the original study. The 10 proteins with the lowest P -values are highlighted in the volcano plots. We observe that patterns in the missForest-imputed data are more similar to those in the observed data compared to data imputed by other methods.

For the prediction of neutralization level, we use the proteomics data from Day 3 to predict the same day's neutralization levels. We build BRF models using the same settings as in our previous

experiments. The model built on the observed data achieved an AUROC of 0.776 (Fig. S11 b), while the models trained on missForest- and PMM-imputed data outperform the observed data and achieve AUROC of 0.788 and 0.779, respectively.

The discrepancies between our findings and those of the original study might result from the limited sample size of our analysis. While the original study analyzed the complete dataset of 218 samples, our evaluation was restricted to a test set of only 43 samples. This substantial reduction in sample size can limit the statistical power, leading to non-significant results and inconsistent findings. Additionally, single imputations tend to underestimate the original variation in the data, potentially resulting in better predictive performance than observed data.

These findings highlight the challenges associated with performing and evaluating imputation, as well as the potential biases introduced by imputation methods. While these results should be interpreted with caution and may not reflect underlying biological reality, we include them to emphasize the complexities of imputation and to support reproducibility in scientific research.

Minor comment:

- please choose a different name than groundtruth, it is misleading

Reply: In this revision, we have replaced the term “groundtruth” with “observed” throughout the manuscript.

- in methods comparisons you perform Wilcoxon tests, please adjust test for multiple testing using an appropriate method (by the way, in this part of your study a Bonferroni correction would also not be inappropriate) and ensure that your interpretation of the results are updated accordingly

Reply: Thank you for this important observation. We have applied Bonferroni correction for Wilcoxon tests on methods comparison and updated the corresponding interpretation accordingly. Please refer to our updated script for details.

- in figures with/containing box plots (especially when illustrating PB) consider using a log scale on the y-axis, to make the box plots more "discernible"

Reply: Thank you for the suggestion. We have applied log scale transformations to the y-axis in the relevant box plots to improve clarity. Additionally, as suggested by another reviewer, we have included the medians for each box in these plots for further clarity. These changes are also included in our updated script.

I support the publication of this manuscript on the condition that the results are fully reproducible (see my remarks on code review) and my above comments have been addressed.

The code is well written and uses established tools of ensuring reproducibility. However, there are still some issues, when it comes to reproducing the results. The authors will need to address these before the manuscript is published to ensure that their results are indeed reproducible.

- some data objects seem to have wrong names (e.g. ``/data/MGH/COVID_imputed/obsNum = 50/LEOPARD.csv``, iso ``../data_LEOPARD.csv``)
- avoid using whitespaces in filenames
- it still requires some hacking to get all going, the documentation should include a link to a jupyter instance to allow for a straightforward running of the method (something that was actually possible in my first review but seemingly not available anymore or I did not find it)

Reply: Thank you for reviewing the code and checking its reproducibility. In the previous version, one file was incorrectly named, which prevented the reproduction of some figures. We have corrected this issue and updated the figures in the Jupyter notebook. To enhance the reproducibility of our script, we have integrated our Jupyter notebook with Google Colab, and users can now directly reproduce our figures on Google Colab (<https://colab.research.google.com/github/Han-Siyu/LEOPARD/blob/main/plot.ipynb>).

We greatly value your comprehensive feedback and insights. We have made revisions in the hope of addressing your concerns. Additional changes to the manuscript have been highlighted in the text.

Reviewer #4 (Remarks to the Author):

The authors have addressed many of the reviewers' comments, and the quality of the manuscript has been improved. However, I still have issues with several of the choices made.

1. Per several reviewers' comments, which were glossed over by the authors by stating "we did not focus on missing value imputation, as existing methods already address this effectively". This simply isn't true (at least in fields like untargeted proteomics and transcriptomics) and simulating data, as the authors did, with values going missing completely at random (MCAR) is not reflective of the mixed missing value mechanisms in several omics, such as untargeted proteomics.

Reply: Thank you for raising this important point. We agree that existing methods may not address missing value imputation effectively in all scenarios. However, these methods have provided valuable solutions for many omics imputation tasks. Despite the availability of various methods for missing data imputation, missing view completion in a multi-timepoint omics context remains a largely unexplored area. Therefore, our study focuses specially on this area and aims to introduce one of the first methods tailored to address this challenge.

From our literature review, we observed that few studies evaluated the impact of missing values in observed views on missing view completion. Many methods also require the observed views to be complete. For example, MI-MFA¹ employs SVD, which requires training data to be fully observed. Considering that missing data points are very prevalent in omics datasets, we designed LEOPARD to tolerate a small proportion of missing values in the observed view. Simulating these missing values as MCAR, while simplifying the case, provided a baseline scenario to evaluate whether LEOPARD can effectively complete missing views when missing values also exist. Our experiments demonstrate that LEOPARD performs robustly in this baseline scenario when the training data contains less than 10% missing values. We recognize that simulating missing data as MCAR simplifies real-world scenarios, and we have acknowledged this limitation in the *Discussion* section (page 26, line 587 in the main text):

Additionally, we assumed that the missing data were MCAR. Additional bias could be introduced if data points are missing at random (MAR) or missing not at random (MNAR) in real-world scenarios.

This highlights the potential biases introduced by mixed missing mechanisms and suggests the need for future work to address this issue more comprehensively.

2. The use of dimension reduction to assess performance is of limited utility. Because it is based on the entire omics profile there are multiple ways to arrive at similar looking plots.

Reply: Thank you for raising this potential issue. It is indeed true that dimension reduction can lead to similar-looking plots when transformations are applied to the entire dataset. This is particularly likely when applying UMAP directly on observed and imputed data. To mitigate this issue, we trained the UMAP model only with observed training data, then applied it to unseen observed and imputed data from the test set (*Methods* section, page 37 line 935 in the main text). This strategy has several advantages which contribute to reduce the possibility of obtaining similar-looking embedding plots from dissimilar datasets:

- The dimensionality reduction space is defined by the observed data, which serves as a fixed reference for evaluating imputed datasets. The UMAP model is not influenced by data imputed by different methods.
- As the observed and different imputed data are transformed in the same way, the structural similarities or discrepancies of their embeddings are more directly attributed to the data, rather than to different dimension reduction processes.
- The training sets have larger sample sizes, which is beneficial for model generalization, providing a more robust evaluation of how well different imputed data match observed data.

To address this valid point and ensure clarity, we further clarified this in the *Methods* section (page 38, line 938 in the main text):

Training UMAP models only with training data can improve the model's generalization and make it serve as a fixed reference. As observed data and different imputed data from test sets are transformed into embeddings in the same way, the structural similarities or discrepancies of their embeddings are more directly related to the data, rather than to variations of different dimension reduction processes. This approach reduces the possibility of obtaining similar-looking embedding plots from dissimilar datasets and guarantees robust evaluations.

Despite our efforts, we acknowledge that all dimensionality reduction techniques inherently have the possibility of producing similar-looking plots from different datasets. To address this, we used percent bias and several case studies to complement our evaluation. The results from these comprehensive assessments will help to evaluate different methods more effectively.

3. In their response to Reviewer 1, the authors claim that they use imputation methods like MICE, missForest, KNNimpute because methods for handling complete missing views in longitudinal studies don't exist. While they did include one new method, all of the other methods presented are designed for single datasets and don't really take advantage of the multi-omics data structure, creating an inherently unfair comparison. This is pointed out by multiple reviewers. Why not use Multiple Imputation Multiple Factor Analysis (MI-MFA) by Voillet et al, 2016, or Late Fusion Incomplete View Multi-View Clustering (LF-IMVC) by Liu et al 2019, or any other of the multiple methods for multi-view data?

Reply: Thank you for your insightful suggestions and for proposing additional methods for our consideration. We appreciate the opportunity to further discuss our selection of benchmark methods. In our evaluation, we aim to benchmark LEOPARD against representative imputation methods. This includes both missing data imputation methods, including PMM and missForest, and a missing view imputation method, cGAN. The cGAN model, adapted from View Imputation via Generative Adversarial Networks² and Deep Adversarial Learning for Multi-Modality Missing Data Completion³, is specifically designed to impute missing views within a multi-omics context (page 8, line 173 in the main text).

Regarding the methods you suggested, LF-IMVC is primarily designed for incomplete multi-view clustering, where it uses kernel techniques to impute clustering matrix. It does not explicitly complete missing views, which does not align with the scope of our study. Additionally, we

attempted to evaluate MI-MFA in response to your comment; however, computational challenges prevented it from being applied on our benchmark dataset. After a detailed inspection of their scripts (<https://github.com/GonzalezIgnacio/missRows/blob/master/R/MIMFA.R>, line 135), we found it creates possible imputations by calling $\text{permutation}(n, r)$ to construct a matrix with $\frac{n!}{(n-r)!}$ columns and r rows, where n is the number of observed samples selected as donors, and r is the number of incomplete samples. Even for the MGH COVID proteomics dataset, our smallest benchmark dataset, this matrix would have approximately 4.769×10^{235} columns, making it computationally unmanageable. In contrast, the original study of MI-MFA evaluated the method on a demo dataset with 60 samples, categorized into nine groups, each containing no more than 10 rows. As the donors are only selected within each group, the computation of $\frac{n!}{(n-r)!}$ on that demo dataset is feasible.

Similar to cGAN, MI-MFA is designed for cross-sectional datasets and does not account for temporal information in multi-timepoint datasets. In response to the comments from Reviewers #1 and #3 regarding including benchmark methods suitable for longitudinal datasets to ensure a fair comparison, we included GLMM in our 1st revision. Although GLMM is not intended for missing view, it can capture temporal changes within longitudinal data. Meanwhile, cGAN is tailored for multi-view context but lacks the capability to capture temporal knowledge. Our evaluation suggests that these methods may not be well-suited for missing view completion in multi-timepoint context, especially when data across timepoints exhibit strong variation. This highlights the need for a method capable of capturing temporal information while accommodating multiple views.

4. The authors were asked “The authors claim that their method is valuable for longitudinal studies but only provide a framework and examples on pairwise studies. Is their method applicable to longitudinal studies with 3 or more timepoints? Can their method handle time as a continuous variable instead of a categorical variable?” While I understand that there may not be readily available datasets for a real-world application, they also have a number of simulated datasets in the manuscript and did not attempt to simulate anything to address these questions.

Reply: Thank you for your comments. In this revision, we have added extra evaluations to demonstrate LEOPARD’s performance on a dataset with three timepoints. Please refer to the *Arbitrary temporal knowledge transfer* section (page 24 in the main text) for the detailed results.

5. In response to Reviewer 4, while the authors make some concessions to extend the number of features to as many as 322, this is still far smaller than a typical proteomics or transcriptomics dataset, as two examples. In practice, there is nothing in this manuscript to give me faith that these methods would withstand the more typical cases in biological data.

Reply: Thank you for raising this important concern. LEOPARD leverages representation disentanglement and style transfer techniques from the computer vision field. These techniques have demonstrated success in handling high-resolution 4K images that involve millions of features^{4,5}. However, due to data availability, we do not have transcriptomics or untargeted omics datasets across multiple timepoints to evaluate LEOPARD on such scales. In response to Reviewer #4, we evaluated a scenario where the dataset contains more variables than samples, a common characteristic of datasets with a large number of variables. This evaluation provides some insights

into LEOPARD's potential to handle larger, high-dimensional datasets. To ensure transparency, we have included this limitation in the manuscript (page 26, line 593 in the main text):

Due to data availability, this study does not include evaluations on large-scale transcriptomics or untargeted omics datasets with thousands of variables across multiple timepoints.

6. Further, the question about minimal sample sizes was answered, but it showed a required sample size of 60 to 80 which, in practice, is rarely a luxury that researchers have. I believe that this makes this method much less compelling to a wide range audience.

Reply: Thank you for highlighting this consideration. LEOPARD is a neural network-based method, and like most neural network approaches, it requires a larger number of samples to capture complex, non-linear data patterns and produce generalizable results. Furthermore, missing view completion tasks typically involve a much higher proportion of missing data compared to traditional missing data point imputation, which requires more samples to effectively learn the distribution of the entire view.

While a sample size of 60 to 80 may not be feasible for all experiments, especially smaller-scale studies, our experiments show that LEOPARD can impute incomplete views for hundreds of samples using only 60 ~ 80 observed samples with 10 in the incomplete data block. This suggests that, given sufficient observed samples, LEOPARD can complete datasets with a much larger number of incomplete samples.

We are continuously working to improve LEOPARD's applicability and scalability. With rapid advancements in computational modeling and growing interest in missing view completion, we also anticipate that LEOPARD will inspire new methods and approaches suitable for a broader range of scenarios.

Thank you again for your constructive feedback and valuable suggestions, which have greatly helped us enhance the clarity and robustness of our study. We hope our revisions have addressed your concerns. Additional modifications to the manuscript have been highlighted in the text.

Reference

1. Voillet, V., Besse, P., Liaubet, L., San Cristobal, M. & González, I. Handling missing rows in multi-omics data integration: multiple imputation in multiple factor analysis framework. *BMC Bioinformatics* **17**, 402 (2016).
2. Shang, C. *et al.* VIGAN: Missing View Imputation with Generative Adversarial Networks. *IEEE Int. Conf. Big Data* **2017**, 766–775 (2017).
3. Cai, L., Wang, Z., Gao, H., Shen, D. & Ji, S. Deep Adversarial Learning for Multi-Modality Missing Data Completion. in *Proceedings of the 24th ACM SIGKDD International Conference on Knowledge Discovery & Data Mining* 1158–1166 (ACM, New York, NY, USA, 2018). doi:10.1145/3219819.3219963.
4. Wang, Z. *et al.* MicroAST: Towards Super-fast Ultra-Resolution Arbitrary Style Transfer. *Proc. AAAI Conf. Artif. Intell.* **37**, 2742–2750 (2023).
5. Chen, Z., Wang, W., Xie, E., Lu, T. & Luo, P. Towards Ultra-Resolution Neural Style Transfer via Thumbnail Instance Normalization. *Proc. AAAI Conf. Artif. Intell.* **36**, 393–400 (2022).

REVIEWERS' COMMENTS

Reviewer #1 (Remarks to the Author):

We thank the authors for their additional work and efforts to improve their manuscript. The authors have either done additional work to address our concerns or admitted limitations in Discussion. While we strongly believe that it is a major weakness of the manuscript not to demonstrate the value of LEOPARD in real-world OMICS studies, it is time for a final decision by the Editor.

Reply: Thank you for acknowledging our efforts and providing constructive feedback throughout the review process. We value the guidance you have provided and appreciate your insights, which have greatly helped in refining our manuscript.

Minor Comments for improvement.

Figure 6:

The use of hollow points to show sign similarity is difficult to highlight global behavior. A better representation of showing whether the sign (effect size) is correlated between the imputed and the real dataset is to do a scatter plot of the effect size per dataset on each axis. Then you can quantify the correlation and the % of features with the same directionality.

Reply: Thank you for this excellent suggestion to enhance our performance evaluation. In this revision, we have computed the correlation of the effect sizes and the percentage of variables with the same directionality. While data agreement restricts us from disclosing biological discoveries from the real KORA datasets, such as displaying exact effect sizes in a scatter plot, we have thoroughly detailed the relevant evaluation results in the main text (lines 343-346, 349-355, and 365-370):

To assess the robustness of the imputation methods in preserving original data characteristics, we evaluate the consistency of the effect signs, the Spearman correlation of effect sizes, and the agreement of significant variables between imputed and observed data.

...

We use the KORA metabolomics dataset to identify metabolites associated with age, controlling for sex. The models are fitted separately to each of the 36 metabolites ($N = 417$). LEOPARD and cGAN each demonstrate 88.9% of metabolites with matching effect signs, followed by LM 75.0%, and both missForest and PMM 69.4%. The Spearman correlation between effect sizes from the observed and imputed data also varies across methods, with LEOPARD showing the highest correlation of 0.708, followed by PMM (0.440), cGAN (0.319), and GLMM (0.110). In contrast, missForest shows a negative correlation of -0.074.

...

We then use the KORA multi-omics dataset to identify proteins associated with the estimated glomerular filtration rate (eGFR), controlling for age and sex. Each model is individually fitted to one of the 66 proteins ($N = 212$). The percentages of proteins with matching effect signs across different methods are as follows: LEOPARD demonstrates 92.4%, missForest 87.9%, cGAN 74.2%, PMM 68.2%, and LM 57.6%. LEOPARD obtains the Spearman correlation score of 0.539, followed by LM (0.473), PMM (0.421), missForest (0.317), and cGAN (0.277).

Complementing to our original evaluation focusing on P -value, we believe these new results provide additional insights from the perspective of effect size.

Figure 3 / Results (lines 314-318):

The IQR is not a very descriptive statistic on its own when describing PB because it provides information on ‘consistency’ rather than accuracy. For example, if a method has a high, yet consistent bias, it will still achieve a low IQR (albeit being consistently wrong). We recommend the authors either remove the IQR or provide the median PB in tandem to showcase both stability (IQR) and accuracy (median PB) for each method.

Reply: Thank you for your insightful suggestions. It is true that the IQR reflects stability but does not indicate accuracy. In this revision, we have included median PB in the main text, as suggested, to provide a more comprehensive evaluation (lines 282-286 in the main text):

When obsNum is 0, missForest displays the largest IQR of 0.186 with a median of 0.205, though it is not significantly different from LEOPARD after Bonferroni adjustment (Fig. 3, middle row). LEOPARD achieves the smallest IQR of 0.094 with a median of 0.209 under the same condition, while cGAN, PMM, and LM obtain IQR values of 0.125, 0.132, and 0.166, with corresponding medians of 0.229, 0.253, and 0.254, respectively.

Again, we truly appreciate your insights and effort, which have helped us improve our work. We are grateful for your contributions to enhancing our study.

Reviewer #2 (Remarks to the Author):

Reply: Thank you for your comment and for participating in the review process. Please refer to our detailed responses provided to Reviewer #1.

Reviewer #3 (Remarks to the Author):

I thank the authors for further addressing my comments. The manuscript has matured and reached a state where it presents a nice method for a specific task in a very comprehensive way. While some comments from other reviewers will still need some attention - among which was an earlier comment also raised by me regarding the ability to compare LEOPARD with other methods essentially doing something else - I stand by my previous assessment that for an assessment of the performance of LEOPARD, the chosen methods and the discussions thereof are sufficient.

Adding the MGH Covid data set has further improved the manuscript. Thank you.

Regarding reproducibility: I was still able to reproduce your plots and results using the github repository - well done (one cannot point that out often enough!). However, your colab does not contain the data folder, therefore, when reaching chunk 5 where the MGH COVID dataset is supposed to be processed, the fun stops... please fix this.

Reply: Thank you for your constructive feedback. Your positive comments are greatly appreciated. We're glad to incorporate the results of the MGH COVID dataset to enrich this study and improve the reproducibility of this paper.

Regarding the Google Colab setup, we appreciate the opportunity to clarify that users typically need to mount their own Google Drive to the Colab environment and load the required data. We included an option `clone_repo` in the original script to clone the data to the Colab environment but set it as `FALSE` by default to prevent unnecessary duplication for users who run the script locally or have previously cloned the repository. We realized that this option may not have been as noticeable as necessary for optimal user convenience. In this revision, we have made this setting more explicit to ensure the functionality is clear.

Reviewer #3 (Remarks on code availability):

The code and data on github is allowing for the reproduction of the authors figures and results. The code is written according to the state of the art in R coding containing a sufficient amount of documentation.

A provided colab Jupyter notebook link was missing the data to be functional, however, contained the results in its cache (I know that's not reproduction, but alas...).

Reply: Thank you for your feedback on the reproducibility of our figures and results. As noted earlier, we have updated the Colab notebook to highlight the `clone_repo` setting. Users can set this as `TRUE` to load the data into the Colab environment and reproduce our figures.

We thank you again for your careful review and helpful feedback, which has helped us enhance our manuscript in many different aspects.